# Rhythmic astrocytic GABA production synchronizes neuronal circadian timekeeping in the suprachiasmatic nucleus

Natalie Ness [1,2], Sandra Díaz-Clavero [1,2], Marieke M B Hoekstra [1,2] & Marco Brancaccio [1,2]✉

## Abstract

**Astrocytes of the suprachiasmatic nucleus (SCN) can regulate sleep-wake cycles in mammals. However, the nature of the information provided by astrocytes to control circadian patterns of behavior is unclear. Neuronal circadian activity across the SCN is organized into spatiotemporal waves that govern seasonal adaptations and timely engagement of behavioral outputs. Here, we show that astrocytes across the mouse SCN exhibit instead a highly uniform, pulse-like nighttime activity. We find that rhythmic astrocytic GABA production via polyamine degradation provides an inhibitory nighttime tone required for SCN circuit synchrony, thereby acting as an internal astrocyte zeitgeber (or "astrozeit"). We further identify synaptic GABA and astrocytic GABA as two key players underpinning coherent spatiotemporal circadian patterns of SCN neuronal activity. In describing a new mechanism by which astrocytes contribute to circadian timekeeping, our work provides a general blueprint for understanding how astrocytes encode temporal information underlying complex behaviors in mammals.**

**Keywords** Circadian Clock; Astrocyte; GABA; MAO-B; SCN
**Subject Category** Neuroscience

## Introduction

Astrocytes are critical players in the specification of complex behaviors, such as cognitive performance, feeding/fasting and sleep/wake cycles (Nagai et al, 2021). However, the astrocyte–neuronal communication mechanisms that regulate these processes are unclear.

We have previously found that astrocytes are capable of cell-autonomously driving circadian patterns of rest-activity cycles (Brancaccio et al, 2017, 2019). In genetically arrhythmic *Cry1/2* -null mice, rescuing *Cry1* expression only in astrocytes of the hypothalamic suprachiasmatic nucleus (SCN) is sufficient to restore rest-activity cycles with an intrinsic periodicity

distinguishable (shorter) from similarly treated neurons (Brancaccio et al, 2019). Furthermore, loss of function studies with astrocyte-specific conditional knockout of clock gene *Bmal1* showed altered rest-activity behavior in temporal isolation, including lengthened periodicity and dampened expression of clock genes in the SCN (Tso et al, 2017; Barca-Mayo et al, 2017).

While the available evidence demonstrates that astrocytes can act as a primary source of circadian temporal information, propagated from the SCN to specify circadian cycles of behavior (Brancaccio et al, 2017, 2019), a coherent model addressing the nature of astrocyte-released information, as well as its integration within the existing SCN neuronal circuitry, is missing.

Here, we first focused on investigating how neuronal and astrocytic circadian activity is concertedly regulated in space and time by simultaneously imaging neurons and astrocytes in organotypic SCN explants. We found that astrocytes show a remarkable uniformity in their spatiotemporal activity across the SCN, as opposed to the neuronal orderly phase wave progression spanning multiple hours. The spatiotemporal uniformity of astrocyte activity led us to hypothesize the presence of an internal circadian zeitgeber (shortened to "astrozeit") secreted by SCN astrocytes to synchronize neurons. As SCN neurons are mostly GABAergic (>95%) (Moore and Speh, 1993), we reasoned that the main site of action of such an astrozeit would likely be the GABAergic synapse, a critical site for astrocyte–neuronal communication (Mederos and Perea, 2019). However, we found presynaptic calcium and extracellular GABA levels to be dissociated temporally, peaking during the subjective day and night, and in space-time, following the orderly neuronal progression versus astrocytic homogenous phase, respectively. Moreover, inhibiting GABAergic neurotransmission desynchronized neuronal activity and clock gene expression across the network, but left circadian oscillations of extracellular GABA unaffected. We therefore hypothesized that extracellular GABA rhythms may be directly generated by astrocytes. We found that SCN astrocytes specifically express a noncanonical pathway for the synthesis of GABA from the degradation of polyamines (Lee et al, 2010; Yang et al, 2023; Jo et al, 2014; Nam et al, 2020). Inhibition of this pathway disrupts circadian oscillations of extracellular GABA in the SCN, and elicits neuronal hyperactivation and desynchronization across the SCN circuit. Our findings point to astrocytic GABA biosynthesis by

[1]Department of Brain Science, Imperial College London, London, UK. [2]UK Dementia Research Institute at Imperial College London, London, UK.
✉E-mail: m.brancaccio@imperial.ac.uk

polyamine degradation as a key metabolic pathway involved in astrocyte-mediated circadian synchronization of the SCN circuit.

# Results

## Circadian rhythms of SCN astrocytes show a uniform phase, distinct from spatiotemporal phase waves of neuronal activity

Circadian rhythms of clock gene expression and neuronal activity follow coherent spatiotemporal patterns in the SCN, giving rise to characteristic phase waves of synchronized activity progressing from the dorsomedial to the ventrolateral aspect of the nucleus (Evans et al, 2011; Enoki et al, 2012; Pauls et al, 2014). This orderly spatiotemporal pattern of activity underpins key aspects of circadian function, including robust timekeeping and the delivery of precisely timed circadian signals to downstream brain targets (VanderLeest et al, 2007; Myung et al, 2015; Hastings et al, 2018). Moreover, variance in the relative phases of neuronal oscillators within the SCN is essential to transform variations in day length into adaptive downstream seasonal physiology, a key example of circuit plasticity underpinning daily behavior in mammals (Meijer et al, 2010).

To investigate whether astrocytes also show coherent organization of their spatiotemporal activity across the SCN, we used organotypic SCN explants, which maintain functional circadian rhythms and spatiotemporal organization for several weeks (Brancaccio et al, 2013). SCN slices were transduced with genetically encoded reporters of astrocyte activity, as well as with co-expressed reporters of neuronal activity or clock gene expression. We then performed longitudinal multimodal live imaging to simultaneously record circadian activity in neurons and astrocytes across the SCN over several weeks. All the reporters showed strong circadian oscillations with relative phase relationships consistent with what has previously been shown (Brancaccio et al, 2017; Hastings et al, 2018) (Appendix Fig. S1A–C and Appendix Table S1).

To identify differences in activity across regions of the SCN, we used unsupervised K-Means clustering, adapting a method to identify spatiotemporal clusters of clock gene expression in the SCN (Foley et al, 2011). Consistent with previous findings (Evans et al, 2011; Enoki et al, 2012; Pauls et al, 2014), we identified clock gene expression phase clusters spreading about 3 h from the dorsomedial to the ventrolateral SCN in SCN slices from the PER2::LUC knock-in mouse line (Appendix Fig. S1F and Appendix Table S1). As expected, co-detected circadian rhythms of neuronal intracellular calcium, monitored by the genetically encoded reporter Syn-jRCaMP1a (Dana et al, 2016; Brancaccio et al, 2017), mirrored this spatiotemporal organization, with similar inter-cluster phase dispersal, as measured by the circular variance of cluster phases (Appendix Fig. S1F,G).

In contrast, astrocytic intracellular calcium, simultaneously measured across the SCN by the astrocyte-restricted, membrane-tethered intracellular indicator gfaABC1D-lck-GCaMP6f exhibited highly synchronous patterns, characterized by a uniform phase (Fig. 1A). Extracellular glutamate, released by astrocytes in the SCN (Brancaccio et al, 2017) and measured by the astrocytic membrane-tethered gfaABC1D-iGluSnFR also showed uniform spatiotemporal

patterns (Fig. 1B). When compared to neuronal calcium, which showed large temporal variance across cluster time series (captured by the wider standard deviation in Fig. 1C), co-detected astrocytic calcium and extracellular glutamate rhythms showed much smaller variability (Fig. 1C,E). To compare the inter-cluster phase dispersal in astrocytic versus neuronal reporters, we quantified the circular variance of the clusters across co-recorded reporters (Fig. 1D,F; Appendix Table S1) and confirmed a significantly smaller variance in both astrocyte reporters compared to co-detected neuronal calcium. Based on these data, a probability density function (PDF) was derived to illustrate the distribution of expected phase variances for each of the reporters. This showed a significantly sharper distribution of phase variances for astrocytic reporters when compared to co-detected neuronal calcium (Fig. 1D,F; Appendix Table S1). Thus, our findings highlight a significant difference between neuronal and astrocytic activity within the SCN :neurons are organized in spatiotemporal phase waves extending over 3 h, whereas astrocytes share a common phase.

We ruled out any contributions to this phase distribution from uneven viral transductions of neuronal and astrocytic reporters across different SCN regions, as expression levels were homogeneous throughout the SCN tissue (Fig. EV1A–C). Moreover, a uniform phase distribution was consistently found across different slices independently of their position along the anterior-posterior SCN axis for both gfaABC1D-lck-GCaMP6f and gfaABC1D-iGluSnFR (Fig. EV1D–F; Appendix Table S1).

## Circadian oscillations of presynaptic calcium and extracellular GABA show distinct spatiotemporal patterns, resembling the neuronal versus astrocytic dichotomy

The strong mismatch between astrocytic phase uniformity and neuronal phase dispersal suggests a complex temporal interplay between astrocytes and the neuronal SCN circuitry, with variable phase relationships between the two cell types across the SCN.

Synapses are a critical interface in astrocyte–neuronal interplay (Semyanov and Verkhratsky, 2021) and we previously proposed that glutamate released by SCN astrocytes may promote the synaptic release of GABA via activation of presynaptic NMDA-NR2C receptors (Brancaccio et al, 2017). To understand the circuit-level organization of the SCN, we aimed to monitor the circadian dynamics of synaptic activity and neurotransmitter release. To do so, we transduced SCN slices with genetically encoded reporters of presynaptic calcium (EF1α-DIO-Synaptophysin::GCaMP6s, co-expressed with Syn-mCherry::Cre to restrict its expression to neurons, referred to as Syp::GCaMP6s) and extracellular GABA, measured by the membrane-tethered Syn-GABASnFR (Marvin et al, 2019; Patton et al, 2023), to evaluate the spatiotemporal distribution of synaptic activity and GABAergic tone.

Both indicators showed strong circadian oscillations (Appendix Fig. S1D,E) with the same period and similarly robust rhythms as co-expressed PER2::LUC or Syn-jRCaMP1a (Fig. 2A; Appendix Table S1). Relative to co-expressed PER2::LUC, which peaks at circadian time 12 (CT12), presynaptic calcium peaked at CT6 (Fig. 2B), consistent with data from (Brancaccio et al, 2013, 2017), whereas extracellular GABA peaked at CT19 (Figs. 2B and EV2A), thus indicating a temporal mismatch between synaptic activity and extracellular GABA levels.

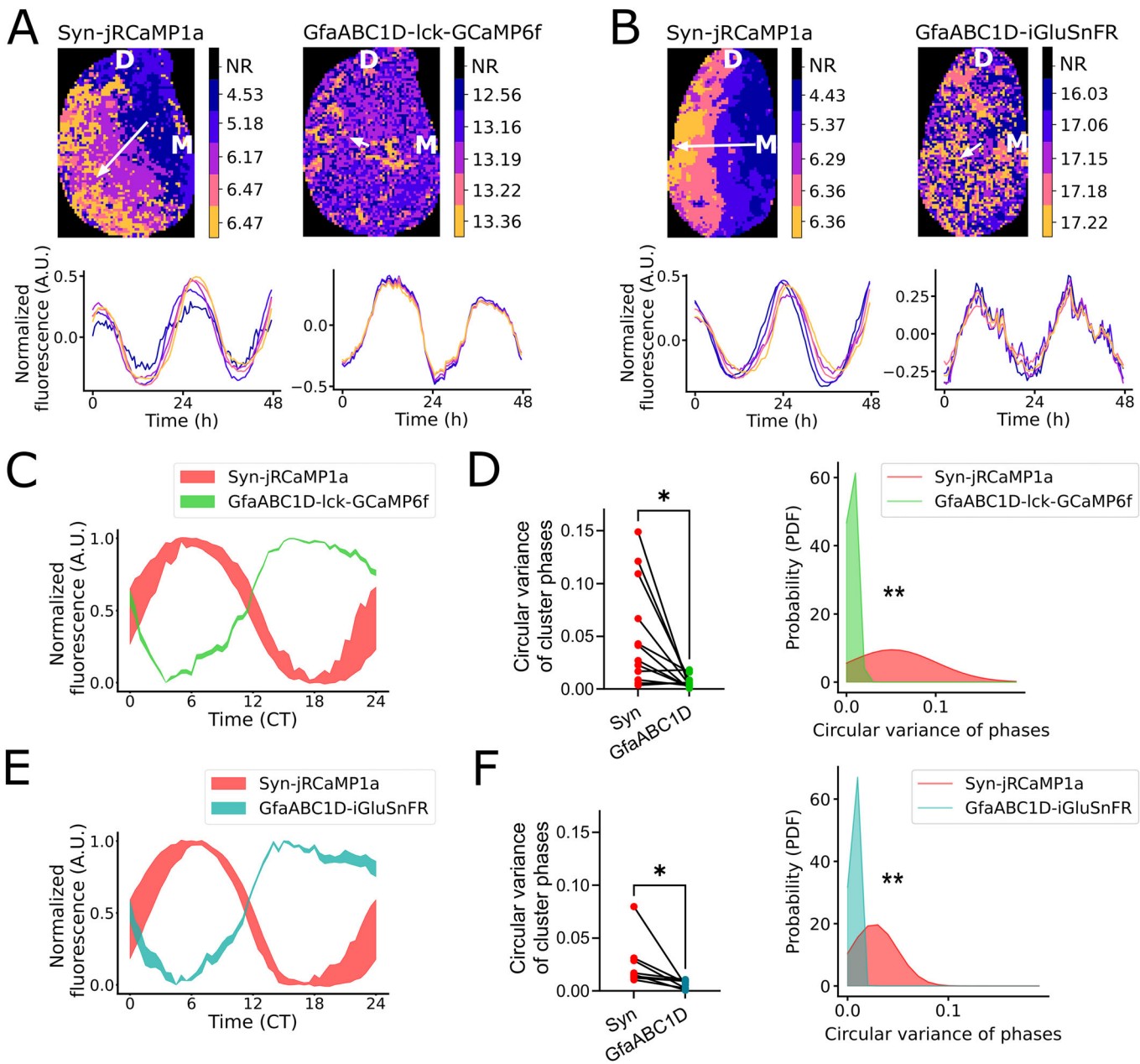

**Figure 1. Astrocytes and neurons of the SCN show distinct patterns of network synchronization.**

(A, B) Representative circadian phase map of Syn-jRCaMP1a co-detected with GfaABC1D-lck-GCaMP6f (A), or GfaABC1D-iGluSnFR (B), in SCN slices. One SCN is shown (top: dorsal—D, right: medial—M), showing dorsomedial to ventrolateral spatiotemporal progression of neuronal activity (reported by Syn-jRCaMP1a), as opposed to uniform phase distribution of astrocytic activity (reported by GfaABC1D-lck-GCaMP6f and GfaABC1D-iGluSnFR). The color bar shows the circadian phase of each cluster. NR = non-rhythmic. White vectors indicate the direction of phase progression. Time series of each cluster with the corresponding color is shown below. (C) Representative standard deviation of cluster time series within SCN co-expressing Syn-jRCaMP1a and GfaABC1D-lck-GCaMP6f, showing less variance across clusters in astrocytic calcium compared to neuronal calcium. CT = Circadian time. (D) Inter-cluster phase dispersal (measured by circular variance) of co-detected Syn-jRCaMP1a and GfaABC1D-lck-GCaMP6f (left), with PDF of cluster phase variance (right). $N = 12$ SCN slices. Left panel shows paired two-tailed $t$ test, $P = 0.0105$. Right panel shows Kolmogorov–Smirnov test, $P = 0.0079$. (E) Representative standard deviation of cluster time series within a slice co-expressing Syn-jRCaMP1a and GfaABC1D-GluSnFR, showing similarly reduced variance across clusters of astrocytic glutamate, when compared to neuronal calcium. CT = Circadian time. (F) Inter-cluster phase dispersal of co-detected Syn-jRCaMP1a and GfaABC1D-iGluSnFR (left), with PDF of cluster phase variance (right). $N = 8$ SCN slices. Left panel shows paired two-tailed $t$ test, $P = 0.0477$. Right panel shows Kolmogorov–Smirnov test, $P = 0.0025$. Each data point represents one SCN slice. *$P < 0.05$, **$P < 0.01$. A detailed statistical report for this figure is provided in Appendix Table S1. Source data are available online for this figure.

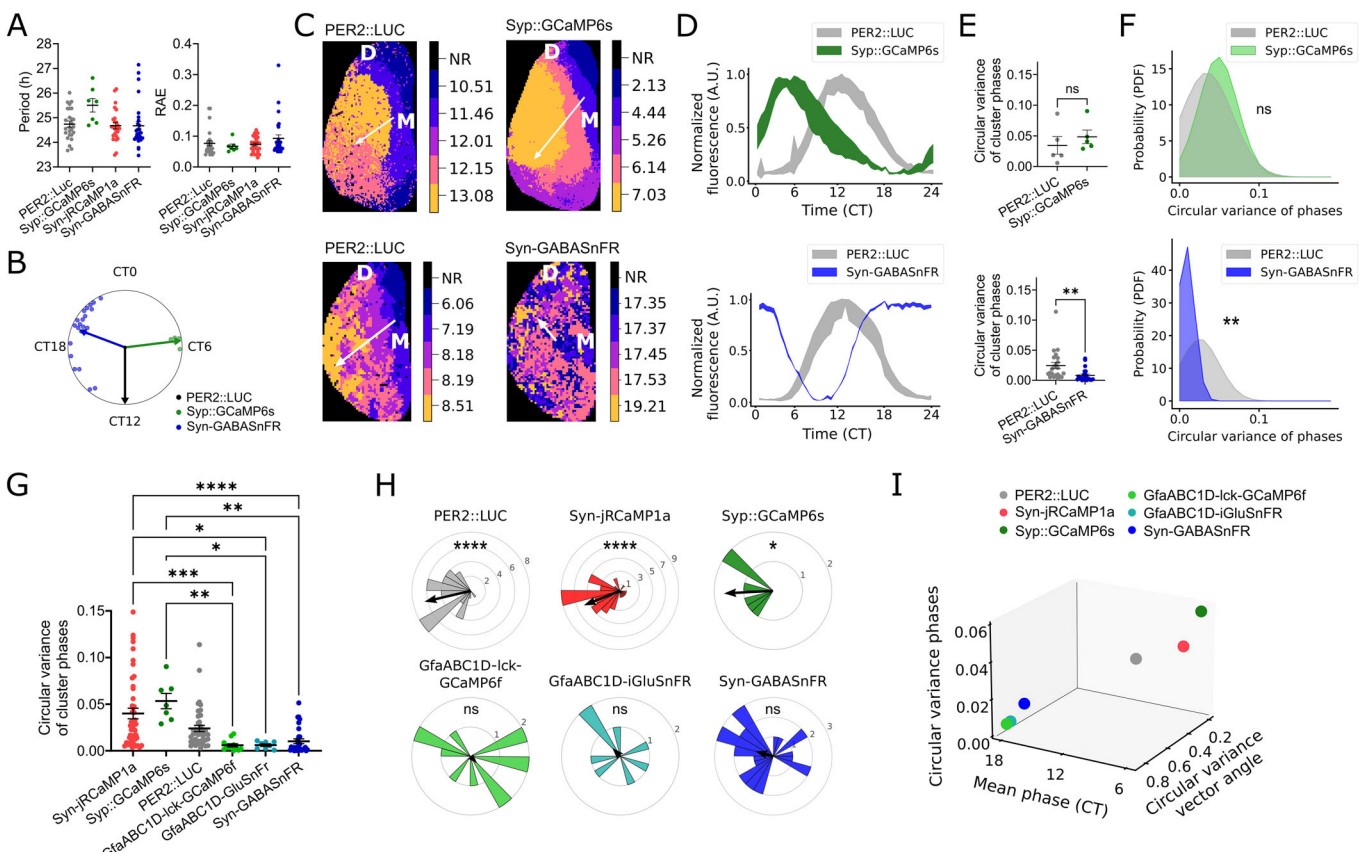

**Figure 2. Circadian rhythms of extracellular GABA co-segregate with reporters of astrocyte activity and not with neuronal ones.**

(A) Period and relative amplitude error (RAE) of circadian oscillations of PER2::LUC, Syp::GCaMP6s, Syn-jRCaMP1a and Syn-GABASnFR. One-way mixed-effects model with matching, and post hoc Tukey's test shown. All non-significant. Each dot presents one SCN slice. PER2::LUC (N = 26 SCN slices), Syp::GCaMP6s (N = 7), Syn-jRCaMP1a (N = 25) and Syn-GABASnFR (N = 28). (B) Rayleigh plot showing circadian phase of Syp::GCaMP6s (dark green) and Syn-GABASnFR (blue) rhythms, relative to co-detected PER2::LUC. Each point indicates 1 SCN slice. (C) Representative circadian phase cluster map of co-detected PER2::LUC and Syp::GCaMP6s (top) or Syn-GABASnFR (bottom). One SCN nucleus is shown (dorsal (D) and medial (M) area indicated). Color bars indicate cluster phases, NR = non-rhythmic. White arrow indicates the direction of the phase progression. (D) Representative standard deviation of cluster time series of co-detected PER2::LUC and Syp::GCaMP6s (top) or Syn-GABASnFR (bottom). (E) Inter-cluster phase dispersal (measured by circular variance) of co-detected PER2::LUC and Syp::GCaMP6s (N = 5 SCN slices, P = 0.480), or Syn-GABASnFR (N = 22, P = 0.0079). Paired two-tailed t test shown. (F) PDF of cluster phase variance for each co-detected reporter, with Kolmogorov–Smirnov test: Syp::GCaMP6s P = 0.357 and Syn-GABASnFR P = 0.00236. (G) Inter-cluster phase dispersal of Syn-jRCaMP1a (N = 46 SCN slices), Syp::GCaMP6s (N = 7 SCN slices), PER2::LUC (N = 42 SCN slices), GfaABC1D-lck-GCaMP6f (N = 12 SCN slices), GfaABC1D-GluSnFR (N = 8 SCN slices) and Syn-GABASnFR (N = 30 SCN slices). Mixed-effects analysis with matching, P < 0.0001, and Tukey's post hoc test shown. Significant comparisons in order shown top to bottom: ****P < 0.0001, **P = 0.0029, *P = 0.0157, *P = 0.0143, ***P = 0.001, and **P = 0.0035. (H) Circular histogram of directionality of phase progression across the SCN (see representative white arrows in (C)). Frequency of SCN slices within bar indicated by y-axis circle labels. The vector angle indicates the mean direction, length of the vector indicates circular dispersion. Rayleigh test of uniformity for PER2::LUC P < 0.0001, Syn-jRCaMP1a P < 0.0001, Syp::GCaMP6s P = 0.0151, GfaABC1D-lck-GCaMP6f P = 0.753, GfaABC1D-iGluSnFR P = 0.763, and Syn-GABASnFR P = 0.0870. (I) Correlation of mean circular variance of cluster phases, circular variance of phase wave directionality and mean phase (CT). All scatter graphs show mean ± SEM. ns = non-significant, *P < 0.05, **P < 0.01, ***P < 0.001, ****P < 0.0001. A detailed statistical report for this figure is provided in Appendix Table S1. Source data are available online for this figure.

Moreover, while presynaptic calcium showed similar spatio-temporal patterns to co-detected PER2::LUC, extracellular GABA did not show a spatiotemporal phase wave, and instead displayed phase-synchronous rhythms across the SCN, as previously shown for astrocytic calcium and extracellular glutamate (Fig. 2C,D). The variance in inter-cluster phase dispersal was as expected in presynaptic calcium, but significantly lower for extracellular GABA rhythms compared to co-detected PER2::LUC and Syn-jRCaMP1a (Figs. 2E and EV2B–D), which was confirmed by a significantly sharper PDF of phases for GABA clusters compared to co-detected PER2::LUC or Syn-jRCaMP1a clusters (Figs. 2F and EV2E; Appendix Table S1). In contrast, phase variances between presynaptic

calcium and PER2::LUC rhythms were highly overlapping (Fig. 2E,F and Appendix Table S1). This suggests that extracellular GABA rhythms are highly synchronized across the SCN and not coupled to presynaptic activity. To further confirm this using an alternative approach, we also evaluated the phase distribution across single cells within the SCN and found a significantly higher synchronization of GABA rhythms across cells, when compared to co-recorded neuronal calcium rhythms (Fig. EV2F,G). As for the other astrocytic reporters used, we consistently observed a homogenous phase distribution of GABA rhythms regardless of the location of the SCN slice along the anterior-posterior axis (Fig. EV1G; Appendix Table S1).

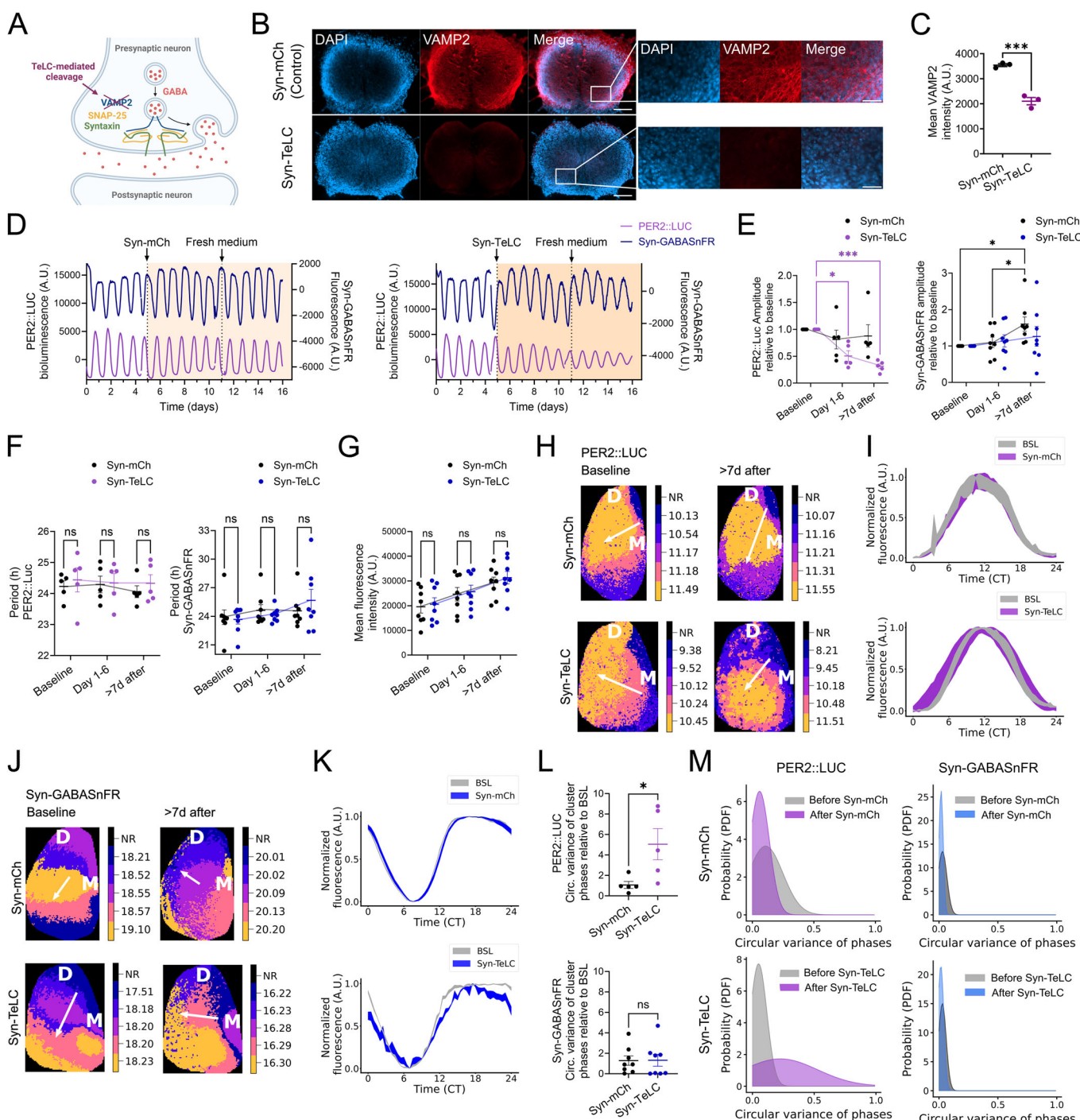

Comparing the circuit-level spatiotemporal organization of circadian rhythms across all reporters, we find that extracellular GABA rhythms co-segregate with astrocytic rhythms and differ from neuronal rhythms in three key ways. First, like other astrocyte reporters, GABA rhythms have a uniform phase, as opposed to neuronal reporters showing significantly higher inter-cluster phase variance (Fig. 2G; Appendix Table S1). Second, neuronal reporters showed a consistent dorsomedial to ventrolateral direction of their phase wave, whereas astrocytic reporters and GABA did not show a

coherent phase progression (Fig. 2H; Appendix Table S1). Third, GABA rhythms peak at nighttime together with astrocytic activity and in anti-phase to neuronal rhythms peaking during the daytime (Appendix Fig. S1A,D; Fig. 2B). Taken together, these findings show a clear segregation of astrocyte versus neuronal activity within the SCN (Fig. 2I).

Together, our findings indicate a strong discrepancy between circadian timing and spatial distribution of synaptic activity and GABA, with a mean 13 h difference in phase, suggesting that timing

**Figure 3.  Disrupting synaptic GABA transmission via tetanus toxin light chain desynchronizes PER2::LUC clusters without affecting circadian oscillations of GABA.**

(A) Schematic showing mechanism of blockade of GABAergic synaptic transmission by TeLC-dependent cleavage of the SNARE complex protein VAMP2. (B) Representative widefield images and insets of SCN slices expressing Syn-mCherry (control, top) or Syn-TeLC-mCherry (bottom) labeled with anti-VAMP2 antibody and DAPI, scale bar = 200 μm. Inset scale bar = 20 μm. (C) Quantification of mean VAMP2 intensity within the SCN. $N_{Syn-mCh}$ = 3, $N_{Syn-TeLC}$ = 3 SCN slices, two-tailed $t$ test, $P$ = 0.0008. (D) Representative detrended time series of SCN slices expressing reporters for extracellular GABA (Syn-GABASnFR) and PER2::LUC before and after treatment with Syn-mCherry (left) or Syn-TeLC-mCherry (right). (E) Amplitude relative to baseline of rhythms of PER2::LUC (left) and Syn-GABASnFR (right). Two-way ANOVA with post hoc Šidák's test, PER2::LUC: $P$ = 0.0152 for baseline-Day 1-6; $P$ = 0.0002 for baseline->7 d after; Syn-GABASnFR: $P$ = 0.042 for BSL->7 d after, $P$ = 0.0249 for Day 1-6->7 d after. (F) Circadian period of PER2::LUC (left) and Syn-GABASnFR rhythms (right). $N_{PER2::LUC}$ = 5 SCN slices, $N_{Syn-GABASnFR}$ = 8 SCN slices. Two-way ANOVA with post hoc Šidák's test shown. (G) Mean fluorescence intensity of Syn-GABASnFR signal across timepoints. $N_{PER2::LUC}$ = 5, $N_{Syn-GABASnFR}$ = 8 SCN slices. Two-way ANOVA with post hoc Šidák's test shown. (H) Representative circadian phase cluster map of PER2::LUC before and >7 days after transduction with Syn-TeLC-mCherry (bottom) or mCherry control (top). One SCN nucleus is shown, orientation as indicated (dorsal—D, medial—M). Color bar indicates circadian phases of clusters, NR = non-rhythmic. White vector indicates the directionality of phase progression. (I) Representative standard deviation of cluster time series shown in (H). (J) Representative circadian phase cluster map of Syn-GABASnFR co-detected with PER2::LUC shown in (H) before and >7 days after viral transduction. (K) Representative standard deviation of cluster time series shown in (J). (L) Inter-cluster phase dispersal of PER2::LUC (top) or Syn-GABASnFR (bottom) relative to baseline, with two-tailed $t$ test, PER2::LUC $P$ = 0.0338, Syn-GABASnFR $P$ = 0.259; $N_{PER2::LUC}$ = 5, $N_{Syn-GABASnFR}$ = 8 SCN slices. (M) PDF of cluster phase variance shown in (L); $N_{PER2::LUC}$ = 5, $N_{Syn-GABASnFR}$ = 8 SCN slices. All data shown mean ± SEM unless otherwise indicated. For longitudinal data, connecting lines are shown between means. ns = non-significant. *$P$ < 0.05, ***$P$ < 0.001. A detailed statistical report for this figure is provided in Appendix Table S1. Source data are available online for this figure.

of synaptic release is likely not the major determinant of extracellular GABA levels. Instead, GABA rhythms closely follow rhythms of astrocytic activity (Fig. 2I), pointing towards an astrocytic origin of the detected GABA rhythms in the SCN.

## Inhibition of GABAergic neurotransmission desynchronizes clock gene expression but does not affect circadian rhythms of GABA in the SCN

Given the observed temporal discrepancy between presynaptic calcium and extracellular GABA, we proceeded to functionally test whether disruption of synaptic GABAergic neurotransmission affects extracellular GABA rhythms.

We first transduced SCN slices with AAVs expressing the tetanus toxin light chain (Syn-TeLC-mCherry), which prevents membrane fusion of synaptic vesicles by cleavage of vesicular synaptic membrane protein 2 (VAMP2) (Kobayashi et al, 2008; Pashkovski et al, 2020) and demonstrated a significant reduction of VAMP2 expression in TeLC-treated SCNs (Fig. 3A–C). We then longitudinally monitored the circadian rhythmicity of co-expressed Syn-GABASnFR and PER2::LUC before and after TeLC treatment and compared it with SCN slices expressing Syn-mCherry as a control (Fig. 3D–M; Appendix Table S1).

SCN slices expressing Syn-TeLC-mCherry showed a progressively reduced amplitude of PER2::LUC circadian oscillations, which was unaffected in control SCNs. Interestingly, this reduced amplitude was not accompanied by a similar reduction in amplitude of co-detected GABA in TeLC-treated SCNs (Fig. 3D,E; Appendix Table S1; Movie EV1). Period and rhythm robustness (measured by relative amplitude error, RAE) were not affected in any condition (Fig. 3F; Appendix Fig. S2 and Appendix Table S1). While the overall fluorescence intensity of Syn-GABASnFR slightly increased over time (due to sustained expression of AAV), it was not significantly different across experimental conditions (Fig. 3G; Appendix Table S1). Thus, preventing synaptic release of GABA weakened overall circadian amplitude of PER2::LUC expression but did not affect circadian oscillations of co-detected extracellular GABA.

To test whether the observed amplitude reduction in PER2::LUC could reflect impaired circadian synchronization due to decreased synaptic coupling, we performed unsupervised clustering of time series before and after transduction, and found clusters of

PER2::LUC to be desynchronized, whereas GABA clusters remained highly synchronous across the SCN (Fig. 3H–L; Appendix Table S1). Coherent with this increased inter-cluster phase variance, the PDF of phase variances showed that PER2::LUC clusters shifted to a less sharp, broader distribution, indicating increased variability in phase variances, whereas the GABA PDF was not broadened (Fig. 3M). Thus, TeLC expression desynchronized circadian rhythms of PER2::LUC across the SCN, but left co-detected GABA rhythms unaffected, consistent with their predicted non-synaptic origin.

## The noncanonical polyamine-to-GABA biosynthetic pathway is specifically expressed in hypothalamic and SCN astrocytes

Given that extracellular GABA rhythms closely followed the uniform nighttime circadian pattern of astrocytic activity, as opposed to neuronal and presynaptic activity, and persisted upon inhibition of synaptic GABA vesicle release, we investigated whether the observed GABA rhythms may directly originate from astrocytes by examining astrocytic molecular pathways of GABA biosynthesis. Astrocytes can directly synthesize GABA via noncanonical monoamine oxidase B (MAO-B)-dependent or D-amino acid oxidase (DAO)-dependent degradation of the polyamine putrescine (Koh et al, 2023). MAO-B-mediated GABA biosynthesis in astrocytes occurs under physiological conditions in the striatum and cerebellum, and is observed in the hippocampus, substantia nigra and cortex in mouse models of neurodegeneration (Koh et al, 2023). In both contexts, astrocytic GABA synthesis is associated with tonic inhibition of neurons, suggesting astrocytes can synthesize and release GABA to modulate neuronal activity (Yoon et al, 2014; Woo et al, 2018; Nam et al, 2020; Yang et al, 2023).

To examine whether these astrocytic GABA synthesis pathways are expressed in SCN astrocytes, we analyzed publicly available data from a recent large single-cell RNA sequencing dataset of the mouse brain, consisting of a total of ~2.3 million cells, by the Allen Brain Institute (Yao et al, 2023). We compared gene expression of relevant biosynthetic pathways in hypothalamic astrocytes, SCN neurons and hypothalamic GABAergic neurons (Fig. 4A,B). Hypothalamic GABAergic neurons, SCN neurons and hypothalamic astrocytes showed characteristic expression of cell-type specific

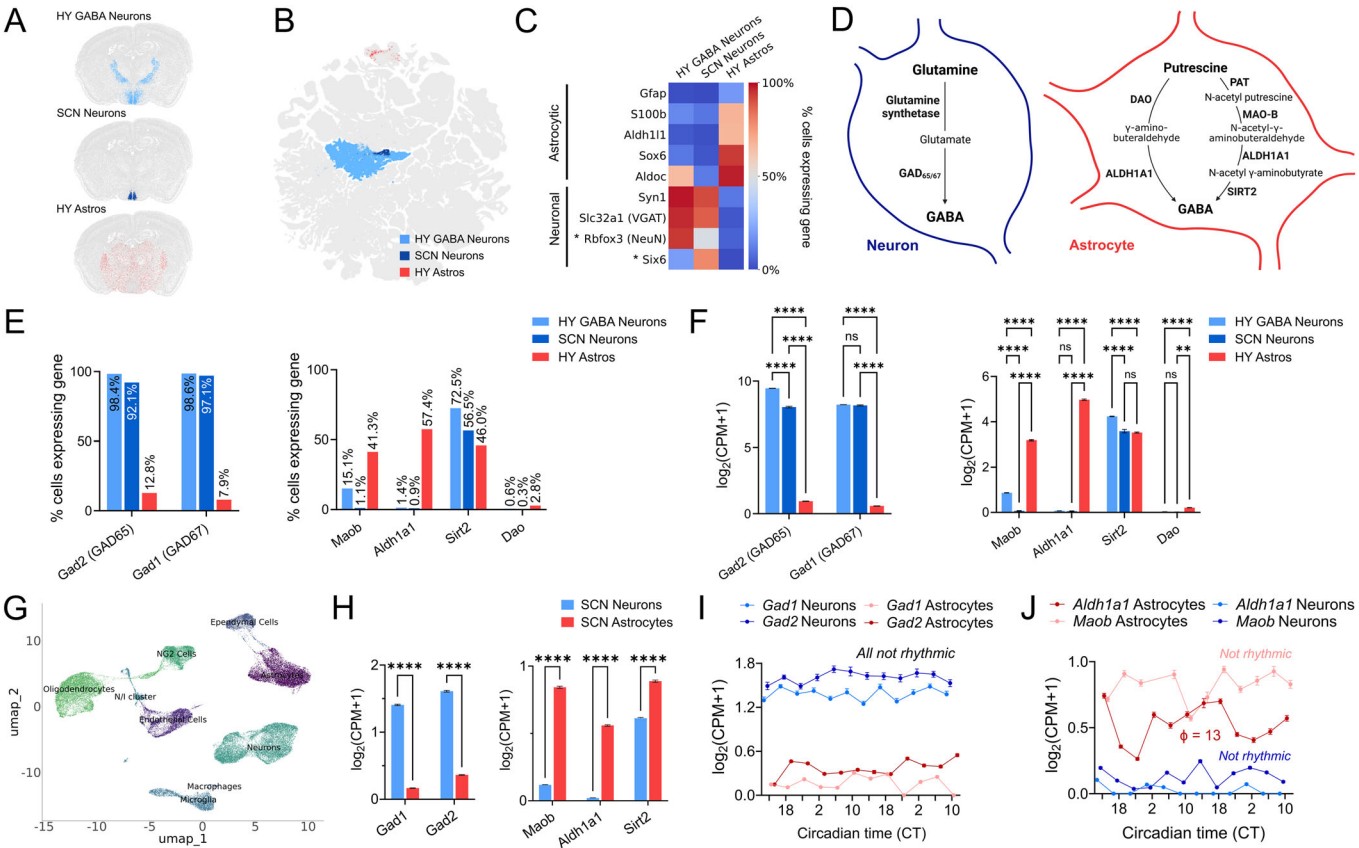

**Figure 4. The polyamine-to-GABA biosynthetic pathway is specifically expressed in hypothalamic astrocytes but not in SCN neurons.**

(A) Schematic of data from (Yao et al, 2023) spatial cell-type atlas of the mouse brain acquired using MERFISH, indicating the approximate positions of hypothalamic GABAergic neurons, SCN neurons and astrocytes in a mouse brain. (B) UMAP plot of data from (Yao et al, 2023) scRNA-Seq dataset, showing hypothalamic GABAergic neurons ('HY GABA Neurons', light blue), SCN neurons (blue) and hypothalamic astrocytes ('HY Astros', red), used for analyses from (C–F). (C) Heatmap showing the percentage of cells expressing various cell-type specific markers in hypothalamic GABAergic neurons, SCN neurons and hypothalamic astrocytes. Genes marked with * are known to be differentially expressed in SCN neurons. (D) Schematic depicting GABA biosynthetic pathways previously observed in neurons or astrocytes. Left diagram shows neuronal GABA synthesis from glutamine via GAD65/67 (canonical pathway), right diagram shows astrocytic GABA biosynthetic pathways from putrescine (noncanonical pathway). (E) Percentage of cells expressing genes from the GABA biosynthetic pathways shown in D by cell-type. GAD65/67 are highly expressed in neurons, but not astrocytes, while noncanonical MAO-B-dependent putrescine-to-GABA synthesis pathway components are specifically expressed in astrocytes. (F) Normalized gene expression levels of genes shown in (E). $N = 36{,}019$ hypothalamic GABAergic neurons, $N = 1836$ SCN neurons and $N = 20{,}549$ hypothalamic astrocytes. Two-way ANOVA with post hoc Šidák's test. (G) UMAP plot of SCN-restricted scRNA-Seq dataset (Wen et al, 2020), with cell-type annotation. (H) Normalized expression levels of genes involved in GABA biosynthesis. $N = 12{,}018$ SCN neurons and $N = 8429$ SCN astrocytes. Two-way ANOVA with post hoc Šidák's test shown. (I) Time series of normalized gene expression of neuronal GABA biosynthesis genes Gad1 and Gad2 in SCN neurons and astrocytes. $N = 12{,}018$ neurons and $N = 8429$ astrocytes with 238–2410 cells/timepoint. eJTK Cycle rhythmicity test with Benjamini–Hochberg correction, all $P > 0.05$. (J) Time series of normalized expression levels of astrocytic GABA biosynthesis genes Aldh1a1 and Maob in SCN neurons and astrocytes. $N = 12{,}018$ neurons and $N = 8429$ astrocytes with 238–2410 cells/ timepoint. eJTK Cycle rhythmicity test with Benjamini–Hochberg correction, $P = 0.022$ for Aldh1a1 in astrocytes peaking at CT13 (indicated as φ on the plot), all other time series $P > 0.05$. All data show mean ± SEM, ns = non-significant, **$P < 0.01$, ****$P < 0.0001$. A detailed statistical report for this figure is provided in Appendix Table S1. Source data are available online for this figure.

markers, (e.g., astrocytic Aldh1l1 and S100β versus neuronal Syn1 and Slc32a1 expression) (Fig. 4C). While the expression of neuronal markers across SCN neurons and hypothalamic GABAergic neurons was generally similar, well-known SCN-specific differences, such as Six6 expression specifically in SCN neurons (Conte et al, 2005) and low expression of the neuronal marker NeuN (Geoghegan and Carter, 2008) were recapitulated (Fig. 4C).

Having verified specific segregation of astrocytic vs neuronal marker gene expression in the SCN, we examined genes involved in canonical GABA biosynthesis across these two cell types (Fig. 4D) and found that while Gad1 and 2, encoding GAD67

and 65 (Koh et al, 2023), were expressed almost ubiquitously at high levels in SCN neurons, they were only scarcely expressed in hypothalamic astrocytes (Fig. 4E,F; Appendix Table S1). In contrast, when we examined the noncanonical molecular pathways involved in GABA synthesis from the polyamine putrescine (Bhalla et al, 2023), we found that enzymes of this pathway, such as Maob and Aldh1a1 were strongly and selectively expressed by hypothalamic astrocytes (comparable to S100β and Aldh1l1), but virtually absent in SCN neurons (<1% of neurons) (Fig. 4E,F; Appendix Table S1). Components of an alternative DAO-dependent GABA synthesis pathway from putrescine (Bhalla et al, 2023) were

expressed at very low levels in the hypothalamus, irrespective of cell type (Fig. 4E,F; Appendix Table S1).

To validate our findings and ensure that SCN astrocytes do not diverge from hypothalamic astrocytes in terms of GABA biosynthetic pathway expression, we also interrogated a second SCN-specific scRNA-Seq dataset for the expression of this pathway in SCN astrocytes (Wen et al, 2020) (Fig. 4G). We confirmed enriched expression of *Maob* and *Aldh1a1* in SCN astrocytes and high expression of *Gad1* and *Gad2* in SCN neurons, while *Sirt2* was expressed in both cell types (Fig. 4H; Appendix Table S1), thus confirming findings from the larger ABC Atlas dataset. As the (Wen et al, 2020) dataset contained 12 circadian timepoints, we were also able to assess time-of-day-dependent expression patterns. Neuronal expression of *Gad1* and *Gad2* was not rhythmic (Fig. 4I; Appendix Table S1). In contrast, *Aldh1a1* (but not *Maob*) was rhythmically expressed, with a peak of expression at CT13 (Fig. 4J; Appendix Table S1), consistent with the rising phase of measured GABA levels (Fig. 2D). Thus, analysis of the (Wen et al, 2020) dataset confirmed the cell-type specific dichotomy of GABA biosynthesis in SCN astrocytes versus neurons, and ruled out any time-of-the-day-dependent "ectopic" upregulation of the respective GABA pathways, which may have not been captured by the larger ABC Atlas (due to the limited timepoints available). Moreover, it also highlighted rhythmic expression of *Aldh1a1*, peaking during the subjective night, as a potential mechanism for the circadian modulation of polyamine-derived GABA synthesis in SCN astrocytes.

We further validated astrocytic expression of ALDH1A1 and MAO-B proteins in SCN slices (Fig. EV3A–F) by immunohisto-fluorescence staining. ALDH1A1 was expressed in the majority of SCN astrocytes labeled with a Gfap-mCherry::Cre-expressing AAV (~90%) (Fig. EV3A,B), and also colocalized with GFAP signal (>85%) (Fig. EV3C). MAO-B was also expressed in the majority, but slightly smaller percentage, of Gfap-mCherry::Cre[+] SCN astrocytes (~65%) (Fig. EV3D,E), and also highly overlapped with GFAP signal (>90%) (Fig. EV3F).

Further interrogation of the (Yao et al, 2023) scRNA-Seq dataset suggested that ~24.7% of hypothalamic astrocytes co-expressed *Aldh1a1* and *Maob* (Fig. EV3G), and a smaller percentage (~10%) co-expressed *Aldh1a1* and *Maob*, as well as key enzymes involved in glutamate synthesis from glutamine or GABA (Fig. EV3G). We further characterized expression of channels and transporters previously implicated in GABA release from astrocytes and found hypothalamic and SCN astrocytes express *Slc6a1* and *Slc6a11*, which encode GAT1 and GAT3 respectively, and may mediate GABA uptake or release (Patton et al, 2023; Héja et al, 2012; Wójtowicz et al, 2013; Barakat and Bordey, 2002), but not *Bestrophin1*, a GABA channel involved in astrocytic GABA release in other brain areas (Koh et al, 2023) (Fig. EV3H–J; Appendix Table S1 and "Discussion").

## Pharmacological inhibition of astrocytic GABA synthesis impairs circadian rhythms of extracellular GABA in the SCN

Having demonstrated the selective expression of key enzymes of the noncanonical polyamine-to-GABA pathway in SCN astrocytes, we pharmacologically inhibited MAO-B and ALDH1A1 and assessed effects on extracellular GABA rhythms in SCN slices. To do so, we used Selegiline (R-Deprenyl) and A37 (CM037), highly selective and potent inhibitors of MAO-B (Knoll and Magyar, 1972; Knoll et al, 1978; Szökő et al, 2018) and ALDH1A1 (Morgan and Hurley, 2015) respectively, and compared the observed effects to vehicle (DMSO)-treated samples (Fig. 5A).

Selegiline impaired extracellular GABA rhythms within 1–3 days of treatment, whereas vehicle-treated slices remained rhythmic (Fig. 5B; Movie EV2). By using an established rhythmicity test (eJTK cycle) (Hutchison et al, 2015), we compared the GABA signal against a set of cosine waves and found that the correlation between the signal and the tested cosine waves (τ coefficient), was significantly lower after treatment with Selegiline compared to DMSO, indicating that rhythms had a worse fit. Moreover, when compared to random noise data, GABA rhythms were no longer significantly rhythmic after treatment with Selegiline ($P > 0.00001$) (Fig. 5C; Appendix Table S1, see "Methods"). Selegiline treatment also significantly reduced GABA levels in SCN slices (Fig. EV4A–C). To investigate whether the reduction of GABA in the SCN was associated with a reactive astrocyte phenotype, we quantified GFAP expression in SCN slices treated with Selegiline and compared it with DMSO-treated slices. The SCN expresses substantial levels of GFAP under physiological conditions (Lavialle and Servière, 1993; Leone et al, 2006; Becquet et al, 2008), with increased expression in astrogliosis (Eeza et al, 2021; Stopa et al, 1999). We found no increase in GFAP levels or in the number of Gfap-mCherry::Cre-expressing cells, consistent with a lack of reactive astrogliosis (Fig. EV4A–C).

Consistent with the idea that Selegiline treatment selectively impairs GABA synthesis, we found that circadian rhythmicity of extracellular glutamate, released by astrocytes in the SCN (Brancaccio et al, 2017), was not significantly affected by the treatment (Fig. EV4D–H; Appendix Table S1).

Having shown that Selegiline specifically impairs circadian oscillations of extracellular GABA, we monitored effects on co-detected neuronal calcium (measured by Syn-jRCaMP1a) and clock gene expression (measured by PER2::LUC). We found that intracellular neuronal calcium retained overall circadian rhythmicity (eJTK cycle rhythmicity test $P$ value $P \le 0.0001$, before and after Selegiline), and amplitude and robustness were not significantly affected (Fig. 5D–G; Appendix Table S1). However, the period of circadian neuronal calcium oscillations was significantly shortened by Selegiline (Fig. 5H; Appendix Table S1). PER2::LUC oscillations also retained strong rhythmicity after treatment (eJTK cycle rhythmicity test $P$ value $P \le 0.0001$), but showed lower amplitude and robustness (higher RAE) compared to controls (Fig. 5I–L, Appendix Table S1), potentially indicating weakening of cellular clock gene expression rhythms across the SCN. As for neuronal calcium, the period of PER2::LUC oscillations was significantly shorter after Selegiline treatment (Fig. 5M; Appendix Table S1).

Treatment with ALDH1A1 inhibitor A37 significantly reduced the amplitude and robustness of circadian GABA oscillations (Fig. 6A–D; Appendix Table S1; Movie EV3). The amplitude and robustness of GABA rhythms was diminished immediately after treatment for 2–3 circadian cycles, albeit only transiently (Fig. 6B,C; Appendix Table S1). A37-treated slices only showed a slight but not significantly lower fit against cosine waves using a rhythmicity test, as indicated by a reduced coefficient τ (Fig. 6E; Appendix Table S1), with all SCN slices treated with A37 or DMSO vehicle remaining significantly rhythmic after treatment ($P \le 0.00001$). Period of the

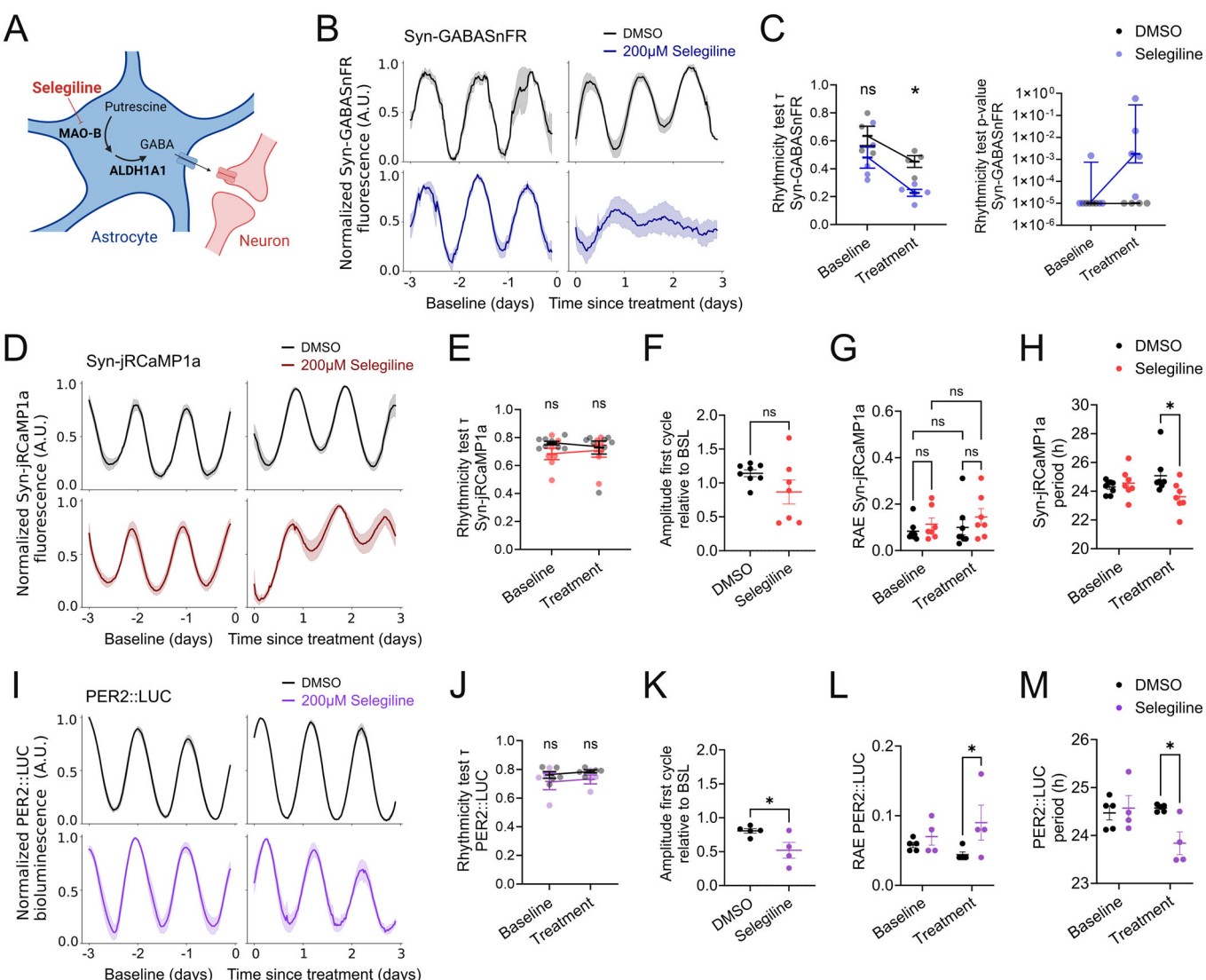

**Figure 5. Pharmacological inhibition of MAO-B abolishes circadian rhythms of extracellular GABA in SCN slices and shortens the circadian period of neuronal calcium and clock gene expression.**

(A) Schematic of Selegiline action on polyamine GABA biosynthesis in astrocytes. (B) Averaged, aligned time series of Syn-GABASnFR in SCN slices before and after treatment with 200 μM Selegiline (blue) or DMSO vehicle (black). $N_{DMSO} = 4$, $N_{Selegiline} = 5$ SCN slices. (C) Left panel shows τ values obtained from eJTK Cycle rhythmicity test on time series of Syn-GABASnFR before and within 1–3 days after treatment with Selegiline or DMSO. Right panel shows the $P$ value obtained from eJTK Cycle rhythmicity test empirically calculated against random noise data. $N_{DMSO} = 4$, $N_{Selegiline} = 5$ SCN slices. $P = 0.0303$ for DMSO vs Selegiline at treatment. (D) Averaged, aligned time series of neuronal calcium (Syn-jRCaMP1a) before and after treatment with 200 μM Selegiline (red) or DMSO (black). $N_{DMSO} = 8$, $N_{Selegiline} = 7$ SCN slices. (E) τ values obtained from eJTK Cycle rhythmicity test on time series of Syn-jRCaMP1a before and within 1–3 days after treatment with Selegiline or DMSO; $N_{DMSO} = 8$, $N_{Selegiline} = 7$ SCN slices. (F) Syn-jRCaMP1a amplitude of the first cycle of rhythms (over 30 h) after treatment with Selegiline or DMSO relative to baseline. $N_{DMSO} = 8$, $N_{Selegiline} = 7$ SCN slices. Two-tailed unpaired $t$ test, $P = 0.138$. (G) RAE of Syn-jRCaMP1a rhythms before and after treatment $N_{DMSO} = 8$, $N_{Selegiline} = 7$ SCN slices. (H) Period of Syn-jRCaMP1a rhythms before and after treatment, $P = 0.0176$ for DMSO vs Selegiline at treatment. $N_{DMSO} = 8$, $N_{Selegiline} = 7$ SCN slices (I) Averaged, aligned time series of PER2::LUC before and after treatment with 200 μM Selegiline (purple), or DMSO (black). $N_{DMSO} = 5$, $N_{Selegiline} = 4$ SCN slices. (J) τ values obtained from eJTK Cycle rhythmicity test on time series of PER2::LUC before and within 1–3 days after treatment with Selegiline or DMSO; $N_{DMSO} = 5$, $N_{Selegiline} = 4$ SCN slices. (K) PER2::LUC amplitude of first cycle of rhythms (over 30 h) after treatment with Selegiline or DMSO, relative to baseline. $N_{DMSO} = 5$, $N_{Selegiline} = 4$ SCN slices. Two-tailed unpaired $t$ test, $P = 0.0341$. (L) RAE of PER2::LUC rhythms before and after Selegiline treatment; $N_{DMSO} = 5$, $N_{Selegiline} = 4$ SCN slices; $P = 0.0443$ for DMSO vs Selegiline at treatment. (M) Period of PER2::LUC rhythms before and after Selegiline treatment, $N_{DMSO} = 5$, $N_{Selegiline} = 4$ SCN slices, $P = 0.0214$ for DMSO vs Selegiline at treatment. All graphs, including time series, show mean ± SEM, except right panel in (C), showing median ± interquartile range due to logarithmic scale. All graphs with multiple comparisons show two-way mixed-effects analysis with matching, with post hoc Šidák's test. ns=non-significant, *$P < 0.05$. A detailed statistical report for this figure is provided in Appendix Table S1. Source data are available online for this figure.

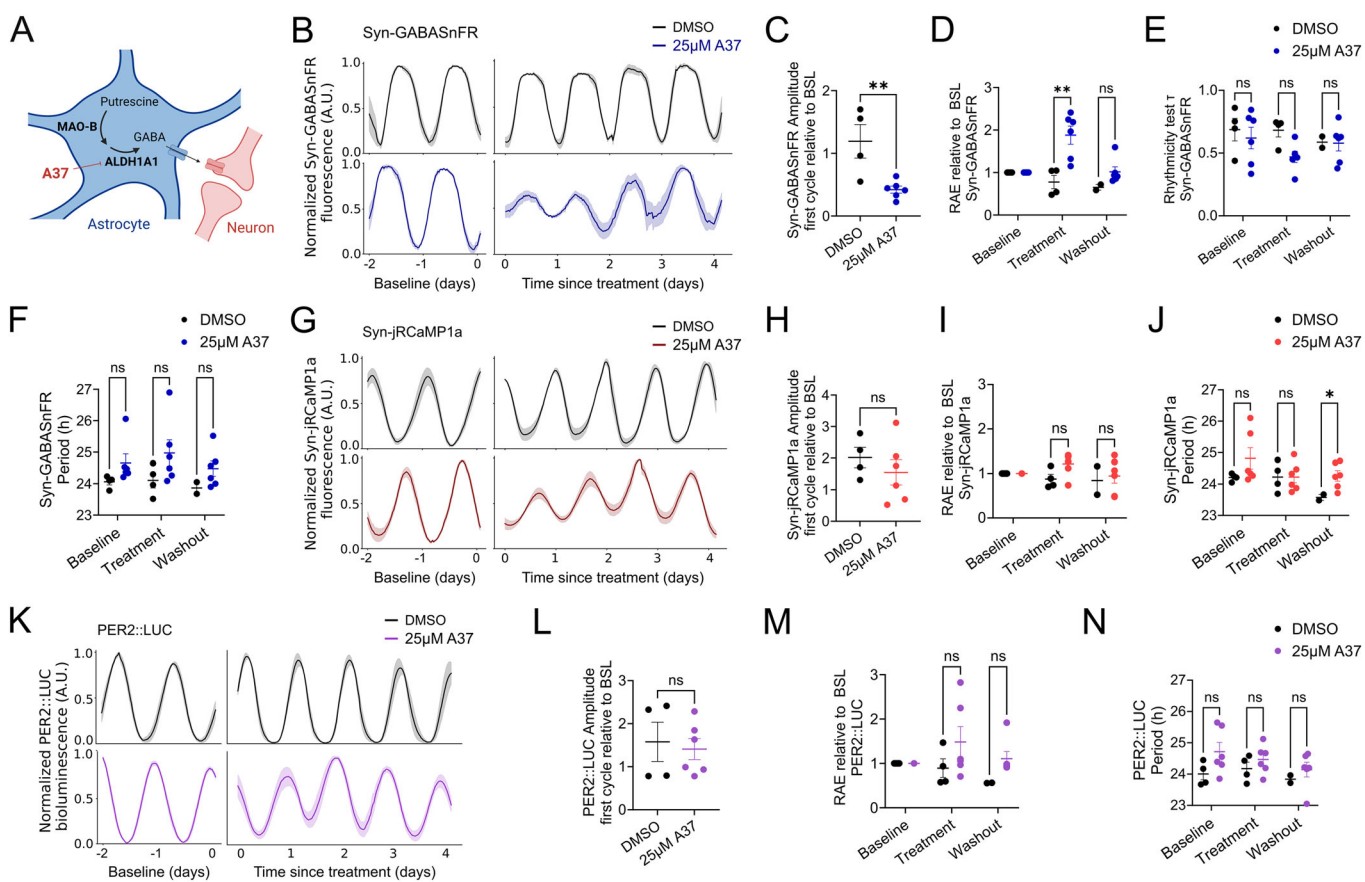

**Figure 6. Pharmacological inhibition of ALDH1A1 temporarily suppresses rhythms of extracellular GABA in SCN organotypic slices.**

(A) Schematic of A37 action on polyamine GABA biosynthesis in astrocytes. (B) Averaged, aligned time series of reporter of extracellular GABA (Syn-GABASnFR) in organotypic SCN slices before and after treatment with 25 μM A37 (blue) or DMSO vehicle (black). Baseline and treatment: $N_{DMSO} = 4$, $N_{A37} = 6$ SCN slices; washout: $N_{DMSO} = 2$, $N_{A37} = 6$ SCN slices. (C) Amplitude of first circadian cycle (over 30 h) after treatment with A37 or DMSO relative to baseline, $P = 0.0085$. $N_{DMSO} = 4$, $N_{A37} = 6$ SCN slices. (D) RAE before and after treatment, and after washout. Baseline and treatment: $N_{DMSO} = 4$, $N_{A37} = 6$ SCN slices; washout: $N_{DMSO} = 2$, $N_{A37} = 6$ SCN slices, $P = 0.0092$ for DMSO vs A37 after treatment. (E) τ values obtained from eJTK Cycle rhythmicity test on time series of Syn-GABASnFR before, after treatment and after washout of A37 or DMSO vehicle. Baseline and treatment: $N_{DMSO} = 4$, $N_{A37} = 6$ SCN slices; washout: $N_{DMSO} = 2$, $N_{A37} = 6$ SCN slices (F) Period of Syn-GABASnFR rhythms before and after treatment. Baseline and treatment: $N_{DMSO} = 4$, $N_{A37} = 6$ SCN slices; washout: $N_{DMSO} = 2$, $N_{A37} = 6$ SCN slices. (G) Averaged, aligned time series of co-detected neuronal calcium (Syn-jRCaMP1a) before and after treatment with A37 (red) or DMSO (black). $N_{DMSO} = 4$, $N_{A37} = 6$ SCN slices. (H) Amplitude of first cycle of rhythms (over 30 h) after treatment with A37 or DMSO relative to baseline, $N_{DMSO} = 4$, $N_{A37} = 6$ SCN slices, $P = 0.424$. (I) RAE of Syn-jRCaMP1a relative to baseline after treatment and washout of A37 or DMSO Baseline and treatment: $N_{DMSO} = 4$, $N_{A37} = 6$ SCN slices; washout: $N_{DMSO} = 2$, $N_{A37} = 6$ SCN slices. (J) Period of Syn-jRCaMP1a rhythms before and after treatment. $P = 0.0364$ for DMSO vs A37 after washout. Baseline and treatment: $N_{DMSO} = 4$, $N_{A37} = 6$ SCN slices; washout: $N_{DMSO} = 2$, $N_{A37} = 6$ SCN slices. (K) Averaged, aligned time series of co-detected PER2::LUC before and after treatment with A37 (purple) or DMSO (black). $N_{DMSO} = 4$, $N_{A37} = 6$ SCN slices. (L) Amplitude of first cycle of rhythms (over 30 h) after treatment with A37 or DMSO relative to baseline, $N_{DMSO} = 4$, $N_{A37} = 6$ SCN slices, $P = 0.7328$. (M) RAE of PER2::LUC relative to baseline after treatment and washout of A37 or DMSO. Baseline and treatment: $N_{DMSO} = 4$, $N_{A37} = 6$ SCN slices; washout $N_{DMSO} = 2$, $N_{A37} = 6$ SCN slices. (N) Period of PER2::LUC rhythms before and after treatment. Baseline and treatment: $N_{DMSO} = 4$, $N_{A37} = 6$ SCN slices; washout: $N_{DMSO} = 2$, $N_{A37} = 6$ SCN slices. All graphs, including time series, show mean ± SEM. Pairwise comparison in (C, H, L) show two-tailed unpaired $t$ test. All other graphs show two-way mixed-effects analysis with matching, with post hoc Šidák's test. ns = non-significant, $*P < 0.05$, $**P < 0.01$. A detailed statistical report for this figure is provided in Appendix Table S1. Source data are available online for this figure.

GABA rhythms was not affected (Fig. 6F; Appendix Table S1). The A37-mediated suppression of extracellular GABA amplitude and rhythm robustness was dose-dependent (Fig. EV5A–C; Appendix Table S1) and fully reversible, whereas higher doses (100 μM) were lethal to SCN slices, as indicated by the immediate loss of all rhythmic signals (PER2::LUC and GABASnFR) (Fig. EV5D).

Co-detected overall circadian oscillations of neuronal calcium and clock gene expression appeared only mildly, and not significantly, affected by the treatment (Fig. 6G–N; Appendix Table S1), indicating that ALDH1A1 inhibition specifically suppresses rhythms of extracellular GABA, similar to MAO-B inhibition. However, unlike MAO-B inhibition, ALDH1A1 inhibition did not affect the period of neuronal calcium or clock gene expression rhythms (Fig. 6J,N; Appendix Table S1), consistent with the milder impact on GABA rhythms.

Thus, the independent inhibition of two different astrocyte-specific enzymes that catalyze GABA synthesis from polyamines impaired circadian oscillations of extracellular GABA in SCN explants, consistent with an astrocytic origin of these rhythms.

## Astrocytic GABA is required for neuronal circuit synchronization in the SCN

To investigate whether astrocytic GABA may provide a synchronizing cue to coordinate neuronal circuit activity and clock gene expression across the SCN, we focused on samples treated with ALDH1A1 inhibitor A37, as GABA rhythms were significantly reduced, but not abolished (Fig. 6B,C; Appendix Table S1), and the overall effects on associated neuronal and clock gene expression generally milder.

Treatment with A37 induced an increase in neuronal calcium, as shown by the significantly increased slope of the Syn-jRCaMP1a signal (Fig. 7A,B; Appendix Table S1), consistent with depolarization of SCN neurons following decreased extracellular GABAergic tone. We then performed single-cell analysis using automated cell detection (see Methods) to analyze the effects of GABA suppression on neurons across the SCN. While ALDH1A1 inhibition did not significantly alter the fraction of rhythmic cells across the SCN or their robustness (Fig. 7C–E), the circadian phase synchronization of SCN neurons was significantly reduced after ALDH1A1 inhibition (Fig. 7F,G; Appendix Table S1), as shown by shallower and wider phase frequency distribution compared to baseline (Fig. 7F) and DMSO-treated slices (Fig. 7G), showing that impairing astrocytic GABA rhythms desynchronizes SCN neurons.

We then focused on the effect of astrocytic GABA inhibition on co-detected PER2::LUC expression. Similar to Syn-jRCaMP1 induction, ALDH1A1 inhibition also caused a significant increase in PER2::LUC levels, consistent with the presence of $Ca^{2+}$/cAMP Responsive Elements (CRE) in the *Per2* promoter (Koyanagi et al, 2011) (Fig. 7H,I; Appendix Table S1). Single-cell analysis showed no significant reduction in the fraction of rhythmic cells across the network or rhythm robustness, as for neuronal calcium. Moreover, phase synchronization of PER2::LUC was significantly reduced following ALDH1A1 inhibition, thus fully recapitulating effects observed on co-detected neuronal calcium (Fig. 7J–N; Appendix Table S1; Movie EV3).

Overall, our findings show that inhibition of astrocytic GABA rhythms by ALDH1A1 inhibition induces increased neuronal calcium and clock gene expression, consistent with changes in an inhibitory GABAergic tone, and desynchronizes circadian rhythms of neuronal calcium and clock gene expression across the SCN.

## Discussion

Our investigation reveals a fundamental dichotomy in the temporal organization of astrocytic and neuronal circadian activity in the SCN, and identifies astrocytic GABA signaling as a critical pathway for synchronization of neuronal activity across the SCN, as summarized in Fig. 8.

Unlike neurons, which display a phase wave of activity spanning several hours across the SCN, we found that astrocytes are remarkably synchronous, with astrocytic activity reporters showing a coherent peak during the nighttime. Intriguingly, extracellular GABA levels also showed highly synchronous circadian oscillations peaking at night, as opposed to presynaptic and intracellular neuronal calcium peaking during the day, suggestive of their astrocytic, rather than neuronal origin. This idea was further supported by our experiments of molecular degradation of proteins responsible for synaptic release of GABA (via TeLC expression) which desynchronized the SCN neuronal circuit, while leaving circadian oscillations of co-detected extracellular GABA intact, coherent with a non-synaptic origin of extracellular GABA rhythms. Single-cell RNA sequencing showed a similar astrocyte–neuronal dichotomy in their biochemical means of GABA production, with neurons expressing the GAD65/67 GABA synthesis pathway from glutamate degradation, and astrocytes expressing the MAO-B/ALDH1A1-dependent GABA synthesis pathway from putrescine. Pharmacological inhibition of astrocytic GABA synthesis disrupted extracellular GABA rhythms and led to an induction of neuronal calcium and clock gene expression, coherent with an inhibitory role of the endogenous astrocyte-derived GABA rhythm. Moreover, this desynchronized neuronal activity and clock gene expression across the SCN, consistent with its role as an internal circuit synchronizer (or zeitgeber), which we refer to as "astrozeit".

Spatiotemporal waves of neuronal calcium and clock gene expression in the SCN encode photoperiodic input controlling seasonal behavioral adaptations (Evans and Gorman, 2016), and are implicated in the differentially phased engagement of downstream brain regions (Evans et al, 2011; Yamaguchi et al, 2003). In contrast, astrocytes display a highly uniform, sustained nighttime activation across the SCN, with no discernible spatial waves, more akin to a pulsatile rhythm (Figs. 1 and 2I), which may suggest a more within-SCN role in timekeeping, rather than circadian engagement of downstream targets.

If the spatiotemporal activity of astrocytes is not just the nighttime mirror image of neurons, are neuronal and astrocytic networks inextricably linked in mutually reinforcing daily negative feedback loops, or could they be disentangled? Our experiments using TeLC show that impairment of GABAergic neurotransmission desynchronized neurons without significantly altering rhythms and synchronization of extracellular GABA (Fig. 3), suggesting that synchronized astrocytic rhythms can proceed regardless of weakened synaptic coupling. We hypothesize that a dual mechanism of synchronization may confer a unique resilience to phase shifts to the SCN, and enable orderly recovery of robust timekeeping even in deeply unphysiological conditions, such as acute jet-lag.

While GABA is the only classical neurotransmitter produced in the SCN (Moore and Speh, 1993; Herzog et al, 2017), experimentation over the last 30 years has led to a somewhat confusing picture of its role within the circuit. While daily addition of GABA can synchronize SCN neurons (Liu and Reppert, 2000), inhibition of GABA_A or GABA_B receptors does not affect SCN synchronization (Aton et al, 2006), and may in fact be associated with neuronal desynchronization (Freeman et al, 2013), in apparent contradiction to the original findings. Moreover, SCN explants from GAD65/67 knockout mice, in which the canonical pathway of GABA synthesis is abolished, show virtually unperturbed circadian oscillations of clock gene expression, leading to the hypothesis that GABA may not be relevant for the internal synchronization of SCN, but rather involved in the timed engagement of downstream targets (Ono et al, 2019). To add further confusion, while SCN neurons, which are >95% GABAergic, show a daytime peak in their firing rate and intracellular calcium levels (Brancaccio et al, 2013; Hastings et al, 2019), this was inconsistent with early measurements of GABA (measured by HPLC in rat SCNs) peaking during the nighttime

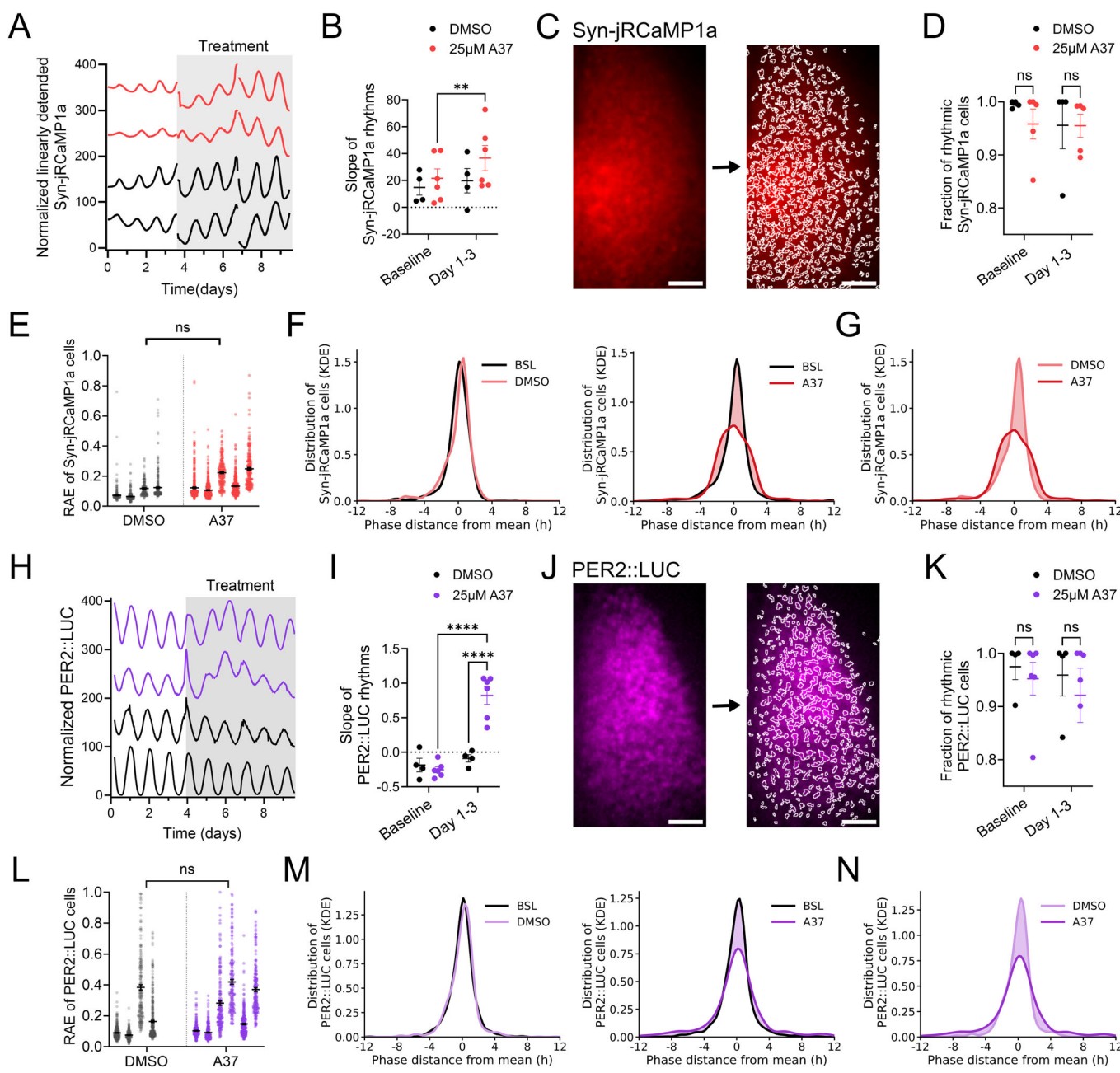

**Figure 7. Inhibition of astrocytic GABA biosynthesis desynchronizes circadian oscillations of neuronal calcium and clock gene expression in the SCN.**

(A) Representative normalized Syn-jRCaMP1a rhythms monitored in SCN slices before and after treatment with A37 (red traces) or vehicle (DMSO; black traces). (B) Slope of Syn-jRCaMP1a signal before and after treatment. Each dot represents one SCN slice. $N_{DMSO} = 4$, $N_{A37} = 6$ SCN slices, $P = 0.0085$ for A37 BSL-Day 1–3. (C) Representative image of one SCN expressing Syn-jRCaMP1a and detected single cells to the right. Scale bar = 100 µm. (D) Fraction of rhythmic Syn-jRCaMP1a cells across SCN slices. $N_{DMSO} = 4$, $N_{A37} = 5$ SCN slices. (E) Relative amplitude error (RAE) of individual cells across SCN slices treated with DMSO or A37 $N_{DMSO} = 4$, $N_{A37} = 5$ SCN slices, ($n = 200–400$ cells/SCN slice), $P = 0.0807$. (F) Phase frequency distribution of phase distance from the mean SCN phase for Syn-jRCaMP1a cells. Distribution before and after treatment with DMSO (left), and before and after treatment with A37 (right). Difference in distribution is indicated by shading. $N_{DMSO} = 4$, $N_{A37} = 5$ SCN slices, ($n = 200–400$ cells/SCN slice). (G) Phase frequency distribution after treatment with DMSO or A37. $N_{DMSO} = 4$, $N_{A37} = 5$ SCN slices, ($n = 200–400$ cells/SCN slice). (H) Representative normalized PER2::LUC rhythms before and after treatment with A37 (purple traces) or vehicle (DMSO; black traces). (I) Slope of PER2::LUC rhythm before and after treatment. Each dot represents one SCN slice. $N_{DMSO} = 4$, $N_{A37} = 6$ SCN slices. (J) Representative image of SCN expressing PER2::LUC with tissue outlined and detected single cells to the right. Scale bar = 100 µm. (K) Fraction of rhythmic PER2::LUC cells across SCN slices. $N_{DMSO} = 4$, $N_{A37} = 6$ SCN slices, ($n = 150–350$ cells /SCN slices). (L) RAE of individual cells across SCN slices treated with A37 or DMSO, $N_{DMSO} = 4$, $N_{A37} = 6$ SCN slices, ($n = 150–350$ cells/SCN slices), $P = 0.550$. (M) Phase frequency distribution of phase distance from the mean SCN phase for PER2::LUC cells. Distribution before and after treatment with DMSO (left), and before and after treatment with A37 (right). The difference in distribution is indicated by shading. $N_{DMSO}$ 4, $N_{A37} = 6$ SCN slices, ($n = 150–350$ cells/SCN slice). (N) Phase frequency distribution after treatment with DMSO or A37. $N_{DMSO} = 4$, $N_{A37} = 6$ SCN slices, ($n = 150–350$ cells/SCN slice). All data shown mean ± SEM. For comparisons across SCN slices shown in (B, D, I, K), a two-way ANOVA with pairing and post hoc Šidák's test is shown. For nested single-cell comparisons in (E, L), a nested two-tailed t test is shown. ns = non-significant, **P < 0.01, ****P < 0.0001. A detailed statistical report for this figure is provided in Appendix Table S1. Source data are available online for this figure.

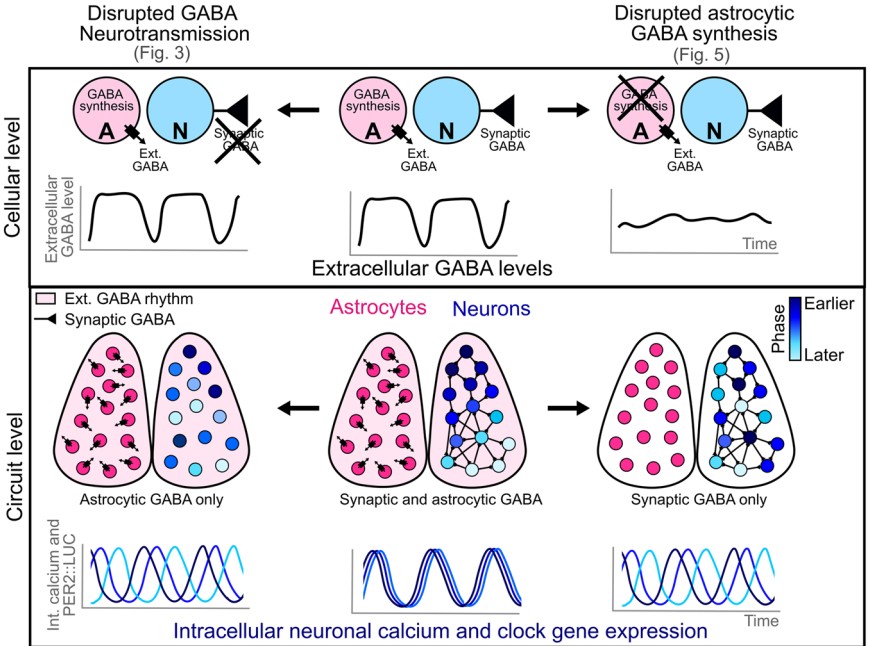

**Figure 8. Proposed model for the role of astrocytic and synaptic GABA in SCN function.**

Top panel summarizes findings at the cellular level. In brief, in the presence of astrocytic GABA synthesis and GABA released by synapses, we observe extracellular GABA rhythms. If synaptic GABA transmission is blocked via genetic TeLC expression (left panel; see Fig. 3), extracellular GABA rhythms persist. If astrocytic GABA synthesis is inhibited (right panel; see Figs. 5 and 6), extracellular GABA rhythms are disrupted. Bottom panel summarizes findings at the SCN circuit level (see Fig. 7), showing astrocytic rhythms are synchronized to a uniform phase in every condition. Neuronal intracellular calcium rhythms and rhythms of clock gene expression are desynchronized across the network when synaptic GABA transmission is disrupted (left panel) and when astrocyte-derived extracellular GABA rhythm is disrupted (right panel), showing that both synaptic GABA transmission and astrocyte-derived extracellular GABA contribute to neuronal synchrony in the SCN.

(Aguilar-Roblero et al, 1993), which we and others (Patton et al, 2023) have now confirmed using genetically encoded fluorescent probes for extracellular GABA.

Our experiments move towards reconciling the role of GABA in the SCN from the perspective of astrocyte–neuronal interplay. By analyzing data from two independent scRNA-Seq datasets (Yao et al, 2023; Wen et al, 2020), we found that SCN neurons and astrocytes express two distinct pathways for GABA biosynthesis: the canonical pathway by GAD65/67-dependent glutamate degradation is mostly restricted to SCN neurons, whereas the noncanonical pathway of MAO-B/ALDH1A1-dependent degradation of putrescine is specifically expressed in astrocytes (Fig. 4D–H).

The robust expression of enzymes of the noncanonical polyamine GABA synthesis pathway in astrocytes observed in the SCN across different independent scRNA-Seq datasets and immunostaining experiments (Figs. 4 and EV3) is consistent with reports showing selective astrocytic expression of the polyamine degradation pathway of GABA synthesis in both mouse and human brains, except for a small subset of serotonergic neurons (Riederer et al, 1987; Levitt et al, 1982; Thorpe et al, 1987). In areas such as the striatum and cerebellum, this pathway has been shown to contribute to tonic inhibition of associated neuronal circuitry under physiological conditions (Woo et al, 2018; Yoon et al, 2014; Lee et al, 2022; Koh et al, 2023).

While we confirmed that synaptic GABAergic neurotransmission is also important for SCN circuit synchronization, as shown by

TeLC-mediated desynchronization of the SCN circuit (Fig. 3H,L), we also found astrocytic GABA to be important for SCN circuit synchronization (Fig. 7). The presence of two potentially redundant pathways for the biosynthesis of GABA may explain why knocking out both GAD65 and GAD67 in mice has only modest effects on circadian timekeeping (Ono et al, 2019), presumably due to astrocytic compensation of GABA biosynthesis. Our experiments suggest that astrocytic GABA synthesis may, however, not be equally compensated, given the desynchronization of neuronal calcium and PER2::LUC oscillations upon ALDH1A1 inhibition. Conversely, astrocytic rhythms of extracellular GABA can persist and remain synchronous when synaptic GABA transmission is impaired by TeLC expression (Fig. 3J–M).

Identifying further mechanisms regulating GABA release and uptake from astrocytes and neuronal sensing will be crucial to build a model of the mutually reinforcing astrocyte–neuronal interplay underpinning circadian timekeeping in the SCN. While Bestrophin1 (Best1) has been implicated in GABA release from astrocytes in other brain regions (Koh et al, 2023), we found it expressed at very low levels in the hypothalamus and SCN (Fig. EV3H,I). As astrocyte subpopulations have been shown to release GABA (Wang et al, 2013) and glutamate (de Ceglia et al, 2023) through vesicular release in other brain regions, we also examined expression of the *Vglut1* and *Vgat*, responsible for vesicular glutamate and GABA release, respectively. We found very low expression levels of both *Vglut1* (0.1%) and *Vgat* (2.9%) in hypothalamic astrocytes, with only 0.7% of astrocytes co-expressing *Maob*, *Aldh1a1*, and *Vgat* (Fig. EV3G). While this does not

rule out contributions to GABA levels by astrocytic VGAT-mediated release in the SCN, especially at shorter timescales, they appear unlikely mediators of the circadian-scale tissue-wide oscillations of extracellular GABA reported here. Nevertheless, future tissue-wide investigations of astrocyte–neuronal signaling at multiple timescales within the SCN, also inclusive of different anatomical planes (e.g., sagittal) may help disentangle the inherently complex features mediating astrocyte–neuronal interplay within the SCN and beyond.

In contrast to *Best1* and *Vgat*, we found *Slc6a11* and *Slc6a1*, which encode the GABA transporters GAT3 and GAT1, respectively, to be highly expressed in SCN astrocytes. GAT3 has been shown to mediate extracellular GABA uptake in astrocytes in other brain regions (Moldavan et al, 2015, 2017). However, reverse GABA transport through GATs can also occur in astrocytes via GAT3 (Héja et al, 2012; Wójtowicz et al, 2013), and in neurons and Bergmann glia via GAT1 (Wu et al, 2007; Barakat and Bordey, 2002; Allen et al, 2004).

Notably, pharmacological inhibition of GAT3 in SCN slices leads to an accumulation of extracellular GABA, showing that GAT3 mediates GABA uptake in the SCN (Patton et al, 2023). If astrocytes can regulate extracellular GABA rhythms both via synthesis and uptake, how are GABA levels regulated by astrocytes across the circadian day? Using the scRNA-Seq dataset from (Wen et al, 2020), we found that *Aldh1a1* expression peaks at CT13 (Fig. 4J), consistent with an independent dataset from (Pembroke et al, 2015) (wgpembroke.com/hiny/SCNseq). In contrast, *Slc6a11*, which encodes GAT3, peaked at CT6 in SCN astrocytes, consistent with (Patton et al, 2023) (Fig. EV3J). This suggests that astrocytes may generate extracellular GABA rhythms by switching from increased daytime GAT3-mediated GABA uptake, removing synaptically released GABA, to nighttime astrocytic GABA synthesis, replenishing extracellular GABA levels and inhibiting SCN neurons. This daily astrocyte switch will generate the circadian oscillations of extracellular GABA observed here and by (Patton et al, 2023). Disruption of either GABA production or uptake leads to a dysregulation of extracellular GABA rhythms due to low (Fig. EV4B), or excess extracellular GABA (Patton et al, 2023), both ultimately disrupting circadian cycling of extracellular GABA.

It is not known how GABA production and uptake are regulated by the circadian clock or whether there are other shared upstream pathways. Rev-erbα has been shown to positively regulate *Slc6a1* and *Slc6a11* expression by repressing E4bp4, a transcriptional repressor of multiple transporters, in the hippocampus and cortex (Zhang et al, 2021). However, a molecular link between Rev-erbα and *Aldh1a1* regulation has not yet been described.

Our data suggest that circadian astrocytic GABA tone plays a significant role in synchronizing neuronal circadian rhythms within the SCN. This suggests that disturbances of SCN GABA rhythms may also indirectly weaken coordination of peripheral clocks by reducing coherent SCN output to the periphery. Whether or not GABA may also play a more direct "astrozeit" synchronization role within peripheral brain oscillators, remains to be tested.

Of note, augmented GABA levels by increased MAO-B expression in astrocytes have previously been associated with reactive gliosis in several mouse models of neurodegeneration, including Alzheimer's Disease (AD). In these models MAO-B inhibition also reduces GABA, increases synaptic activity, and rescues learning and memory impairments (Jo et al, 2014; Park

et al, 2019; Wu et al, 2014; Mathys et al, 2024). Pathological modifications of the SCN are observed in AD and involve early reactive astrogliosis before significant neurodegeneration occurs (Brancaccio et al, 2021; Hastings et al, 2023). Our experiments suggest a mechanistic link between MAO-B-dependent GABA synthesis and neuronal synchrony in the SCN, raising the question whether reactive gliosis in the SCN and early weakening of circadian cycles in AD could be linked via the dysregulation of physiologically occurring rhythms of GABA production (Brancaccio et al, 2021; Hastings et al, 2023).

# Methods

**Reagents and tools table**

| Reagent/resource | Reference or source | Identifier or catalog number |
|---|---|---|
| **Experimental models** | | |
| PER2::LUC (B6.129S6-Per2tm1Jt/J) | Jax Laboratories | RRID: IMSR_JAX:006852 |
| **Recombinant DNA** | | |
| pAAV.Syn.NES.jRCaMP1a.WPRE.SV40 | Addgene | 100848-AAV1 |
| pZac2.1 gfaABC1D-lck-GCaMP6f | Addgene | 52924-AAV5 |
| pAAV.GfaABC1D.GluSnFr.SV40 | Addgene | 100889-AAV5 |
| pAAV.hSynap.iGABASnFR | Addgene | 112159-AAV1 |
| pAAV-Ef1a-DIO-Synaptophysin-GCaMP6s | Addgene | 105715-AAV5 |
| hSyn-mCherry | UNC Vector Core | AV8209 |
| hSyn-mCherry::Cre | UNC Vector Core | AV5052C |
| Syn1-TeLC-P2A-mCherry-WPRE | Vectorbuilder | VB220825-1449bff |
| Gfap-mCherry::Cre | UNC Vector Core | AV5056C |
| **Antibodies** | | |
| Anti-VAMP2 antibody, rabbit polyclonal | Proteintech | 10135-1-AP; RRID: AB_2256918 |
| Anti-GFP antibody, goat polyclonal | Abcam | ab6673; RRID: AB_305643 |
| Anti-GFAP antibody, chicken polyclonal | Abcam | ab4674; RRID: AB_304558 |
| Anti-GABA antibody, guinea pig polyclonal | Sigma-Aldrich | AB175; RRID: AB_91011 |
| Anti-ALDH1A1 antibody, rabbit polyclonal | Proteintech | 15910-1-AP; RRID: AB_2305276 |
| Anti-MAO-B antibody, rabbit polyclonal | Invitrogen | PA5-95036; RRID: AB_2806842 |
| Alexa Fluor 647 Donkey Anti-Rabbit antibody | Abcam | ab150075; RRID: AB_2752244 |
| Alexa Fluor 488 Donkey Anti-Goat antibody | Invitrogen | A11055; RRID: AB_2534102 |
| Alexa Fluor 647 Donkey Anti-Chicken antibody | Jackson Immunoresearch | 703-605-155; RRID: AB_2340379 |

| Reagent/resource | Reference or source | Identifier or catalog number |
|---|---|---|
| Alexa Fluor 488 Goat Anti-Guinea Pig antibody | Invitrogen | A11073; RRID: AB_2534117 |
| Alexa Fluor 488 Donkey Anti-Rabbit antibody | Invitrogen | A21206, RRID: AB_2535792 |
| **Chemicals, enzymes, and other reagents** | | |
| Selegiline ((R)-( − )-N, α-Dimethyl-N-(2-propynyl) phenethylamine hydrochloride) | Sigma-Aldrich | Cat. No. M003; CAS No. 14611-52-0 |
| A37 (Ethyl 2-[[3,4-dihydro-4-oxo-3-[3-(1-pyrrolidinyl) propyl][1]benzothieno[3,2-d] pyrimidin-2-yl]thio]acetate) | Tocris Bioscience | Cat. No. 5802; CAS No. 896795-60-1 |
| **Software** | | |
| BioDare2 | Zielinski et al, 2014 | https://biodare2.ed.ac.uk/ |
| Fiji | Schindelin et al, 2012 | https://fiji.sc/ |
| Matlab R2022b | MathWorks | |
| Prism 10 | GraphPad | http://www.graphpad.com/scientific-software/prism/ |
| Python3 | N/A | https://www.python.org/ |

## Methods and protocols

### Mouse lines

Experiments were conducted according to the United Kingdom Animals (Scientific Procedures) Act 1986 with Local Ethical Review by the Imperial College London Animal Welfare and Ethical Review Body Standing Committee (AWERB). For ex vivo SCN slice experiments, female and male animals were used and were housed on a 12:12 light-dark schedule. Food and water were provided ad libitum. Pups were kept in the parent cage and sacrificed using cervical dislocation at P11-P15.

PER2::LUC knock-in mice (Yoo et al, 2004) (allele *B6.129S6-Per2tm1Jt/J*) were a gift from Dr Michael Hastings (MRC Laboratory of Molecular Biology, Cambridge, UK).

All experiments were performed on at least three animals, with the number of experimental replicates (N) shown in text and figure legends. Samples were randomly assigned to experimental conditions.

Data from all the experiments were included in the analysis, with the exclusion of SCN slices which died for unrelated technical reasons or, for experiments investigating synchronization of rhythms across the SCN, low-quality samples with highly desynchronized rhythms at baseline, before any treatment was performed.

### SCN organotypic slice preparation

SCN organotypic slices were prepared as previously described (Brancaccio et al, 2017). Briefly, brain was isolated after cervical dislocation and placed in ice-cold dissection medium (GBSS [G9779, Sigma] with 5 mg/mL glucose [158968, Sigma], 100 nM MK801 [M107, Sigma], 3 mM MgCl2 [AM9530G, Invitrogen],

50 µM AP-5 [0106, Tocris] filtered with a 0.22-µm pore size Steriflip). Medial-ventral part of the brain containing the SCN was isolated and 300 µM coronal slices were cut using a tissue chopper (McIlwain). Slices containing the SCN were identified, and surrounding tissue was removed under a light microscope. SCN tissue was placed on a membrane insert (PICM0RG50 or PICM01250, Millipore) and kept in the initial plating medium for 2–4 h (air medium working solution, supplemented with 100 nM MK801, 3 mM MgCl$_2$ and 50 µM AP-5; air medium working solution: 500 mL ddH$_2$O with 4.15 g DMEM [D5030, Sigma], 0.175 g NaHCO$_3$, 2.25 g glucose [158968, Sigma], 5 mL penicillin/streptomycin [P4333, Sigma], 5 mL HEPES 1 M [H0887, Sigma], 5% horse serum [10270106, Gibco], 1% B27 [17504044, Gibco] and 0.5% Glutamax [35050-038, Invitrogen]). Membrane inserts were then transferred into air medium working solution and kept sealed at 37 °C. Slices were kept undisturbed for at least 6 days, and the medium was refreshed weekly.

### AAV transduction of SCN slices

Slices were transduced via direct addition of 1–1.5 µl of AAVs onto the slice and left undisturbed for at least 4 days.

AAVs encoding Syn-NES-jRCaMP1a-WPRE-SV40 (Douglas Kim; 100848), GfaABC1D-lck-GCaMP6f (Baljit Khakh; 52924), Syn-GABASnFR (Loren Looger; 112159), GfaABC1D-iGluSnFR-SV40 (Baljit Khakh; 100889), and Ef1α-DIO-Synaptophysin::GCaMP6s (Rylan Larsen; 105715) were purchased from Addgene.

hSyn-mCherry (AV8209), hSyn-mCherry::Cre (AV5052C) and Gfap-mCherry::Cre (AV5056C) were purchased from UNC vector core.

SCN slices transduced with Ef1α-DIO-Synaptophysin::GCaMP6s were also transduced with hSyn-mCherry::Cre after 6 days to induce expression in neurons.

Syn1-TeLC-P2A-mCherry-WPRE was designed and purchased from Vectorbuilder with tetanus toxin light chain (TeLC) sequence from Addgene plasmid #159102 (Sandeep Datta). To quantify the functional effects of this AAV on SCN slices, we recorded 5 days pre-transduction (baseline) and 12 days post-transduction, using the first 5 days prior to transduction as a baseline, immediately after transduction (days 1–6) to assess effects during partial expression, and more than 7 days post-transduction (days 7–12) to assess effects of full TeLC expression.

### Drug treatments

Selegiline (R-(-)-Deprenyl hydrochloride; Cat. No. M003, Sigma-Aldrich) was reconstituted in DMSO at 30 mg/ml for addition to SCN slice medium at a concentration of 200 µM. A37 (Cat. No. 5802, Tocris) was reconstituted in DMSO at 100 mM for addition to SCN slice medium at a concentration of 10, 25, 50, or 100 µM as stated. The corresponding volume of DMSO was used for controls.

To wash out drugs, slices were washed with medium two times and transferred to fresh medium.

### Immunofluorescence on SCN slices

SCN slices were fixed with 4% PFA for 2 h at room temperature. For immunofluorescence, slices were incubated in day 1 buffer (1× PBS, 1% bovine albumin, 0.3% triton-X) with 10% donkey serum (Abcam) or goat serum (Invitrogen) at room temperature, then incubated with primary antibodies against VAMP2 1:500 (Proteintech #10135-1-AP), GFP 1:200 (Abcam #ab6673), GFAP 1:1000 (Abcam #ab4674), GABA 1:200 (Sigma-Aldrich #AB175), ALDH1A1 1:500 (Proteintech #15910-1-AP) or MAO-

B 1:200 (Invitrogen # PA5-95036) overnight at 4 °C. Slices were washed twice in day 2 buffer (1:3 dilution of day 1 buffer in 1× PBS), and incubated with 1:1000 donkey anti-rabbit Alexa Fluor 647 (Abcam), donkey anti-goat Alexa Fluor 488 (Invitrogen), goat anti-guinea pig Alexa Fluor 488 (Invitrogen), donkey anti-rabbit Alexa Fluor 488 (Invitrogen) or donkey anti-chicken Alexa Fluor 647 (Jackson Immunoresearch) for 1 h at room temperature. After two further 5 min washing steps in day 2 buffer, and 3 × 20 min washes in 1× PBS, slices were mounted on a glass slide with mounting medium with NucBlue (Prolong Glass). Slices were imaged at ×20 magnification on a widefield microscope (Zeiss Axio Observer) to assess VAMP2 levels or at ×20 and ×63 magnification on a confocal microscope (Zeiss LSM-780 inverted) as indicated to assess spatial distribution of AAV-encoded reporters, GABA, MAO-B, ALDH1A1 or GFAP signal.

To assess VAMP2 levels in Syn-TeLC-mCherry-expressing SCN slices, a subset of slices used for the longitudinal experiment were fixed at the end of the experiment on day 12 post-transduction.

To assess GABA levels after Selegiline treatment, SCN slices were treated with 200 μM Selegiline or DMSO and fixed 4 days post-treatment.

To assess ALDH1A1 and MAO-B protein expression in SCN slices, SCN slices were transduced with Gfap-mCherry::Cre AAV to mark astrocytic nuclei, fixed 10 days post-transduction and stained with antibodies against GFAP and ALDH1A1 or MAO-B.

### Multi-channel long-term live imaging of SCN slices

Multi-channel bioluminescence or fluorescence imaging was performed using an LV200 system (Olympus) or Incucyte S3 (Sartorius). For LV200 recordings, images were acquired with a ×40 long-distance range objective, and exposure varied between 5 and 40 ms for fluorescent reporters, while bioluminescence signal was acquired over 4 min. For Incucyte recordings, images were acquired at ×4 magnification, and exposure varied between 300 and 400 ms for fluorescent reporters. Images were acquired every 30 min. For bioluminescent recordings, 100 μM Luciferin (#12505, AAT Bioquest) was added to the medium. SCN slices were imaged in glass-like bottom tissue culture plates (P06-1.5H-N or P24-1.5H-N, CellVis) (LV200) or plastic tissue culture plates (734-2777 or 734-2779, VWR) (Incucyte), and sealed with a plastic film (Z369667, Excel Scientific) to prevent medium evaporation. Temperature was kept at 37 °C, with top plate temperature at 38 °C to prevent condensation, using a TOKAII HIIT (LV200).

### Image processing

Raw imaging files from the LV200 or Incucyte were processed in ImageJ/FIJI (Schindelin et al, 2012). Image stacks were registered using the descriptor-based series registration 2 d/3 d + t plugin (https://github.com/fiji/Descriptor_based_registration).

For extraction of mean time series, the SCN was manually delineated and signal intensity measured. In the case of Syn-TeLC-mCherry or Syn-mCherry transduced slices, a moving average with a 24.5 h period was applied using the moving average subtract tool by Jay Unruh at the Stowers Institute for Medical Research in Kansas City, MO in FIJI.

**Detection of single cells in SCN slice**: For single-cell image analysis, PER2::LUC image stacks were first processed to remove bioluminescence artifacts in ImageJ as follows. Noise was removed using 'Despeckle', a 3 × 3 pixel median filter, cosmic rays were removed using 'Remove Outliers', and a blurred copy of the image was subtracted using 'Unsharp Mask' to increase contrast of structures, enabling better edge detection of cell outlines in subsequent thresholding steps.

For all fluorescent and bioluminescent reporters, image stacks were averaged and single cells detected using a Laplacian operator with the SARFIA package (Dorostkar et al, 2010) in Matlab.

### Data analysis

**Immunofluorescence image analysis**: VAMP2 expression levels were quantified using the raw mean intensity within each manually delineated SCN in ImageJ.

Spatial expression of AAV-encoded Syn-jRCaMP1a and GfaABC1D-lck-GCaMP6f across SCN slices was determined in fixed slices stained with antibodies against GFP to enhance the GCaMP6f signal and GFAP to investigate astrocyte distribution across the SCN. Fluorescence signal across different areas of the SCN was quantified by measuring the raw mean intensity within 148 × 148 pixel rectangles 10% of the height or width away from the dorsal, ventral, medial and lateral edge of the SCN.

To assess the expression of ALDH1A1 and MAO-B, a rolling ball background subtraction was applied to confocal images in ImageJ. To identify Gfap-mCherry::Cre-positive cells, images across a 4 μM Z-stack with 1 μM interval were averaged, a Gaussian filter ("Gaussian Blur") was applied to the mCherry signal, a custom threshold was applied to create a mask, and particles identified using the Analyze Particles macro in ImageJ. Mean fluorescence intensities of GFAP and MAO-B or ALDH1A1 antibodies were measured in each mCherry$^+$ particle to assess co-localization. To assess signal overlap with GFAP, the JaCoP plugin (Bolte and Cordelières, 2006) was used to calculate the Mander's coefficient of overlap between GFAP and MAO-B or ALDH1A1 on a single Z-stack frame for each sample.

To assess the number of Gfap-mCherry::Cre-expressing cells in SCN slices treated with seleginine, confocal images were processed as above to detect Gfap-mCherry::Cre particles, and scaled by the area of the SCN in which they were detected.

**Time series and circadian analysis**: Mean SCN time series were detrended using either a linear regression or third-order polynomial and plotted in GraphPad Prism v10. Circadian parameters were analyzed for detrended time series using BioDare2 (https://biodare2.ed.ac.uk) (Zielinski et al, 2014). Circadian period was estimated using a Fast Fourier transform and non-linear least squares regression (FFT-NLLS). The averaged peaks of the time series were used to estimate circadian phase. For rhythms displaying a waveform with a flat peak, including rhythms of GfaABC1D-lck-GCaMP6f, GfaABC1D-iGluSnFR and Syn-GABASnFR, the midpoint of the peak was interpolated by the sum of the phase of the inverted rhythm (the trough) and half of the period.

Where indicated, the rhythmicity of a time series was analyzed on raw traces using the BD2 eJTK rhythmicity test in BioDare2 (Hutchison et al, 2015), a modified version of empirical JTK_CYCLE with asymmetry search. Presented τ values range within [−1,1], with 1 and −1 indicating perfect correlation or anticorrelation respectively with tested time series. We deemed a time series as rhythmic if the empirical $P$ value calculated against random noise data was ≤0.0001, as this was the most stringent $P$ value threshold available in BioDare2.

To visualize mean time series with standard error of the mean of a reporter across multiple SCN slices, time series of individual slices

were aligned based on their first peak using a custom script in Python (https://github.com/natalieness/CircData). Rayleigh plots of circadian phase were generated using custom Python script (https://github.com/natalieness/CircData).

**Spatiotemporal circadian activity analysis**: For analysis of spatiotemporal dynamics of reporter signal, image stacks were divided into rectangular regions of interest of 3 × 3 pixels covering the SCN. Time series were extracted from each region of interest and linearly detrended using the linear stack detrend tool by Jay Unruh at the Stowers Institute for Medical Research in Kansas City, MO in FIJI. Extracted time series were then normalized in Python, and using a modified approach from (Foley et al, 2011), a k-means clustering algorithm was implemented using the classical EM-style Lloyd algorithm in *scikit-learn 1.2.2* (Pedregosa et al, 2011) with k = 5. Circadian parameters of the mean time series of each cluster were estimated using BioDare2, using the absolute circadian phase, which is not scaled by the estimated period, to avoid effects of variation in period estimates across clusters. Vectors of spatiotemporal progression of the clusters were calculated by linear regression of the center of mass of the clusters, using the center of mass function from the *scipy* multidimensional image processing package *scipy.ndimage* (Virtanen et al, 2020), followed by fitting a linear regression using *scikit-learn* (Pedregosa et al, 2011). Circular variance of cluster phases was calculated as follows:

$$V = 1 - \frac{\sqrt{(\sum \sin(x))^2 + (\sum \cos(x))^2}}{n}$$

Where $x$ is the phases or vector angles in radians and n is the number of phases or vector angles.

Maps of clusters, cluster time series and circular histograms of vector directionality were generated using the *matplotlib* package (Hunter, 2007). The custom ImageJ macro and Python script for this analysis is available at https://github.com/natalieness/spatiotemporal-slice-map.

To visualize the correlations between circular variance of cluster phases, circular variance of directionality of vector angle, and phase, the mean values of these parameters for each reporter was plotted in 3D using the *matplotlib* package (Hunter, 2007) in Python.

To investigate differences in spatiotemporal dynamics across the anterior-posterior axis, SCN slices used for analysis of spatiotemporal patterns of astrocytic calcium, extracellular glutamate or GABA were categorized into anterior, medial or posterior by a blinded colleague based on visual inspection of SCN shape in a single frame image of the SCN slice with brightfield, Syn-jRCaMP1a and PER2::LUC expression recorded on the LV200 (Olympus), see Fig. EV1D for SCN shape criteria and representative images of each region. Phase synchrony and phase relationship to neuronal calcium rhythms were then compared across regions.

### Analysis of scRNA-Seq data

scRNA-Seq data were analyzed from publicly available single-cell transcriptome datasets acquired by the Allen Brain Institute as part of the Allen Brain Cell Atlas (ABC) (Yao et al, 2023). Pre-processed and normalized transcript counts were downloaded from the public dataset

at https://allen-brain-cell-atlas.s3.us-west-2.amazonaws.com/index.html and analyzed in Python as described here: https://alleninstitute.github.io/abc_atlas_access/intro.html. Data were visualized using *matplotlib* in Python or GraphPad Prism v10.

SCN-specific scRNA-Seq data were analyzed using a publicly available Drop-seq single-cell RNA sequencing dataset from (Wen et al, 2020) (GEO accession number: GSE118404) using the *Seurat* pipeline in R (Hao et al, 2024), following the analysis procedure described by (Wen et al, 2020) as outlined below. This dataset contains cells isolated from adult mouse SCNs across 12 circadian timepoints.

To exclude low-quality cells, the cells were filtered by gene count, removing cells with less than 200 or more than 800 genes, and by mitochondrial gene detection ratio, with the threshold set to 10%. A total of 45,517 cells were processed for further clustering and analysis.

We then performed PCA and used an elbow plot to determine the number of principle components to be used for clustering, which was determined to be 20. A graph-based smart local moving algorithm with a resolution factor of 0.4 was applied for clustering, which resulted in 16 cell clusters. We then used Uniform Manifold Approximation and Projection (UMAP) as a dimensionality reduction method to visualize the clusters.

To annotate the cell types, we used manual annotation based on the genes used to label cells in the original paper to ensure it closely matched the original findings.

To group cells based on cell type and/or time of day, we used an integrated function from the *Seurat* package called *AggregateExpression*. We set the scale factor to 1e6 to extract counts per million (CPM), which were then transformed to $\log_2(CPM + 1)$.

Data were visualized using GraphPad Prism v10. To analyze rhythmicity and determine circadian phase of gene expression time series, we used the BD2 eJTK rhythmicity test with eJTK classic analysis preset and Benjamini–Hochberg correction in BioDare2 (Moore et al, 2014).

### Statistical analysis

All statistical analyses were conducted in GraphPad Prism v10, unless otherwise specified below. Data distributions were assumed to be normal, without formal testing for normality due to the low number of samples used in most experiments ($N < 30$).

All statistical comparisons are as specified in the figure legend. Statistical analyses comparing two groups were conducted using parametric paired or unpaired two-tailed *t* tests, as specified.

Comparisons involving more groups were conducted using one-way ANOVA, or two-way ANOVA with matching to account for either time, where measurements were taken across different time intervals, or sample, where multiple fluorescent or bioluminescent reporters were co-detected in the same SCN slice. Mixed-effects analysis with matching was used in the same manner, where measurements were not matched in sample size across groups. For post hoc comparisons, Tukey's post hoc test was used when comparing all means, for example comparing every reporter to every other reporter. Šidák's post hoc test was used when only comparing means between groups that differed by no more than one variable, for example, treatment group or time.

For single-cell data, nested equivalents of the above were used to account for pairing of single cells from the same SCN slices.

Rayleigh test of uniformity was calculated using the *pycircstat* package (github.com/circstat/pycircstat) in Python.

Probability density function of cluster phase variances, kernel density estimation of single-cell phase distributions and associated statistics, including descriptive statistics and 2-sample Kolmogorov–Smirnov test, were performed using the *scipy.stats* module (Virtanen et al, 2020), and plotted using *matplotlib* (Hunter, 2007), in Python.

### Graphics

Schematics in Figs. 3A, 5A, and 6A were created using Biorender.com.

## Data availability

All data will be made available upon reasonable request. The code produced in this study is available in the following databases: Scripts used for unsupervised clustering analysis: Github (https://github.com/natalieness/spatiotemporal-slice-map); Scripts used for circular data and plotting: Github (https://github.com/natalieness/CircData).

The source data of this paper are collected in the following database record: biostudies:S-SCDT-10_1038-S44318-024-00324-w.

## Peer review information

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

## Acknowledgements

This work is supported by the UK Dementia Research Institute [award number UKDRI-5007 to MB] through UK DRI Ltd, principally funded by the UK Medical Research Council, Imperial College London President's PhD Scholarship awarded to NN and Michael Uren Foundation to MB. We would like to thank Imperial College London Centre Biomedical Services (CBS) for their support with animal husbandry and the Facility for Imaging by Light Microscopy (FILM) for training and support with imaging and imaging processing. We thank all past and present members of the Brancaccio laboratory, and especially Yongyi Dai, Anne Wolfes, Renaud Bussiere, Aina Badia Soteras, and Marco Ferrari, for discussions about the project. We also thank Maria Tsalenchuk for the helpful discussion about single-cell RNA-Seq data.

## Author contributions

**Natalie Ness**: Conceptualization; Data curation; Software; Formal analysis; Funding acquisition; Validation; Investigation; Visualization; Methodology; Writing—original draft; Writing—review and editing. **Sandra Diaz-Clavero**: Investigation; Methodology; Writing—review and editing. **Marieke M B Hoekstra**: Methodology; Writing—review and editing. **Marco Brancaccio**: Conceptualization; Resources; Supervision; Funding acquisition; Writing—original draft; Project administration; Writing—review and editing.

Source data underlying figure panels in this paper may have individual authorship assigned. Where available, figure panel/source data authorship is listed in the following database record: biostudies:S-SCDT-10_1038-S44318-024-00324-w.

## Disclosure and competing interests statement

The authors declare no competing interests.

# Expanded View Figures

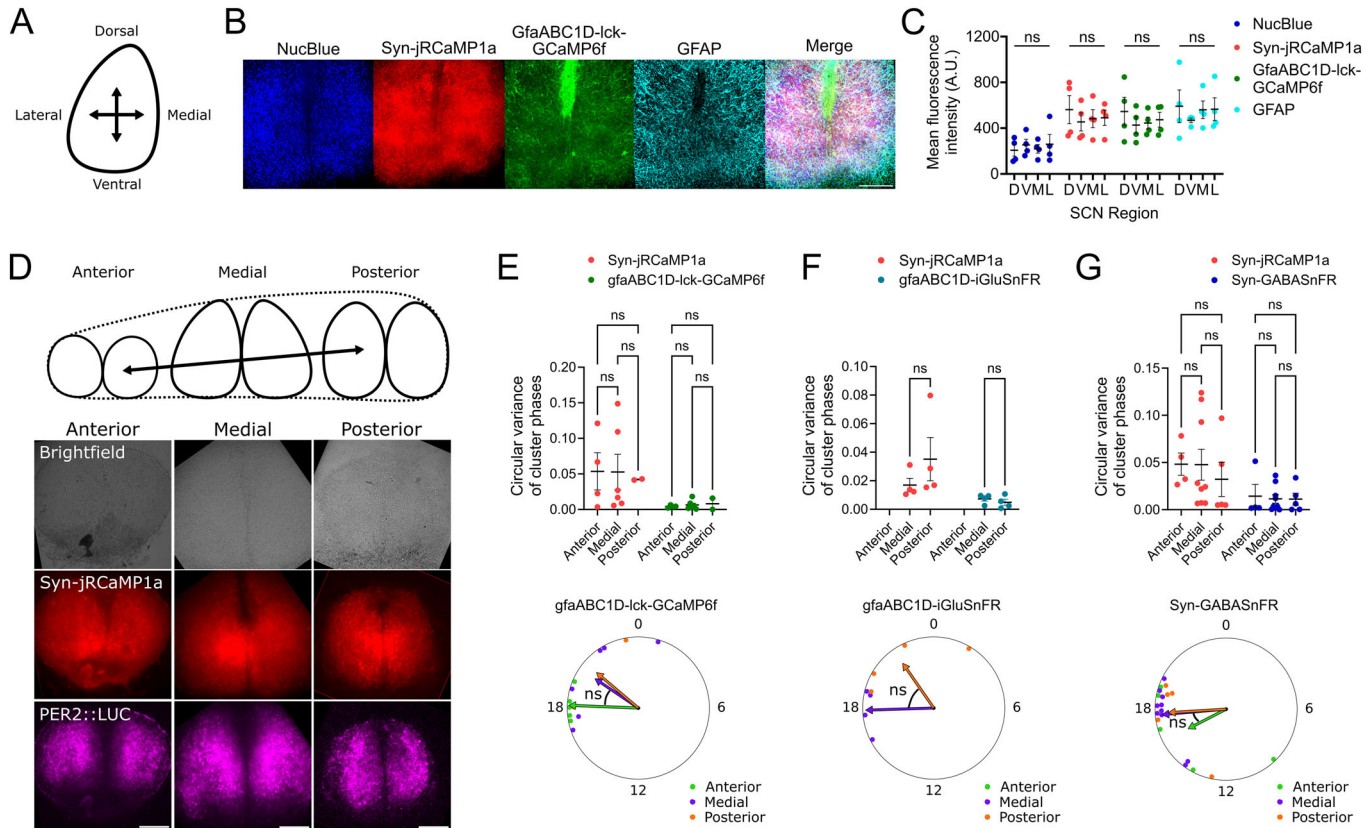

**Figure EV1. Astrocytic calcium reporter is evenly expressed across the SCN, and spatiotemporal activity of astrocyte reporters is homogenous along the SCN anterior-posterior axis.**

(A) Schematic showing spatial regions within a coronal SCN slice with dorsal, medial, lateral and ventral edges. (B) Representative confocal image of an SCN slice expressing neuronal (Syn-jRCaMP1a) and astrocytic (GfaABC1D-lck-GCaMP6f) calcium reporters, counterstained with NucBlue and GFAP antibody. Scale bar = 200 μm. (C) Quantification of mean fluorescence intensity of each reporter in the dorsal (D), ventral (V), medial (M) and lateral (L) SCN regions (see Methods), showing no detectable spatial differences in the expression of the neuronal or astrocytic calcium indicators, or GFAP staining intensity within the different SCN regions. N = 4 SCN slices, two-way ANOVA with matching and post hoc Šídák's test. (D) Schematic showing the shape of SCN nuclei along the anterior-posterior axis, with images of one representative SCN expressing Syn-jRCaMP1a and PER2::LUC for each region. Scale bar = 200 μm. (E) Top panel shows circular variance of phases across clusters in SCN slices expressing Syn-jRCaMP1a and GfaABC1D-lck-GCaMP6f divided by region across A-P axis as shown in (D). N = 2–6 SCN slices per region. Bottom panel shows Rayleigh plot of circadian phases of GfaABC1D-lck-GCaMP6f relative to co-detected Syn-jRCaMP1a (peaking at CT6) within each region across the A-P axis. Each dot represents one SCN slice, vector direction indicates mean phase, and vector length inversely indicates circular dispersion. (F) Top panel shows circular variance of cluster phases of Syn-jRCaMP1a and GfaABC1D-iGluSnFR by region. N = 4 SCN slices per region. Bottom panel shows Rayleigh plot of circadian phases of GfaABC1D-iGluSnFR relative to co-detected Syn-jRCaMP1a by region. (G) Top panel shows circular variance of cluster phases of Syn-jRCaMP1a and Syn-GABASnFR by region. N = 4–10 SCN slices per region. Bottom panel shows Rayleigh plot of circadian phases of Syn-GABASnFR relative to co-detected Syn-jRCaMP1a by region. All linear graphs show mean ± SEM, graphs in top panel of (E–G) show two-way mixed-effects analysis with matching and post hoc Šídák's test. Circular Rayleigh plots in bottom panel of (E–G) show Watson-Williams test of homogeneity of means. ns = non-significant. A detailed statistical report for this figure is provided in Appendix Table S1. Source data are available online for this figure.

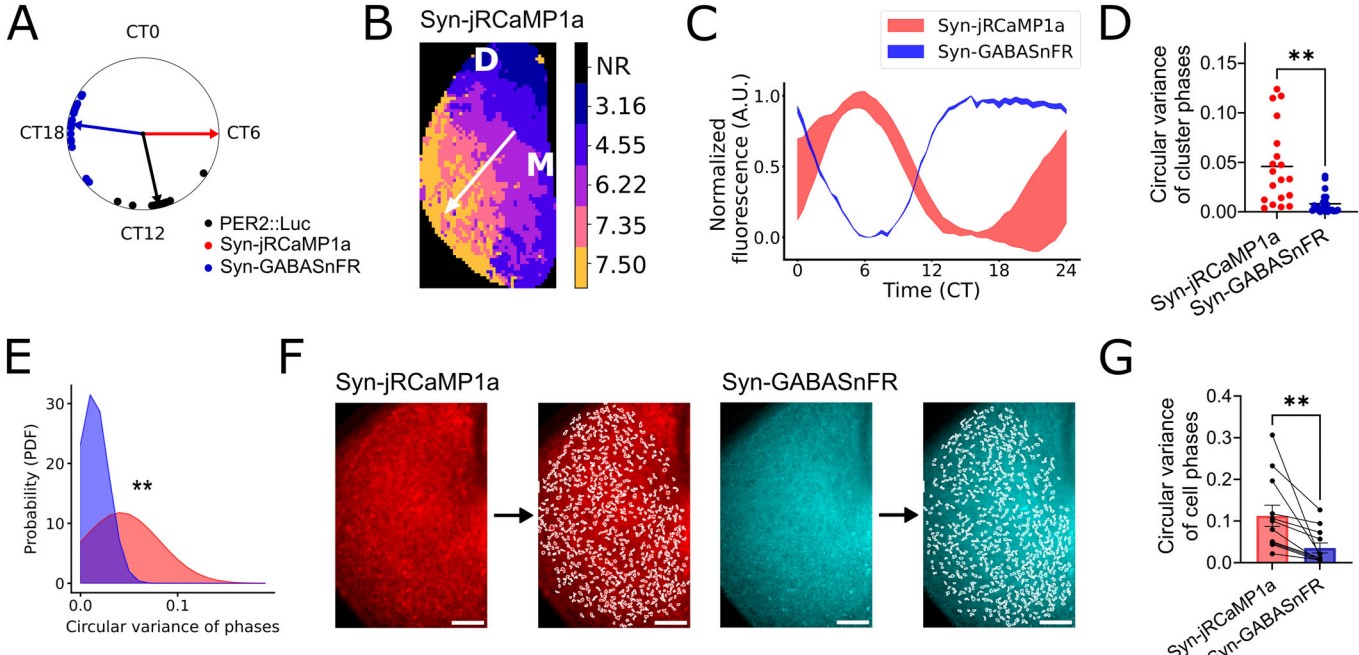

**Figure EV2. Characterization of phase relationship between Syn-jRCaMP1a and Syn-GABASnFR and their synchronization across the SCN network and single cells.**

(A) Rayleigh plot showing circadian phase of Syn-GABASnFR (blue) and PER2::LUC (black) relative to co-detected Syn-jRCaMP1a. (B) Representative circadian phase cluster map of Syn-jRCaMP1a, showing same SCN as co-detected PER2::LUC and Syn-GABASnFR phase map in Fig. 2C. One SCN nucleus is shown (dorsal (D) and medial (M) area indicated). Color bars indicate cluster phases, NR = non-rhythmic. White arrow indicates the direction of the phase progression. (C) Representative standard deviation of cluster time series of co-detected Syn-jRCaMP1a and Syn-GABASnFR. (D) Inter-cluster phase dispersal (measured by circular variance) of co-detected Syn-jRCaMP1a and Syn-GABASnFR (N = 19 SCN slices). Paired two-tailed $t$ test, $P = 0.0013$. (E) PDF of cluster phase variance for each co-detected reporter, with Kolmogorov–Smirnov test, $P = 0.00397$. (F) Representative images of averaged Syn-jRCaMP1a (left) and Syn-GABASnFR (right) signal in an SCN slice with detected single cells indicated in white to the right. Scale bar = 100 μm. (G) Circular variance of circadian phases of Syn-jRCaMP1a or Syn-GABASnFR across individual cells across the SCN, each data point represents 1 SCN slice, with 200–300 cells measured per slice. Two-tailed paired $t$ test, $P = 0.0076$. All graphs are mean ± SEM, $^{**}P < 0.01$. A detailed statistical report for this figure is provided in Appendix Table S1. Source data are available online for this figure.

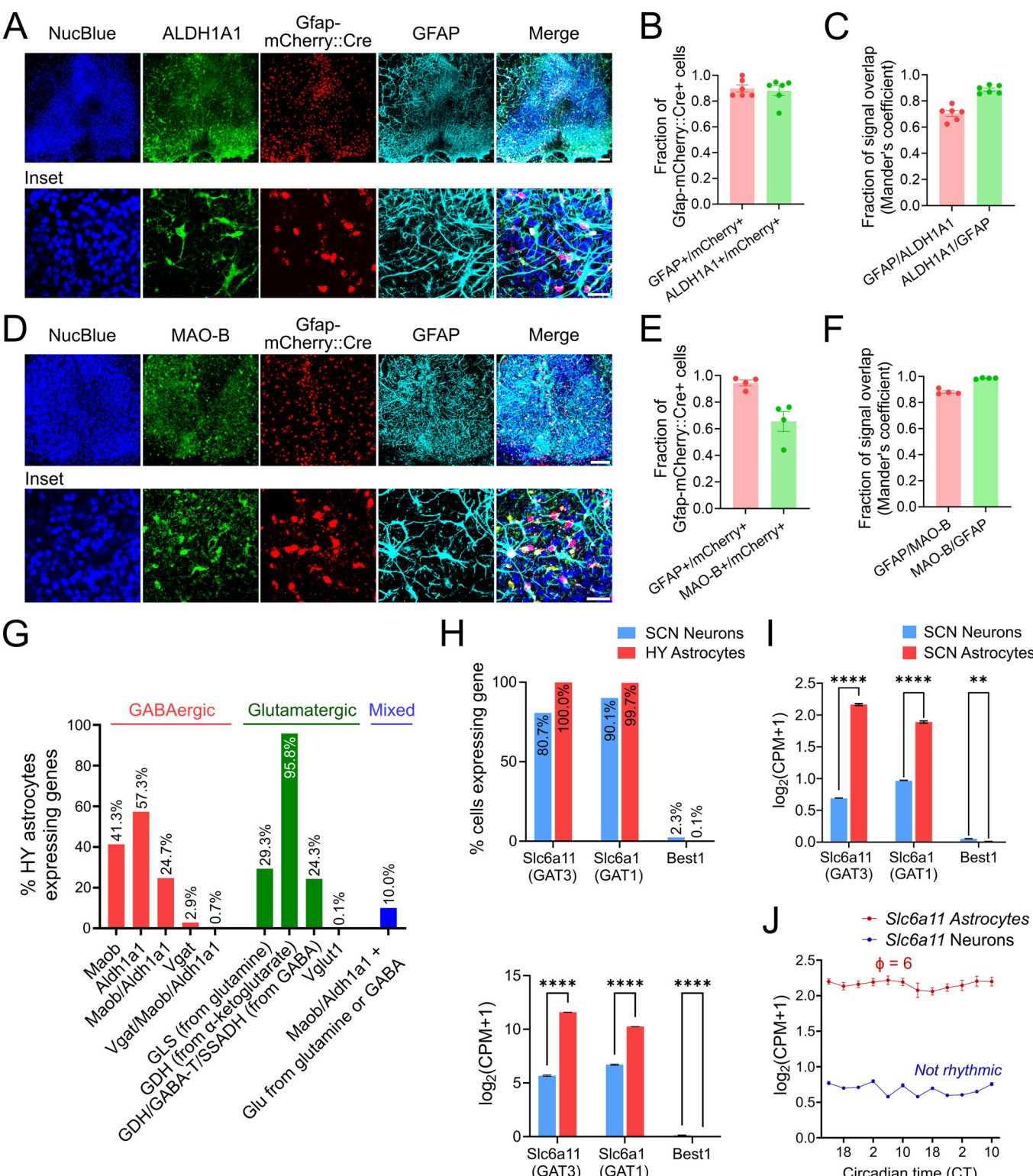

◀ **Figure EV3. MAO-B and ALDH1A1 protein expression in SCN astrocytes and characterization of GABAergic astrocytes.**

(A) Representative confocal image of an SCN slice expressing Gfap-mCherry::Cre counterstained with NucBlue, ALDH1A1 and GFAP antibodies. Inset with higher magnification shown below. Scale bar = 100 μm (top row), 30 μm (bottom row). (B) Fraction of Gfap-mCherry::Cre+ astrocytes co-expressing GFAP or ALDH1A1. $N = 6$ SCN slices. (C) Fraction of relative signal overlap, as determined by Mander's coefficient of GFAP and ALDH1A1. N = 6 SCN slices. (D) Representative confocal image of an SCN slice expressing Gfap-mCherry::Cre counterstained with NucBlue, MAO-B and GFAP antibodies. Inset with higher magnification shown below. Scale bar = 100 μm (top row), 30 μm (bottom row). (E) Fraction of Gfap-mCherry::Cre+ astrocytes co-expressing GFAP or MAO-B. $N = 4$ SCN slices. (F) Fraction of relative signal overlap, as determined by Mander's coefficient of GFAP and MAO-B. N = 4 SCN slices. (G) Characterization of potential GABA- and/or glutamate-producing hypothalamic astrocytes based on scRNA-Seq data from the Yao et al (2023) dataset. Graph shows percentage of astrocytes expressing one or more genes involved in GABA production (GABAergic), glutamate production (Glutamatergic) or both (Mixed). N = 20,549 hypothalamic astrocytes. (H) Top panel shows percentage of SCN neurons and hypothalamic astrocytes expressing GABA transporters *Slc6a11, Slc6a1* or *Best1* in the Yao et al, scRNA-Seq (2023) dataset. Bottom panel shows normalized gene expression levels of each GABA transporter gene. $N = 1836$ SCN neurons and $N = 20,549$ hypothalamic astrocytes. (I) Normalized gene expression levels of GABA transporters in SCN neurons and SCN astrocytes from the Wen et al, (2020) scRNA-Seq dataset. $N = 12,018$ SCN neurons and $N = 8,429$ SCN astrocytes. ****$P < 0.0001$, **$P = 0.0038$. (J) Time series of normalized gene expression levels of *Slc6a11*, encoding GAT3, in SCN astrocytes and neurons. $N = 12,018$ SCN neurons and $N = 8,429$ SCN astrocytes, with 238–2410 cells/timepoint. eJTK Cycle rhythmicity test with Benjamini–Hochberg correction, $P = 0.004$ with circadian peak at CT6 in astrocytes (indicated as ϕ on the plot), not significantly rhythmic in neurons as indicated. All graphs show mean ± SEM, and panels (H, I) show two-way ANOVA with post hoc Šidák's test. A detailed statistical report for this figure is provided in Appendix Table S1. Source data are available online for this figure.

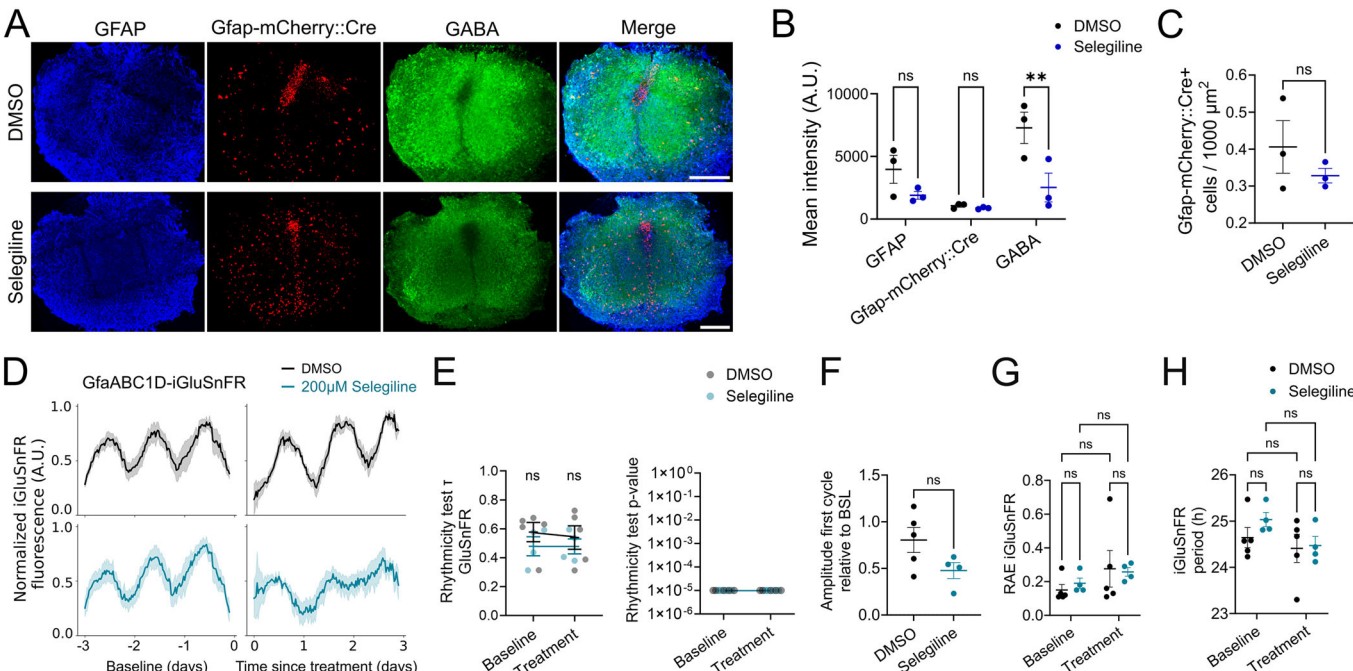

**Figure EV4. Selegiline treatment decreases GABA concentration in SCN slices without significantly affecting GFAP immunoreactivity or circadian rhythms of extracellular glutamate.**

(A) Representative confocal images of fixed SCN slices expressing Gfap-mCherry::Cre, and stained with antibodies against GFAP and GABA, 4 days after treatment with Selegiline or DMSO. Scale bar = 200 μm. (B) Quantification of mean fluorescence intensity of GFAP antibody, Gfap-mCherry::Cre and GABA antibody in SCN slices treated with Selegiline or DMSO. $N_{DMSO}$ = 3, $N_{Selegiline}$ = 3 SCN slices. Two-way ANOVA with matching and post hoc Šídák's test, $P$ = 0.0053 for GABA DMSO vs Selegiline. (C) Number of Gfap-mCherry::Cre-expressing cells per 1000 μm$^2$ tissue in slices treated with DMSO or Selegiline. $N_{DMSO}$ = 3, $N_{Selegiline}$ = 3 SCN slices. Two-tailed unpaired $t$ test. (D) Averaged, aligned time series of extracellular glutamate reporter (GfaABC1D-iGluSnFR) before and after treatment with 200 μM Selegiline (teal) or DMSO (black). $N_{DMSO}$ = 5, $N_{Selegiline}$ = 4 SCN slices. (E) Left panel shows $\tau$ values obtained from eJTK Cycle rhythmicity test on time series of GfaABC1D-iGluSnFR before and within 1–3 days after treatment with Selegiline or DMSO. Right panel shows the $P$ value obtained from eJTK Cycle rhythmicity test empirically calculated against random noise data. $N_{DMSO}$ = 5, $N_{Selegiline}$ = 4 SCN slices. (F) GfaABC1D-iGluSnFR amplitude of first cycle of rhythms (over 30 h) after treatment with Selegiline or DMSO relative to baseline. $N_{DMSO}$ = 5, $N_{Selegiline}$ = 4 SCN slices. Two-tailed unpaired $t$ test. (G) RAE of GfaABC1D-iGluSnFR rhythms before and after Selegiline treatment. (H) Period of GfaABC1D-iGluSnFR rhythms before and after Selegiline treatment. $N_{DMSO}$ = 5, $N_{Selegiline}$ = 4 SCN slices. Graphs (E, G, H) show two-way mixed-effects analysis with matching, with post hoc Šídák's test. All graphs, including time series, show mean ± SEM, except right panel in (E) which shows median ± interquartile range due to logarithmic scale, ns = non-significant, **$P$ < 0.01. A detailed statistical report for this figure is provided in Appendix Table S1. Source data are available online for this figure.

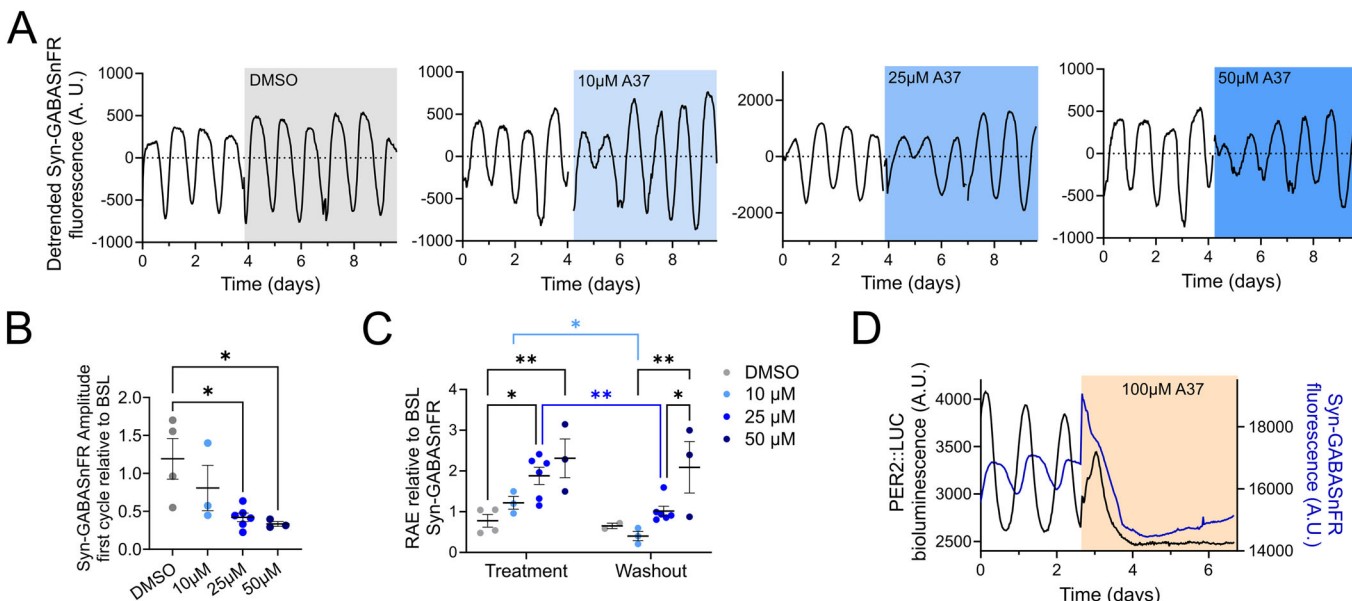

Figure EV5.    A37-mediated ALDH1A1 inhibition suppresses extracellular rhythms of GABA in a dose-dependent manner.

(A) Representative time series of SCN slices expressing Syn-GABASnFR before and after treatment with increasing concentrations of A37 from left to right: DMSO (0 μM A37), 10 μM, 25 μM and 50 μM A37. (B) Amplitude of the first cycle (30 h) of Syn-GABASnFR rhythms after treatment with increasing concentrations of A37 relative to baseline. One-way ANOVA, with post hoc Tukey's $t$ test shown, $P = 0.0161$ for DMSO-25μM, and $P = 0.0233$ for DMSO-50μM. $N_{DMSO} = 4$, $N_{10} = 3$, $N_{25} = 6$; $N_{50} = 3$. (C) RAE of Syn-GABASnFR rhythms with A37 treatment and after washout of increasing concentrations of A37 relative to baseline. Treatment: $N_{DMSO} = 4$, $N_{10} = 3$, $N_{25} = 6$; $N_{50} = 3$, washout: $N_{DMSO} = 2$, $N_{10} = 3$, $N_{25} = 6$; $N_{50} = 3$. Mixed-effects analysis with matching, timepoint effect $P < 0.01$, A37 dose effect $P < 0.01$, with post hoc Sidak's test, treatment: DMSO-25 μM, $P = 0.0206$, DMSO-50 μM, $P = 0.0052$; washout: 10 μM–50 μM, $P = 0.004$, 25 μM–50 μM, $P = 0.0477$; comparisons across timepoints are shown in the corresponding color. (D) Representative time series of PER2::LUC and Syn-GABASnFR before and after treatment with 100 μM A37, showing immediate tissue death. All graphs are mean ± SEM. *$P < 0.05$, **$P < 0.01$. A detailed statistical report for this figure is provided in Appendix Table S1. Source data are available online for this figure.

