## [Peer Review File · The EMBO Journal]

Rhythmic astrocytic GABA production synchronizes neuronal circadian timekeeping in the suprachiasmatic nucleus

Natalie Ness, Sandra Diaz-Clavero, Marieke Hoekstra, and Marco Brancaccio

Corresponding author(s): Marco Brancaccio (m.brancaccio@imperial.ac.uk)

Review Timeline:

Submission Date:	16th Apr 24
Editorial Decision:	3rd Jun 24
Revision Received:	16th Sep 24
Editorial Decision:	30th Sep 24
Revision Received:	18th Oct 24
Accepted:	4th Nov 24

Editors: Kelly M Anderson and Ioannis Papaioannou

Transaction Report:

Dear Dr. Brancaccio,

Thank you for submitting your manuscript for consideration by the EMBO Journal. It has now been seen by three referees whose comments are shown below.

Should you be able to address these criticisms in full, we could consider a revised manuscript. I should remind you that it is EMBO Journal policy to allow a single round of revision only and that, therefore, acceptance or rejection of the manuscript will depend on the completeness of your responses in this revised version. It would be good to discuss you plan to address the referee concerns and I will be available to do so in the coming weeks by email or zoom.

If you decide to thoroughly revise the manuscript for the EMBO Journal, please include a detailed point-by-point response to the referees' comments. Please bear in mind that this will form part of the Review Process File, and will therefore be available online to the community. For more details on our Transparent Editorial Process, please visit our website: <https://www.embo.org/embo-press>

Thank you for the opportunity to consider your work for publication. I look forward to your revision.

Yours sincerely,

Kelly M Anderson, PhD
Editor, The EMBO Journal
k.anderson@embojournal.org

- a point-by-point response to the referees' comments, with a detailed description of the changes made (as a word file).
- a word file of the manuscript text
- individual production quality figure files (one file per figure)
- a complete author checklist, which you can download from our author guidelines (<https://www.embopress.org/page/journal/14602075/authorguide>).
- Expanded View files (replacing Supplementary Information)

The revision must be submitted online within 90 days; please click on the link below to submit the revision online before 1st Sep 2024.

Referee #1:

The article entitled "Astrocytic GABA produced from polyamines synchronizes neuronal circadian timekeeping in the suprachiasmatic nucleus" by Brancaccio and colleagues provides evidence of a new role of astrocytes in the suprachiasmatic nuclei, the master circadian pacemaker in mammals. They found that astrocytes produce GABA by polyamine degradation, which is critical to maintain extracellular GABA rhythms within the SCN circuit.

First, they showed that while neuronal activation in the SCN organotypic slice shows a spatiotemporal gradient, astrocytes activate with a common phase. Then, by following presynaptic calcium and extracellular GABA levels over time, authors conclude that extracellular GABA levels might co-occur with astrocyte's activation. Indeed, when they inhibit neurotransmission, the oscillations of extracellular GABA are unaffected.

They further found that astrocytic GABA synthesis (through polyamine pathway) might be at least another mechanism regulating the circadian oscillation of extracellular GABA.

The article is well written, the work is technically sound and well performed, further experiments and explanations might increase the strength of the conclusions.

Major concerns:

1. While these data are solid, a demonstration of the transduction efficiency of neuronal and astrocytic reporters across the whole SCN slice might be necessary to make sure that no technical artifacts are responsible of the detection of cell-specific spatiotemporal gradient.
2. The circadian phase of extracellular GABA and Glutamate (described before) are both linked to astrocyte activation. Are those astrocytes different from each other? Are they differentially located across the SCN?
3. It has been shown that astrocytes are able to control extracellular GABA by increasing or decreasing the neurotransmitter uptake. Authors should discuss (or demonstrate) whether these mechanisms might be regulated by similar or different upstream pathways?
4. To attribute GABA synthesis to astrocytes and to the activation of the non-canonical polyamine biosynthetic pathway, authors explore the expression of the key enzymes involved in hypothalamic astrocytes taking advantage of publicly available single-cell RNA-sequencing data set of the mouse brain. Given the extremely high complexity and cell diversity of the SCN, the assessment of a SCN-based data set (already published as well) seems to be more appropriate. Besides RNA expression, the demonstration of the presence of MAO-B and ALDH1A1 in SCN astrocytes might be necessary. Moreover, it might be interesting to assess if the expression of these enzymes is rhythmically regulated and if there is a particular spatial distribution within astrocytes from the SCN.
5. Authors showed that inhibition of ALDH1A1 with A37 reduces the amplitude of extracellular GABA rhythms, here it could be interesting to show whether this effect depends on the dose of A37. In addition, it would be interesting to discuss (or demonstrate) how is the neuropeptide production within the SCN affected by the inhibition of the GABA non-canonical synthetic pathway.

Minor concerns:

1. Authors should also discuss whether these mechanisms would take place in vivo and what would be the relevance for the system. How this communication would look like when extra-SCN inputs are integrated? What could be the role of putrescine and the non-canonical pathway of GABA synthesis for circadian physiology in vivo?. Could this be a mechanism of peripheral synchronization of the SCN clock?
2. Since authors did not address directly the mechanism underlying the astrocytic uniform and sustained activation across the SCN network, the paragraph devoted to inter-astrocytic communication should be removed or substantially reduced.
3. The article is describing a mechanism taking place in SCN slices in vitro. Any reference to what could happen in the context of neurodegenerative diseases, although interesting, seems to be out of context here and should be removed.

Referee #2:

In this manuscript the authors describe a way how astrocytes can signal and provide timing information to neurons thereby synchronizing the neurons in the SCN. They show that one of the signals from astrocytes is GABA which is synthesized in astrocytes by a non-canonical pathway which involved polyamine degradation.

At first glance the manuscript is very smooth and the results look convincing. However, consulting previous work from the same group raises several questions the authors should address.

1. In Fig. 1C the authors show opposite cycling of the Syn-RCaMP1 and the GfaABC1D-Ick-GCaMP6 reporters with the Syn-RCaMP1 displaying a very fuzzy curve. This fuzziness is the base of this story. Comparing this same experiment with a previous publication (Neuron 93, 1420-1435, 2017) shows a different curve. The Syn-RCaMP1 oscillation is not fuzzy at all (see in the Neuron paper Fig. 1C) leaving the reader with some doubts about the experimentation here or in the previous paper (or may it be simply the different was of plotting, 24h vs 4 days in the Neuron paper?). There is no explanation given for this discrepancy in the discussion. Furthermore, in the Neuron paper the glutamate reporter (Neuron paper, Fig. 1E) shows exactly the same oscillation as the glutamate reporter described here (Fig. 1C). Hence, it is not clear what the relative contributions of GABA and Glutamate are in the synchronization of the SCN. The authors do not discuss this and also do not put the results presented here into the context of their previous model described in the Neuron paper.

2. Why does in Fig. 1C and E the red trace not have the same phase? It's the identical reporter. Is this connected to differences at what level in the SCN the slice has been taken? This raises the questions whether the observations described here (and also in the previous Neuron paper) are only correct at a specific level of the SCN. Are the responses in the anterior, middle and posterior SCN the same?

3. In Fig. 5B the extracellular GABA rhythms are shown. How do change the extracellular glutamate rhythms according to the treatment? And, how are affected the astrocytic intracellular GABA rhythms?

Referee #3:

This manuscript by Ness et al uses organotypic slice cultures of the clock center of the brain, the suprachiasmatic nucleus (SCN) combined with state-of-the-art viral vectors and imaging to try and disentangle the contribution of astrocytic signaling to circadian rhythmicity in the brain. The authors claim that astrocytes are a viable source of extracellular GABA that is necessary for rhythms in neuronal excitability. Overall, the idea of the manuscript is intriguing but the major sticking point is the interpretation of the main hypothesis. The data presented here could support both astrocytic release of GABA and impaired buffering of GABA by astrocytes (uptake vs. release). Several control experiments or additional experiments would be needed in order accurately interpret the findings. Below, please find a list of comments as found in the paper (not in order of concern).

- The imaging for the propagation across the SCN is beautiful. However, is there a concern that by using coronal sections of SCN the authors may be missing anterior/posterior waves across the SCN? Is it really that astrocytes a common phase, or is it that the connectivity of the SCN is altered by slices, eliminating potential signal propagation? The same holds true for the extracellular GABA reporter, though it does support the hypothesis that the GABA and astrocytic calcium are linked.
- For figure 2: perhaps I missed something, but why are the presynaptic GCaMP and the GABASnFR not compared to the same control signal, if comparing the phase and variance of phase is the purpose? Wouldn't it be a stronger, more appropriate comparison to choose either Per2::LUC or the RCaMP for both? Especially given the notable differences in sample size between the experiments.
- Figure 3E is very difficult to read and busy. Would it be possible to simplify and/or make larger? The blue/black combination for the GABA points are extremely hard to differentiate.
- The authors claim there is no amplitude change in the GABA signal after loss of VAMP2, but there looks to be a period change. Have the authors quantified this? Could it indicate a decoupling instead of a loss of rhythmicity in the PER2Luc trace? Is amplitude relative to baseline appropriate for the GABA trace if it appears that the GABA signal may be running down over time (the day/night change may be altered, but the running average may be running down between media changes). I appreciate the desynchrony analysis later on!
- What day post-transfection was the representative VAMP2 staining done on? The amplitude analysis says it was done baseline, days 1-6, and >7. Did the authors confirm knockdown on days 1-6? The methods say 4 days, but it's unclear from the data if the authors took it into account. Are days 1-6 valid for analysis? Were the same # of days used for baseline, days 1-6, and days >7? If this was in the methods and I missed it, I apologize. While the research group has extensive experience analyzing data like this, it would be informative to the audience because these parameters can easily change based on the setup.
- I'm a little confused by figure 3jk. Would it be possible to put in labels for which is per2 and which is GABA? Do you have to space apart the per2luc and gaba so hard in 3k? It makes it very hard to compare the conditions.
- For 3k, why are your baselines for the per2 luc so different? Is that expected? Shouldn't those cultures be, experimentally, the same group pre- virus treatment? It's very hard to see the control groups for the GABA traces with the color choices.
- The authors make a good case for the alternative gaba synthesis pathway to be in SCN astrocytes, and the use of publicly available data is clever. However, is there a possibility astrocytes could switch pathways depending on time-of-day and availability of GABA? Was there any time-of-day info for the brains used in the Alan Brain Atlas?
- Did the authors do any checks on slice health or health of the astrocytes after multi-day treatment with pharmaceutical agents that alter basic metabolic pathways? Can the authors be sure that these effects are specific to GABA metabolism and are not an effect of just making the slice or astrocytes unhealthy or reactive? The washout experiment is appreciated but doesn't really say anything about pathway specificity.
- I think the biggest conceptual sticking point I have is what if it is GABA buffering by astrocytes that is regulated by time-of-day, as buffering neurotransmitters and extracellular ions are a fundamental aspect of astrocyte function, instead of production of

GABA via a non-canonical pathway? Most of the data in this paper could be explained by uptake just as much as release from astrocytes, and it may fit in with the astrocytic literature better. Use of more specific tools to prevent production, or experiments critically testing uptake, would be needed. As of right now, I'm not sure the results are interpretable one way or another.

Dear Dr Anderson,

We are pleased to submit for your consideration our revised manuscript EMBOJ-2024-117617-T:

Astrocytic GABA produced by polyamine degradation synchronizes neuronal circadian timekeeping in the suprachiasmatic nucleus

We would like to thank the Reviewers for the positive evaluation of our manuscript, as well as their suggestions to further strengthening our findings.

While we provide point-on-point responses below, we want to briefly summarize three general key ways by which our original claims are further advanced in our revised manuscript, following Reviewers' feedback:

1) Are GABA biosynthetic enzymes MAOB and ALDH1A1 specifically expressed in SCN astrocytes?

We have addressed this in two ways. First, we have conducted independent immunohistofluorescence experiments in SCN slices and confirmed that the majority of astrocytes express MAOB and ALDH1A1 proteins by colocalization with two astrocyte markers (GFAP and gfap-mCherry::Cre AAV), consistent with scRNA-seq data.

Second, we have further confirmed our findings on hypothalamic astrocytic *Maob* and *Aldh1a1* expression from the Allen Brain Cell Atlas dataset, by analyzing an independent dataset specifically focused on SCN tissue (Wen et al., *Nat. Neurosci.* 2020). As this dataset also contains multiple circadian timepoints, we were able to rule out that SCN neurons or astrocytes could switch GABA synthesis pathways at different times of the day.

Moreover, by conducting this analysis, we revealed that *Aldh1a1* may be the time-rate-limiting enzyme for the circadian regulation of the pathway, with a peak of expression at CT13, consistent with high nighttime GABA.

2) Are the effects observed on GABA specific, or due to poor slice health, altered astrocyte health/ reactivity (gliosis)?

We have followed a multi-pronged approach to address this point. First, we observed no reduction in the number of rhythmic cells expressing reporters of neuronal calcium or clock gene expression throughout the A37 treatment, which inhibits ALDH1A1 (Figure 7), indicating tissue-level maintenance of rhythmicity. Moreover, we have conducted a dose-response experiment on A37, and fully reversible upon drug removal, at the concentration of the drug used in this study (25 μ M). In contrast, when toxic concentrations of the drug were used (100 μ M), all the reporters (not just the GABA one) dramatically and irreversibly flatten (Figure EV5).

Second, effects observed by the "harsh" treatment (MAO-B inhibition) are specific to GABA, as shown by the mild phenotypes observed in co-detected readouts, including neuronal calcium (Syn-jRCaMP1a), clock gene expression (PER2::LUC), and transmitters specifically released/ buffered by astrocytes in the SCN (GfaABC1D-iGluSnFR) (Brancaccio et al., 2017). (Figure 5, 6, EV4).

Finally, we could not detect an increase in markers of gliosis with our treatment (which we measured by GFAP immunoreactivity), or proliferative astrocyte numbers (measured by gfap-mCherry::Cre) (Figure EV4).

3) Are the effects observed on GABA indicative of altered uptake, rather than biosynthesis?

We had no reasons to believe that our drug interventions should be active on membrane-bound GABA transporters, as they target two molecularly unrelated biosynthetic enzymes (MAO-B and ALDH1A1). While these two unrelated drugs have been extensively validated (by other research groups) for being highly selective for their

respective molecular targets, they mutually lead to a coherent picture supporting the role for astrocytic GABA synthesis from polyamines as a synchronizer for the SCN circuit.

Nevertheless, we further addressed the Reviewer's concern, by conducting immunohistofluorescence experiments measuring GABA in SCN slices treated with selegiline and found a strong and significant reduction in GABA levels (Figure EV4), which contrasts with the increase in GABA levels expected when blocking GABA uptake (Patton et al., 2023).

The Reviewer's comment has however prompted us to conduct further investigations into GAT3 (astrocytic GABA transporter) expression in the SCN using scRNA-seq data. We found a peak of expression at circadian time (CT) 6 (Figure EV3), at the opposite time-of-day to *Aldh1a1* peak expression at nighttime (CT13). This is consistent with circadian fluctuations of extracellular GABA, and offers an elegant working model for GABA regulation by SCN astrocytes over the circadian cycle if both Patton et al. *PNAS* 2023 and our work are taken together: high GABA uptake during daytime when SCN GABAergic neurons are active will keep extracellular GABA levels low (by GAT3) whereas GABA production by astrocytes at nighttime will inhibit GABAergic neuronal activity. This model is now outlined in the Discussion.

Itemised responses

Reviewer #1:

The article entitled "Astrocytic GABA produced from polyamines synchronizes neuronal circadian timekeeping in the suprachiasmatic nucleus" by Brancaccio and colleagues provides evidence of a new role of astrocytes in the suprachiasmatic nuclei, the master circadian pacemaker in mammals. They found that astrocytes produce GABA by polyamine degradation, which is critical to maintain extracellular GABA rhythms within the SCN circuit. First, they showed that while neuronal activation in the SCN organotypic slice shows a spatiotemporal gradient, astrocytes activate with a common phase. Then, by following presynaptic calcium and extracellular GABA levels over time, authors conclude that extracellular GABA levels might co-occur with astrocyte's activation. Indeed, when they inhibit neurotransmission, the oscillations of extracellular GABA are unaffected. They further found that astrocytic GABA synthesis (through polyamine pathway) might be at least another mechanism regulating the circadian oscillation of extracellular GABA.

The article is well written, the work is technically sound and well performed, further experiments and explanations might increase the strength of the conclusions.

We thank the Referee for their positive evaluation of our work. We have taken their comments on board and strengthened our manuscript with a number of new experiments, further analyses and explanations.

Major concerns:

- 1. While these data are solid, a demonstration of the transduction efficiency of neuronal and astrocytic reporters across the whole SCN slice might be necessary to make sure that no technical artifacts are responsible of the detection of cell-specific spatiotemporal gradient.*

We thank the Reviewer for raising this potential concern. To address this, we have included new data in Figure EV1 (shown below). Our analysis reveals no significant regional differences (dorsal, ventral, medial, lateral) or biases in the expression of the neuronal (Syn-jRCaMP1a) and astrocytic (GfaABC1D-Ick-GCaMP6f) markers employed in our study

(Figure EV1A-C, see main text and methods for details). Additionally, given that some of our reporters are membrane-bound and can capture diffusible interstitial signals such as GABA and glutamate, we conducted counterstaining with an antibody against GFAP. We could not find any significant regional variations or biases in GFAP distribution across the SCN, based on which we deem it unlikely to introduce any specific bias in our results.

Finally, in response to related points raised by Reviewers 2 (comment #2) and 3 (#1), we also investigated whether the level of the cut across the anterior-posterior (A-P) axis may differentially impact the spatiotemporal distribution of phases of astrocytic GCaMP6, glutamate or GABA, and found no differences in the phase synchrony within coronal SCN slices across the anterior-posterior axis, inconsistent with further biases, due to the A-P position of the cut (Figure EV1D-J).

Figure EV1: Astrocytic calcium reporter is evenly expressed across the SCN, and spatiotemporal activity of astrocyte reporters is homogenous along the SCN anterior-posterior axis. (A) Schematic showing spatial regions within a coronal SCN slice with dorsal, medial, lateral and ventral edges. (B) Representative confocal image of an SCN slice expressing neuronal (Syn-jRCaMP1a) and astrocytic (GfaABC1D-Ick-GCaMP6f) calcium reporters, counterstained with NucBlue and GFAP antibody. Scale bar=200 μ m. (C) Quantification of mean fluorescence intensity of each reporter in the dorsal (D), ventral (V), medial (M) and lateral (L) SCN regions (see Methods), showing no detectable spatial differences in the expression of the neuronal or astrocytic calcium indicators, or GFAP staining intensity within the different SCN regions. N=4 SCN slices, two-way ANOVA with matching and post-hoc Šidák's test. (D) Schematic showing the shape of SCN nuclei along the anterior-posterior axis, with images of one representative SCN expressing Syn-jRCaMP1a and PER2::LUC for each region. (E) Top panel shows circular variance of phases across clusters in SCN slices expressing Syn-jRCaMP1a and GfaABC1D-Ick-GCaMP6f divided by region across A-P axis as shown in D. N=2-6 SCN slices per region. Bottom panel shows Rayleigh plot of circadian phases of GfaABC1D-Ick-GCaMP6f relative to co-detected Syn-jRCaMP1a (peaking at CT6) within each region across the A-P axis. Each dot represents one SCN slice, vector direction indicates mean phase, and vector length inversely indicates circular dispersion. (F) Top panel shows circular variance of cluster phases of Syn-jRCaMP1a and GfaABC1D-iGluSnFR by region. N=4 SCN slices per region. Bottom panel shows Rayleigh plot of circadian phases of GfaABC1D-iGluSnFR relative to co-detected Syn-jRCaMP1a by region. (G) Top panel shows circular variance of cluster phases of Syn-jRCaMP1a and Syn-GABASnFR by region. N=4-10 SCN slices per region. Bottom panel shows Rayleigh plot of circadian phases of Syn-GABASnFR relative to co-detected Syn-jRCaMP1a by region. All linear graphs show mean \pm SEM, graphs in top panel of E to G show two-way mixed-effects analysis with matching and post-hoc Šidák's test. Circular Rayleigh plots in bottom panel of E to G show Watson-Williams test of homogeneity of means. ns=non-significant.

2. *The circadian phase of extracellular GABA and Glutamate (described before) are both linked to astrocyte activation. Are those astrocytes different from each other? Are they differentially located across the SCN?*

We thank the Reviewer for raising this interesting question. As noted, we are monitoring ambient levels of GABA and glutamate, which cannot be easily pinpointed to specific astrocytic subpopulations. While our work is not primarily focused on the characterization of putative GABA/glutamate astrocytic subpopulations, we have now conducted a series of experiments to confirm the presence of astrocytes expressing enzymes to produce GABA and glutamate, and their relative representation in the hypothalamus. We believe this provides a first level description in terms of GABA/glutamate relationships in astrocytes and sets the stage for further functional investigations specifically addressing this point.

To examine the overlap between GABAergic and glutamatergic astrocytes in the SCN, we examined the percentage of astrocytes expressing genes involved in GABA or glutamate synthesis using the Allen Brain Cell Atlas scRNA-Seq data (see Figure EV3G below). Our key findings are:

- ~24.7% of astrocytes simultaneously express key genes necessary for synthesis of GABA (*Aldh1a1* and *Maob*).
- There are slightly more astrocytes that express genes involved in different pathways that could mediate glutamate synthesis, for example 29% of astrocytes express GLS, which is involved in glutamate synthesis from glutamine, and 24% of astrocytes express a range of genes required to synthesize Glutamate from GABA (GABA-T and SSADH to metabolize GABA, GDH to synthesize glutamate)
- There are some, but much fewer (~10%), astrocytes that could potentially mediate both GABA and glutamate synthesis

Our interpretation of these findings is that there are subsets of astrocytes in the hypothalamus that synthesize GABA and glutamate, which may be mostly distinct, but with a small proportion of these astrocytes potentially capable of synthesizing both GABA and glutamate. Note, our newly added immunohistochemical staining of MAOB and ALDH1A1 (see response to question #4) shows higher levels of expression of both of these enzymes in SCN slices (~60% and ~90% of astrocytes, respectively), suggesting the subpopulation of astrocytes capable of synthesizing GABA may be higher in the SCN.

As glutamatergic astrocytes have been identified in other brain regions that express *Vglut1* for vesicular release of glutamate (De Ceglia et al., 2023) and *Vgat* for GABA (Wang et al., 2013), we also investigated expression of these genes and found low expression of *Vgat* (2.9%) and *Vglut* (0.1%) in SCN astrocytes. This suggests that it would be difficult to draw conclusions about specific GABAergic and glutamatergic astrocytes in the hypothalamus based on the expression of these vesicular transporters.

In addition, we have followed the Referee's suggestion to confirm our RNAseq findings in SCN-specific astrocytes available in other studies (Wen et al. Nature Neuroscience 2020) (see answer to comment 4). However, we could not conduct this particular analysis in the SCN specific dataset. This is because there are too few cells with sufficient gene counts to provide an accurate estimate of the percentage of cells expressing a certain gene, which is a common issue in scRNA-Seq data (see Wang et al. *BMC Bioinformatics* 2019). The average gene count per cell in neurons of the Wen et al. dataset is about 10x lower than the average

gene count in neuronal population of the Yao et al. dataset (see Wen et al. Fig 1e vs Yao et al. Extended Data Fig 5a).

As for the spatial distribution of glutamatergic or GABAergic astrocytes, Wen et al. (2020) *Nat. Neurosci.* used laser capture microdissection and bulk RNA-Seq to determine the spatial distribution of RNA expression across the SCN. We have included a figure showing the distribution of genes involved in GABA or glutamate synthesis (see Figure to Reviewers below). This data suggests that genes involved in glutamate synthesis may be more evenly distributed throughout the SCN, whereas enzymes involved in astrocytic GABA synthesis (MAO-B and ALDH1A1) may be more represented in the SCN shell region. While this could be very interesting lead for future investigations into SCN astrocyte subtype specification (and we thank the Reviewer for their prompting us to looking into it), it will have to be further substantiated by more systematic investigations.

Figure EV3 G: Percentage of astrocytes expressing different profiles of genes involved in GABA or glutamate synthesis in the scRNA-Seq dataset from Yao et al. *Nature* (2023).

Figure to Reviewers: Spatial RNA distribution of genes involved in glutamate or GABA synthesis in the SCN. Data obtained from SCN 3D Atlas at yanlab.org.cn/scn-atlas/, showing data from LCM with bulk RNA-Seq from Wen et al. (2020) *Nat. Neurosci.* (A) Spatial distribution of expression of *Agt*, a pan-astrocytic marker as a reference for the approximate distribution of astrocytic genes (B) Spatial distribution of expression of *Gls* and *Glud1* (encoding GDH), which mediate glutamate synthesis from glutamine and alpha-ketoglutarate respectively. (C) Spatial distribution of expression of *Aldh1a1*, *Maob* and *Sirt2*, which mediate astrocytic GABA synthesis from putrescine.

3. It has been shown that astrocytes are able to control extracellular GABA by increasing or decreasing the neurotransmitter uptake. Authors should discuss (or demonstrate) whether these mechanisms might be regulated by similar or different upstream pathways?

SCN astrocytes have been shown to take up GABA from the extracellular space via GAT3 (Patton et al. PNAS 2023). In new analyses based on the Wen et al. *Nat. Neurosci.* (2020) scRNA-Seq SCN dataset, we found *Slc6a11*, which encodes GAT3, expression to be rhythmic in SCN astrocytes with a peak at CT6 (see Fig. EV3 below). This independently confirms the rhythmic astrocytic expression of *Slc6a11* with peak during the circadian daytime reported by Patton et al.

Moreover, we also found GABA synthesis enzyme *Aldh1a1* to be rhythmically expressed with an opposite peak during the circadian nighttime (CT13), while *Maob* expression is not rhythmic (see Fig. 4J below), also in agreement with a third RNA-Seq dataset (Pembroke et al. eLife (2015), wgpembroke.com/shiny/SCNseq).

We have now added the following paragraph in discussion to comment on a possible working model emerging from the new evidence:

“Notably, pharmacological inhibition of GAT3 in SCN slices leads to an accumulation of extracellular GABA, showing that GAT3 mediates GABA uptake in the SCN (Patton et al, 2023). If astrocytes can regulate extracellular GABA rhythms both via synthesis and uptake, how are GABA levels regulated by astrocytes across the circadian day? Using the scRNA-Seq dataset from (Wen et al, 2020), we found that *Aldh1a1* expression peaks at CT13 (Fig. 4J), consistent with an independent dataset from (Pembroke et al, 2015) (wgpembroke.com/hiny/SCNseq). In contrast, *Slc6a11*,

which encodes GAT3, peaked at CT6 in SCN astrocytes, consistent with (Patton *et al*, 2023) (Fig. EV3J). This suggests that astrocytes may generate extracellular GABA rhythms by switching from increased daytime GAT3-mediated GABA uptake, removing synaptically released GABA, to nighttime astrocytic GABA synthesis, replenishing extracellular GABA levels and inhibiting SCN neurons. This daily astrocyte switch will generate the circadian oscillations of extracellular GABA observed here and by (Patton *et al*, 2023). Disruption of either GABA production or uptake leads to a dysregulation of extracellular GABA rhythms due to low (Fig EV4B), or excess extracellular GABA (Patton *et al*, 2023), both ultimately disrupting circadian cycling of extracellular GABA tone.

It is not known how GABA production and uptake are regulated by the circadian clock or whether there are other shared upstream pathways. Rev-erba has been shown to positively regulate *Slc6a1* and *Slc6a11* expression by repressing E4bp4, a transcriptional repressor of multiple transporters, in the hippocampus and cortex (Zhang *et al*, 2021). However, no evidence currently links Rev-erba to *Aldh1a1* regulation.”

Figure EV3I-J: (I) Normalized gene expression levels of GABA transporters in SCN neurons and SCN astrocytes from the Wen *et al*. (2020) scRNA-Seq dataset. N=12,018 SCN neurons and N=8,429 SCN astrocytes. (J) Time series of normalized gene expression levels of *Slc6a11*, encoding GAT3, in SCN astrocytes and neurons. eJTK Cycle rhythmicity test with Benjamini-Hochberg correction, $p=0.004$ with circadian peak at CT6 in astrocytes, not significantly rhythmic in neurons as indicated. All graphs show mean \pm SEM, and panels H and I show two-way ANOVA with post-hoc Šidák’s test, ** = $p<0.01$, **** = $p<0.0001$.

Figure 4G-J: (G) UMAP plot of SCN-restricted scRNA-Seq dataset (Wen *et al*, 2020), with cell type annotation. (H) Normalized expression levels of genes involved in GABA biosynthesis. N= 12,018 SCN neurons and N=8,429 SCN astrocytes. Statistical test: two-way ANOVA with post-hoc Šidák's test. (I) Time series of normalized gene expression of neuronal GABA biosynthesis genes *Gad1* and *Gad2* in SCN neurons and astrocytes. eJTK Cycle rhythmicity test with Benjamini-Hochberg correction, all $p > 0.05$. (J) Time series of normalized expression levels of astrocytic GABA biosynthesis genes *Aldh1a1* and *Maob* in SCN neurons and astrocytes. eJTK Cycle rhythmicity test with Benjamini-Hochberg correction, $p = 0.022$ for *Aldh1a1* in astrocytes peaking at CT13 (indicated as ϕ on the plot), all other time series $p > 0.05$. All data show mean \pm SEM, ns= non-significant, **= $p < 0.01$, ****= $p < 0.0001$.

4. *To attribute GABA synthesis to astrocytes and to the activation of the non-canonical polyamine biosynthetic pathway, authors explore the expression of the key enzymes involved in hypothalamic astrocytes taking advantage of publicly available single-cell RNA-sequencing data set of the mouse brain. Given the extremely high complexity and cell diversity of the SCN, the assessment of a SCN-based data set (already published as well) seems to be more appropriate. Besides RNA expression, the demonstration of the presence of MAO-B and ALDH1A1 in SCN astrocytes might be necessary. Moreover, it might be interesting to assess if the expression of these enzymes is rhythmically regulated and if there is a particular spatial distribution within astrocytes from the SCN.*

The Reviewer raises very important points about a potential specific expression of the GABA biosynthetic pathway within SCN astrocytes when compared to the hypothalamic ones, as well as the temporal and spatial expression of this pathway.

To answer these questions, we have explored a second single-cell RNA-seq dataset from Wen *et al. Nat. Neurosci.* (2020) only including mouse SCN tissue, as well as performed further experiments to investigate MAO-B and ALDH1A1 protein expression in SCN astrocytes, now included in Figure 4G to J and EV3 I to J and associated methods, results and discussion sections.

Briefly, we found that:

- i. Notwithstanding the methodological differences in the Wen *et al.* study, we were able to confirm our findings in the ABC Atlas regarding the dichotomic astrocytic/neuronal segregation of GABA biosynthesis illustrated by astrocyte-specific expression of *Maob* and *Aldh1a1*, as opposed to neuronal expression of *Gad1* and *Gad2* (GAD67 and 65). As mentioned in response to Reviewer comment #2, there are too few cells with sufficient gene counts in this dataset to provide an accurate estimate of the specific percentages of cells expressing any given gene, so we could confirm the relative normalized gene expression levels, but not the percentage of cells expressing a genes, in the Wen *et al.* Dataset. (Figure 4G to J and EV3 I to J)
- ii. To address whether the expression of these enzymes is temporally regulated, we expanded analysis of the Wen *et al.* dataset, as it contains 12 circadian time points. We found that the expression of *Aldh1a1*, but not *Maob*, shows a circadian oscillation peak at CT13 after 2 days in DD, consistent with a night-time increase in extracellular GABA levels. This is consistent with bulk RNA-Seq data from Pembroke *et al. eLife* (2015), which can be accessed through the database wgpembroke.com/shiny/SCNseq, and shows that *Maob* expression is not rhythmic, while *Aldh1a1* expression shows a peak during the circadian nighttime in mouse SCN. This also suggests that time-gating of the pathway by the circadian clock may act specifically on this enzyme. Importantly, we did not observe an induction of the putrescine GABA synthesis pathway in neurons at any point during the circadian cycle, nor an induction of GAD65/67 expression in astrocytes.

Moreover, we confirmed expression of MAO-B and ALDH1A1 proteins in SCN astrocytes by immunohistofluorescence (Fig. EV3A to F, below). We found ALDH1A1 to be co-expressed in the majority of astrocytes labelled by Gfap-mCherry::Cre (90%) and overlapping with GFAP signal staining the astrocytic cytoskeleton (85%). MAO-B was co-expressed in ~65% of Gfap-mCherry::Cre⁺ astrocytes, and also highly overlapping with the spatial staining of GFAP.

Figure EV3: MAOB and ALDH1A1 protein expression in SCN slice astrocytes and characterization of GABAergic astrocytes. (A) Representative confocal image of an SCN slice expressing Gfap-mCherry::Cre counterstained with NucBlue, ALDH1A1 and GFAP antibody. Inset with higher magnification shown below. Scale bar=100µm (top row), 30µm (bottom row). (B) Fraction of Gfap-mCherry::Cre⁺ astrocytes co-expressing GFAP or ALDH1A1. N=6 SCN slices. (C) Fraction of relative signal overlap, as determined by Mander's coefficient of GFAP and ALDH1A1. (D) Representative confocal image of an SCN slice expressing Gfap-mCherry::Cre counterstained with NucBlue, MAOB and GFAP antibody. Inset with higher magnification shown below. Scale bar=100µm (top row), 30 µm (bottom row). (E) Fraction of Gfap-mCherry::Cre⁺ astrocytes co-expressing GFAP or MAOB. N=4 SCN slices. (F) Fraction of relative signal overlap, as determined by Mander's coefficient of GFAP and MAOB.

5. *Authors showed that inhibition of ALDH1A1 with A37 reduces the amplitude of extracellular GABA rhythms, here it could be interesting to show whether this effect depends on the dose of A37.*

Following the Reviewer's suggestion, we have now conducted experiments to investigate dose-response effects of A37 (New Figure EV5). A37 induces a dose-dependent (10 to 50 µM) suppression of GABA rhythms, as measured by progressively reducing amplitude and robustness (high RAE) with increasing doses of A37 (Figure EV5B-C), with GABA rhythms never fully abolished (in contrast to selegiline). Higher concentrations of A37 (100µM) killed the slices as shown by dramatic and irreversible drop of the signal across all reporters (GABASnFR and PER2::LUC) (Figure EV5D).

Figure EV5: A37-mediated ALDH1A1 inhibition suppresses extracellular rhythms of GABA in a dose-dependent manner. (A) Representative time series of SCN slices expressing Syn-GABASnFR before and after treatment with increasing concentrations of A37 from left to right: DMSO (0µM A37), 10µM, 25µM and 50µM A37. (B) Amplitude of the first cycle (30h) of Syn-GABASnFR rhythms after treatment with increasing concentrations of A37 relative to baseline. One-way ANOVA, with post-hoc Tukey's t-test shown. N=3-6 SCNs per condition. (C) RAE of Syn-GABASnFR rhythms with A37 treatment and after washout of increasing concentrations of A37 relative to baseline. Mixed effects analysis with matching, time point effect $p < 0.01$, A37 dose effect $p < 0.01$, with post-hoc Sidak's test. Comparisons across time points are shown in corresponding color. (D) Representative time series of PER2::LUC and Syn-GABASnFR before and after treatment with 100µM A37, showing immediate tissue death. All graphs are mean \pm SEM. *= $p < 0.05$, **= $p < 0.01$.

Minor concerns:

1. Authors should also discuss whether these mechanisms would take place *in vivo* and what would be the relevance for the system. How this communication would look like when extra-SCN inputs are integrated? What could be the role of putrescine and the non-canonical pathway of GABA synthesis for circadian physiology *in vivo*? Could this be a mechanism of peripheral synchronization of the SCN clock?

We thank the Reviewer for raising these important points regarding the *in vivo* relevance of our findings. We have revised a section of the discussion and added the following paragraph to the Discussion:

“Spatiotemporal waves of neuronal calcium and clock gene expression in the SCN encode photoperiodic input controlling seasonal behavioral adaptations (Evans & Gorman, 2016), and are implicated in the differentially phased engagement of downstream brain regions (Evans *et al*, 2011; Yamaguchi *et al*, 2003). In contrast, astrocytes display a highly uniform, sustained nighttime activation across the SCN tissue, with no discernible spatial waves, more akin to a pulsatile rhythm (Fig. 1 and 2I), which may suggest a more within-SCN role in timekeeping, rather than circadian engagement of downstream targets. “

“Our data suggest that circadian astrocytic GABA tone plays a significant role in synchronizing neuronal circadian rhythms within the SCN. This suggests that disturbances of SCN GABA rhythms may also indirectly weaken coordination of peripheral clocks by reducing coherent SCN output to the periphery. Whether or not GABA may also play a more direct “astrozeit” synchronization role within peripheral brain oscillators, remains to be tested.”

2. Since authors did not address directly the mechanism underlying the astrocytic uniform and sustained activation across the SCN network, the paragraph devoted to inter-astrocytic communication should be removed or substantially reduced.

We have now removed this paragraph as requested.

3. The article is describing a mechanism taking place in SCN slices in vitro. Any reference to what could happen in the context of neurodegenerative diseases, although interesting, seems to be out of context here and should be removed.

We have reduced this paragraph significantly. This is object of intense scrutiny in our current research funded by the UK Dementia Research Institute and we believe that the description of physiologically occurring circadian rhythms of GABA, rather than a merely pathological adaptation, may provide a mechanistic link to the early disruption of sleep-wake cycles observed in AD. We have taken care to carefully state how our findings (link between astrocytic GABA production and neuronal synchrony in the SCN) relate to the hypothesis presented for future research.

Reviewer #2:

In this manuscript the authors describe a way how astrocytes can signal and provide timing information to neurons thereby synchronizing the neurons in the SCN. They show that one of the signals from astrocytes is GABA which is synthesized in astrocytes by a non-canonical pathway which involved polyamine degradation.

At first glance the manuscript is very smooth and the results look convincing. However, consulting previous work from the same group raises several questions the authors should address.

We thank the Referee for their generally positive evaluation of our work. We provide answers to the questions highlighted below.

1. In Fig. 1C the authors show opposite cycling of the Syn-RCaMP1 and the GfaABC1D-Ick-GCaMP6 reporters with the Syn-RCaMP1 displaying a very fuzzy curve. This fuzziness is the base of this story. Comparing this same experiment with a previous publication (Neuron 93, 1420-1435, 2017) shows a different curve. The Syn-RCaMP1 oscillation is not fuzzy at all (see in the Neuron paper Fig. 1C) leaving the reader with some doubts about the experimentation here or in the previous paper (or may it be simply the different was of plotting, 24h vs 4 days in the Neuron paper?). There is no explanation given for this discrepancy in the discussion.

We thank the Referee for the opportunity to clarify this point. In Fig. 1C, we show the standard deviation of the Syn-jRCaMP1a signal across the 5 clusters that are spatially distributed across the SCN. This illustrates how the different RCaMP1 expressing clusters across the SCN have differentially phased rhythms. Fig. 1C in Brancaccio et al. (2017) shows the mean expression of Syn-jRCaMP1a across the whole slice, (as opposed to the standard deviation) and no cluster analysis was conducted in that manuscript. This corresponds to what we also show here in Appendix Fig. S1A, showing consistent curves to Fig. 1C in the Brancaccio et al. (2017) paper for Syn-jRCaMP1. Differentially phased rhythms of neuronal calcium across the SCN have also been described by other groups, including Pauls et al. *EJN* 2014, Enoki et al. *PNAS* 2012, and Evans et al. *PLOS ONE* 2011. To further clarify this point, we have now added a more detailed description of what is depicted in Fig. 1C in the Results section.

2. Why does in Fig. 1C and E the red trace not have the same phase? It's the identical reporter. Is this connected to differences at what level in the SCN the slice has been taken? This raises the questions whether the observations described here (and also in the previous Neuron paper) are only correct at a specific level of the SCN.

Are the responses in the anterior, middle and posterior SCN the same?

We thank the Reviewer for their comment. Figure 1C and E show the relative relationship between the same neuronal calcium reporter compared to co-detected glutamate, or astrocyte calcium reporters, respectively. The key takeaway from these figures is that both glutamate and astrocyte calcium reporters are antiphasic to the neuronal calcium signal, and while neuronal calcium shows high standard deviation across the SCN (see point above), co-detected extracellular glutamate or astrocytic calcium are highly synchronous. As for absolute phase, these are distinct experiments, so we could not compare them across in terms of absolute time, therefore we compare them based on the subjective circadian time-of-day of each sample relative to the timing of the peak of neuronal calcium.

We acknowledge the Reviewer's point regarding the potential confusion caused by displaying time in hours rather than circadian time (CT) in Fig. 1C and E (see figure below). Therefore, we have now revised figures throughout the manuscript to consistently reflect circadian time, using reference reporters of known phase (Syn-jRCaMP1a and PER2::LUC, Brancaccio et Neuron 2017) to align other reporters, as indicated in the figure legends.

Regarding the second concern about differences across the anterior-posterior axis of the SCN, we have investigated this question further by dividing our samples into anterior, medial and posterior based on their morphology (labelled by a colleague in blind), and added an analysis of this in a new Figure EV1, shown below. We found no difference in the phase synchrony within SCN slices taken along the anterior-posterior axis in astrocytic calcium, extracellular glutamate or extracellular GABA rhythms (Fig EV1E to G). Moreover, we found no differences along the anterior-posterior axis in the phase distance between neuronal circadian phase and astrocytic calcium, extracellular glutamate or extracellular GABA (Fig EV1E to G), suggesting consistency across the SCN for ensemble astrocyte rhythmicity and their relationship to neurons within the same coronal plane. For an extended discussion on this, please see Reviewer 3 comment #1.

Figure 1C and E: Astrocytes and neurons of the SCN show distinct patterns of network synchronization. (C) Representative standard deviation of cluster time series within SCN co-expressing Syn-jRCaMP1a and GfaABC1D-Ick-GCaMP6f, showing reduced variance across clusters in astrocytic calcium compared to neuronal calcium. (E) Representative standard deviation of cluster time series within a slice co-expressing Syn-jRCaMP1a and GfaABC1D-GluSnFR, showing similarly reduced variance across clusters of astrocytic glutamate, when compared to neuronal calcium.

Figure EV1D-G: Spatiotemporal activity of astrocyte reporters is homogeneously expressed along the SCN anterior-posterior axis. (D) Schematic showing the shape of SCN nuclei along the anterior-posterior axis, with images of one representative SCN expressing Syn-jRCaMP1a and PER2::LUC for each region. (E) Top panel shows circular variance of phases across clusters in SCN slices expressing Syn-jRCaMP1a and GfaABC1D-lck-GCaMP6f divided by region across A-P axis as shown in D. N=2-6 SCN slices per region. Bottom panel shows Rayleigh plot of circadian phases of GfaABC1D-lck-GCaMP6f relative to co-detected Syn-jRCaMP1a (peaking at CT6) within each region across the A-P axis. Each dot represents one SCN slice, vector direction indicates mean phase, and vector length inversely indicates circular dispersion. (F) Top panel shows circular variance of cluster phases of Syn-jRCaMP1a and GfaABC1D-iGluSnFR by region. N=4 SCN slices per region. Bottom panel shows Rayleigh plot of circadian phases of GfaABC1D-iGluSnFR relative to co-detected Syn-jRCaMP1a by region. (G) Top panel shows circular variance of cluster phases of Syn-jRCaMP1a and Syn-GABASnFR by region. N=4-10 SCN slices per region. Bottom panel shows Rayleigh plot of circadian phases of Syn-GABASnFR relative to co-detected Syn-jRCaMP1a by region. All linear graphs show mean ± SEM, graphs in top panel of E to G show two-way mixed-effects analysis with matching and post-hoc Šídák's test. Circular Rayleigh plots in bottom panel of E to G show Watson-Williams test of homogeneity of means. ns=non-significant.

3. In Fig. 5B the extracellular GABA rhythms are shown. How do change the extracellular glutamate rhythms according to the treatment? And, how are affected the astrocytic intracellular GABA rhythms?

To address the first point, we have conducted additional experiments investigating the effect of selegiline on extracellular glutamate rhythms detected by GfaABC1D-iGluSnFR. This is now shown in Figure EV4D-H (also shown below). Extracellular glutamate rhythms are only mildly affected after selegiline treatment, remain highly rhythmic (eJTK cycle rhythmicity test p -value ≤ 0.0001 , Fig. EV4E), with no significant effects on amplitude, RAE and period, but slightly higher variability across SCN slices (Fig. EV1D).

The second point about the potential presence of astrocytic intracellular GABA rhythms is interesting. Unfortunately, there are no available probes for live imaging which would allow us to monitor intracellular GABA in astrocytes, at the best of our knowledge. Creating new probes would be out of the scope of the current work. Nevertheless, we have now added an immunohistochemical assessment of GABA levels in SCN slices treated with selegiline (or DMSO). We find a strong reduction in GABA in selegiline-treated SCN slices (Fig EV4A-B), as expected for a treatment inhibiting GABA biosynthesis.

Figure EV4: Selegiline treatment decreases GABA concentration in SCN slices without significantly affecting GFAP immunoreactivity or circadian rhythms of extracellular glutamate. (A) Representative confocal images of fixed SCN slices expressing Gfap-mCherry::Cre, and stained with antibodies against GFAP and GABA, 4 days after treatment with Selegiline or DMSO. Scale bar = 200 μm. (B) Quantification of mean fluorescence intensity of GFAP antibody, Gfap-mCherry::Cre and GABA antibody in SCN slices treated with Selegiline or DMSO. Two-way ANOVA with matching and post-hoc Šidák's test. (C) Number of Gfap-mCherry::Cre expressing cells per 1000 μm² tissue in slices treated with DMSO or Selegiline. Two-tailed unpaired t-test. (D) Averaged, aligned time series of extracellular glutamate reporter (GfaABC1D-iGluSnFR) before and after treatment with 200 μM Selegiline (teal) or DMSO (black). N=4-5 SCN slices per condition. (E) Left panel shows t values obtained from eJTK Cycle rhythmicity test on time series of GfaABC1D-iGluSnFR before and within 1-3 days after treatment with Selegiline or DMSO. Right panel shows the p-value obtained from eJTK Cycle rhythmicity test empirically calculated against random noise data. (F) GfaABC1D-iGluSnFR amplitude of first cycle of rhythms (over 30h) after treatment with Selegiline or DMSO relative to baseline. Two-tailed unpaired t-test. (G) RAE of GfaABC1D-iGluSnFR rhythms before and after Selegiline treatment. (H) Period of GfaABC1D-iGluSnFR rhythms before and after Selegiline treatment. Graphs E, G and H show two-way mixed effects analysis with matching, with post-hoc Šidák's test. All graphs, including time series, show mean ± SEM, except right panel in E which shows median ± interquartile range due to logarithmic scale, ns=non-significant, **=p<0.01.

Reviewer #3:

This manuscript by Ness et al uses organotypic slice cultures of the clock center of the brain, the suprachiasmatic nucleus (SCN) combined with state-of-the-art viral vectors and imaging to try and disentangle the contribution of astrocytic signaling to circadian rhythmicity in the brain. The authors claim that astrocytes are a viable source of extracellular GABA that is necessary for rhythms in neuronal excitability. Overall, the idea of the manuscript is intriguing but the major sticking point is the interpretation of the main hypothesis. The data presented here could support both astrocytic release of GABA and impaired buffering of GABA by astrocytes (uptake vs. release). Several control experiments or additional experiments would be needed in order accurately interpret the findings. Below, please find a list of comments as found in the paper (not in order of concern).

We thank the Reviewer for their thorough evaluation of our manuscript, and suggestions to further strengthen our dataset. We have now addressed the Reviewer's concerns regarding the distinction between astrocytic release and uptake of GABA, as well as the potential impact of various confounders (aspecificity of responses/ general toxicity/ reactive gliosis) which may impact on interpretation of our findings, by conducting several experiments and analyses itemised below.

- 1. The imaging for the propagation across the SCN is beautiful. However, is there a concern that by using coronal sections of SCN the authors may be missing anterior/posterior waves across the SCN? Is it really that astrocytes a common phase, or is it that the connectivity of the SCN is altered by slices, eliminating potential signal propagation? The same holds true for the extracellular GABA reporter, though it does support the hypothesis that the GABA and astrocytic calcium are linked.*

We thank the Reviewer for raising this point. We have now addressed this question by categorizing our SCN samples into anterior, medial and posterior based on their morphology (labelled by another colleague in blind) (Figure EV1D-G). We then evaluated waves of both neuronal and astrocytic markers as well as their reciprocal phase relationships within coronal slices cut along the anterior-posterior (A-P) axis, to rule out, or indicate, region-specific astrocyte-neuronal interplay. We found no difference in the phase synchrony of neuronal calcium, astrocytic calcium, extracellular glutamate or extracellular GABA rhythms, or their relative phase relationship with neuronal calcium (Fig EV1D-G), suggesting consistency across the A-P axis for ensemble astrocyte rhythmicity and their relationship with neurons cut within the same coronal planes.

Alternative anatomical planes for the cutting have not been pursued, either based on previous knowledge from the literature, e.g. horizontal cuts sever core-to-shell synaptic connections, required for the retention of neuronal timekeeping in SCN slices (Albus et al Current Biology 2005; Abel et al. PNAS 2016), or because technically challenging to set up and analyze within the limited time scope of this revision (sagittal cuts). Based on our findings in Figure EV1, however, we have no evidence to support that astrocyte-neuronal interplay would differ along the A-P axis. This is consistent with the idea that variations of ambient astrocytic GABA and glutamate levels monitored within the timescale of hours considered in this study, will freely diffuse within the SCN interstitial space. While we remain agnostic about any astrocytic vesicle release at shorter timescales (which may be more affected by A-P physical cuts, if selective), we found very low *Vgat* (<3%) and *Vglut1* (<1%) expression levels in hypothalamic astrocytes (Figure EV3G). While a multi-temporal scale, tissue-wide investigation of astrocyte-neuronal signalling within the SCN (and beyond it)

would be of certain interest, we believe it to be beyond the scope of this report. We have added comments regarding this in the updated discussion.

“As astrocyte subpopulations have been shown to release GABA (Wang *et al*, 2013) and glutamate (de Ceglia *et al*, 2023) through vesicular release in other brain regions, we also examined expression of the *Vglut1* and *Vgat*, responsible for vesicular glutamate and GABA release, respectively. We found very low expression levels of both *Vglut1* (0.1%) and *Vgat* (2.9%) in hypothalamic astrocytes, with only 0.7% of astrocytes co-expressing *Maob*, *Aldh1a1* and *Vgat* (Fig. EV3G). While this does not rule out contributions to GABA levels by astrocytic *Vgat*-mediated release in the SCN, especially at shorter time scales, they appear unlikely mediators of the circadian-scale tissue-wide oscillations of extracellular GABA reported here. Nevertheless, future tissue-wide investigations of astrocyte-neuronal signalling at multiple timescales within the SCN, also inclusive of different anatomical planes (e.g. sagittal) may help disentangle the inherently complex features mediating astrocyte-neuronal interplay within the SCN and beyond.”

Figure EV1D-G: Spatiotemporal activity of astrocyte reporters is homogeneously expressed along the SCN anterior-posterior axis. (D) Schematic showing the shape of SCN nuclei along the anterior-posterior axis, with images of one representative SCN expressing Syn-jRCaMP1a and PER2::LUC for each region. (E) Top panel shows circular variance of phases across clusters in SCN slices expressing Syn-jRCaMP1a and GfaABC1D-lck-GCaMP6f divided by region across A-P axis as shown in D. N=2-6 SCN slices per region. Bottom panel shows Rayleigh plot of circadian phases of GfaABC1D-lck-GCaMP6f relative to co-detected Syn-jRCaMP1a (peaking at CT6) within each region across the A-P axis. Each dot represents one SCN slice, vector direction indicates mean phase, and vector length inversely indicates circular dispersion. (F) Top panel shows circular variance of cluster phases of Syn-jRCaMP1a and GfaABC1D-iGluSnFR by region. N=4 SCN slices per region. Bottom panel shows Rayleigh plot of circadian phases of GfaABC1D-iGluSnFR relative to co-detected Syn-jRCaMP1a by region. (G) Top panel shows circular variance of cluster phases of Syn-jRCaMP1a and Syn-GABASnFR by region. N=4-10 SCN slices per region. Bottom panel shows Rayleigh plot of circadian phases of Syn-GABASnFR relative to co-detected Syn-jRCaMP1a by region. All linear graphs show mean ± SEM, graphs in top panel of E to G show two-way mixed-effects analysis with matching and post-hoc Šidák's test. Circular Rayleigh plots in bottom panel of E to G show Watson-Williams test of homogeneity of means. ns=non-significant.

2. For figure 2: perhaps I missed something, but why are the presynaptic GCaMP and the GABASnFR not compared to the same control signal, if comparing the phase and variance of phase is the purpose? Wouldn't it be a stronger, more appropriate comparison to choose either *Per2::LUC* or the *RCaMP* for both? Especially given the notable differences in sample size between the experiments.

We thank the Reviewer for this comment and we agree that comparing to the same reference would facilitate comparisons. We have therefore now conducted further experiments and used *PER2::LUC* as a shared reference for both *Syp::GCaMP6f* and *Syn-GABASnFR* (see revised Figure 2 below). The previous comparison of *Syn-GABASnFR* with *Syn-jRCaMP1a* has been moved to Figure EV2. All additional experiments have also been added to the general comparisons across all reporters shown in Figure 2G-I, which further consolidate our findings.

Figure 2: Circadian rhythms of extracellular GABA co-segregate with reporters of astrocyte activity and not with neuronal ones. (A) Period and relative amplitude error (RAE) of circadian oscillations of *PER2::LUC*, *Syp::GCaMP6s*, *Syn-jRCaMP1a* and *Syn-GABASnFR*. One-way mixed effects model with matching, and post-hoc Tukey's test shown. All non-significant. Each dot presents one SCN slice. *PER2::LUC* (N=26 SCN slices), *Syp::GCaMP6s* (N=7), *Syn-jRCaMP1a* (N=25) and *Syn-GABASnFR* (N=28). (B) Rayleigh plot showing circadian phase of *Syp::GCaMP6s* (dark green) and *Syn-GABASnFR* (blue) rhythms, relative to co-detected *PER2::LUC*. Each point indicates 1 SCN slice. Vector direction indicates mean phase, length of vector is a measure of circular dispersion. (C) Representative circadian phase cluster map of co-detected *PER2::LUC* and *Syp::GCaMP6s* (top) or *Syn-GABASnFR* (bottom). One SCN nucleus is shown (dorsal (D) and medial (M) area indicated). Color bars indicate cluster phases, NR = non-rhythmic. White arrow indicates the direction of the phase progression. (D) Representative standard deviation of cluster time series of co-detected *PER2::LUC* and *Syp::GCaMP6s* (top) or *Syn-GABASnFR* (bottom). (E) Inter-cluster phase dispersal (measured by circular variance) of co-detected *PER2::LUC* and *Syp::GCaMP6s* (N=5 SCN slices), or *Syn-GABASnFR* (N=22). Paired two-tailed t-test shown. (F) PDF of cluster phase variance for each co-detected reporter, with Kolmogorov-Smirnov test shown. (G) Inter-cluster phase dispersal of *Syn-jRCaMP1a* (N=46 SCN slices), *Syp::GCaMP6s* (N=7), *PER2::LUC* (N=42), *GfaABC1D-Ick-GCaMP6f* (N=12), *GfaABC1D-GluSnFR* (N=8) and *Syn-GABASnFR* (N=30). Mixed effects analysis with matching, $p < 0.0001$, and Tukey's post-hoc test shown. (H) Circular histogram of directionality of phase progression across the SCN (see representative white arrows in C). Frequency of SCN slices within bar indicated by y-axis circle labels. The vector angle indicates the mean direction, length of the vector indicates

circular dispersion. Rayleigh test of uniformity shown. (I) Correlation of mean circular variance of cluster phases, circular variance of phase wave directionality and mean phase (CT). All scatter graphs show mean±SEM. ns=non-significant, *=p<0.05, **=p<0.01, ***=p<0.001, ****=p< 0.0001.

- 3. Figure 3E is very difficult to read and busy. Would it be possible to simplify and/or make larger? The blue/black combination for the GABA points are extremely hard to differentiate.*

We increased the size of the graphs and simplified Figure 3E by separating the treatment groups (Syn-mCh and Syn-TeLC) into separate columns to make it easier to compare the groups at each timepoint.

- 4. The authors claim there is no amplitude change in the GABA signal after loss of VAMP2, but there looks to be a period change. Have the authors quantified this?*

Could it indicate a decoupling instead of a loss of rhythmicity in the PER2Luc trace?

We agree with the Reviewer that a change in GABA period would be an important effect to consider, however we found no consistent and significant effect on the period of either PER2::LUC or Syn-GABASnFR rhythms. We have now moved the period quantification of PER2::LUC and Syn-GABASnFR to Figure 3F (below) and added a description in text to further clarify this.

Regarding the second point about the reduction of overall PER2::LUC amplitude indicating a desynchronization rather than a loss of rhythmicity in the PER2::LUC trace, we agree with the Reviewer. Indeed, our data suggest that the decreased PER2::LUC amplitude may be attributable to desynchronization among PER2::LUC-expressing cells, as evidenced by the cluster analysis presented in Figures 3H and L (below). Such a desynchronization does not affect co detected GABA rhythms in amplitude and synchrony, as shown in Figures 3E, J, L and M. We have now elaborated on these findings in the Results section to more explicitly convey this interpretation.

Figure 3: Disrupting synaptic GABA transmission via tetanus toxin light chain de-synchronizes PER2::LUC clusters without affecting circadian oscillations of GABA. (A) Schematic showing mechanism of blockade of GABAergic synaptic transmission by TeLC-dependent cleavage of the SNARE complex protein VAMP2. (B) Representative widefield images and insets of SCN slices expressing Syn-mCherry (control, top) or Syn-TeLC-mCherry (bottom) labelled with anti-VAMP2 antibody and DAPI, scale bar=200 μ m. Inset scale bar =20 μ m. (C) Quantification of mean VAMP2 intensity within the SCN. N=3 slices per each condition, two-tailed t-test. (D) Representative detrended time series of SCN slices expressing reporters for extracellular GABA (Syn-GABASnFR) and PER2::LUC before and after treatment with Syn-mCherry (left) or Syn-TeLC-mCherry (right). (E) Amplitude relative to baseline of rhythms of PER2::LUC (left) (N=5 per condition) and Syn-GABASnFR (right) (N=8 per condition). Two-way ANOVA with post-hoc Šidák's test shown. (F) Circadian period of PER2::LUC (left) and Syn-GABASnFR rhythms (right). Two-way ANOVA with post-hoc Šidák's test shown. (G) Mean fluorescence intensity of Syn-GABASnFR signal across timepoints. Two-way ANOVA with post-hoc Šidák's test shown. (H) Representative circadian phase cluster map of PER2::LUC before and >7 days after transduction with Syn-TeLC-mCherry (bottom) or mCherry control (top). One SCN nucleus is shown, orientation as indicated (dorsal-D, medial-M). Color bar indicates circadian phases of clusters, NR=non-rhythmic. White vector indicates the directionality of phase progression. (I) Representative standard deviation of cluster time series shown in H. (J) Representative circadian phase cluster map of Syn-GABASnFR co-detected with PER2::LUC shown in (H) before and >7 days after viral transduction. (K) Representative standard deviation of cluster time series shown in J. (L) Inter-cluster phase dispersal of PER2::LUC (top) or Syn-GABASnFR (bottom) relative to baseline, with two-tailed t-test. (M) PDF of cluster phase variance shown in L. All data shown mean \pm SEM unless otherwise indicated. For longitudinal data, connecting lines are shown between means. ns=non-significant, *= p <0.05, ***= p <0.001.

5. *Is amplitude relative to baseline appropriate for the GABA trace if it appears that the GABA signal may be running down over time (the day/night change may be altered, but the running average may be running down between media changes). I appreciate the desynchrony analysis later on!*

We agree with the Reviewer that it would be useful to also quantify the running average of the GABA signal across the treatment to see if there are any TeLC-dependent effects on the total concentration of GABA. We have quantified this using the mean GABA concentration before treatment, days 1-6 after transduction and >7 days after transduction with Syn-TeLC or mCherry control AAV. This graph is now shown in Fig. 3G (below). We found a mild increase in the running average of GABA signal in both experimental groups (Syn-TeLC and Syn-mCh) over time, which may be due to the fact that AAV-reporter expression is moderately increasing over time (regardless of TeLC, or control treatment). However, we found no significant difference in mean GABA levels between Syn-TeLC and control AAV Syn-mCh at any timepoint, further corroborating that extracellular GABA levels are not reduced when synaptic GABA vesicle release is impaired.

Fig 3G. Mean Syn-GABASnFR fluorescence intensity across treatment period for SCN slices transduced with Syn-mCherry or Syn-TeLC. Mean \pm SEM, two-way ANOVA with matching with post-hoc Sidak's test, ns = non-significant.

6. *What day post-transfection was the representative VAMP2 staining done on? The amplitude analysis says it was done baseline, days 1-6, and >7.*

Did the authors confirm knockdown on days 1-6?

The methods say 4 days, but it's unclear from the data if the authors took it into account. Are days 1-6 valid for analysis? Were the same # of days used for baseline, days 1-6, and days >7? If this was in the methods and I missed it, I apologize. While the research group has extensive experience analyzing data like this, it would be informative to the audience because these parameters can easily change based on the setup.

We thank the Reviewer for pointing this out and have added further details in the manuscript to clarify this point:

- i. VAMP2 depletion was confirmed by immunohistofluorescence on PFA-fixed slices at the end of the longitudinal experiment, which was day 12 post-transduction. We have now clarified this information in the Methods section.
- ii. We did not directly confirm knockdown on days 1-6, as fixing the slices would have killed them and prevented us from measuring circadian rhythms of PER2::LUC by live imaging, given the longitudinal nature of our experiment. Our reason for providing

data measured over days 1-6 is to show how the circadian phenotype (declining of PER2::LUC amplitude) develops over time, in TeLC-treated samples (but not controls). We also note that Syn-mCherry and TeLC-mCherry were added at the same time to littermate samples within each experimental session to rule out any potential inter-group bias due to random variability in expression dynamics. We divided the analysis into baseline (before Syn-mCherry and TeLC-mCherry addition), days 1-6 (knockdown not fully in effect but slowly getting some knockdown) and days >7 (days 7-12, AAV fully expressed for more than 7 days, full knockdown expected). We accept that precise staging of initial phenotype effects after TeLC addition is somewhat arbitrary, but based on the rise in the expression of mCherry fluorescent from ~day 4 post-transduction (see panel below) and initial observations of a developing phenotype. We have now clarified this in the Methods section and stated that days 1-6 should be interpreted as effects on the circadian phenotype following a partial/initial knockdown.

- iii. We used 5 days for the analysis of the rhythms. We have now specified this in the Methods section.

Figure to Reviewer: Longitudinal mCherry fluorescence signal before and after transduction with SynmCherry or Syn-TeLC-mCherry.

7. *I'm a little confused by figure 3jk. Would it be possible to put in labels for which is per2 and which is GABA? Do you have to space apart the per2luc and gaba so hard in 3k? It makes it very hard to compare the conditions.*

We added the reporter name to the y-axis labels in Fig 3L (formerly 3J), as well as reporter labels at the top of each graph in Fig 3M (formerly 3K). We also rearranged Fig 3M (K) into a grid of reporters and treatment groups, such that all experimental conditions differing by only one variable are directly next to each other to make it easier to compare.

8. *For 3k, why are your baselines for the per2 luc so different? Is that expected? Shouldn't those cultures be, experimentally, the same group pre- virus treatment?*

The probability distribution function of phase synchrony across clusters can vary slightly due to the sample-to-sample baseline variability in phase synchrony. The most likely reason for the differences between baselines of the PER2::LUC signal in this case is the slight difference in sample size between Syn-mCh and Syn-TeLC groups, which have n=4 and 6, respectively, leading to a slightly sharper PDF in the Syn-TeLC baseline which has more samples to draw from. When comparing both raw circular variance of cluster phases and the derived PDF, however, we always compare data matched within samples (i.e. *time-matched*: pre- and post-treatment within the same sample; *readout-matched*: reporters expressed

within the same slices) to mitigate the potential effects of varying sample sizes or biological variability across different experimental groups.

9. *It's very hard to see the control groups for the GABA traces with the color choices.*

We changed the colours used in the GABA data shown in Fig 3M (formerly K) to make the control group more easily visible.

10. *The authors make a good case for the alternative gaba synthesis pathway to be in SCN astrocytes, and the use of publicly available data is clever. However, is there a possibility astrocytes could switch pathways depending on time-of-day and availability of GABA? Was there any time-of-day info for the brains used in the Allen Brain Atlas?*

We thank the Reviewer for this helpful suggestion. There were only 2 circadian time points in the Allen Brain Cell Atlas data (day and night), and mice were kept under different LD conditions (12:12 or 14:10), thus making it difficult to reliably estimate a time-of-day difference.

To address to this comment, as well as related questions from R1 (#2 and #4), and strengthen our dataset, we have now analysed an independent scRNA-Seq dataset from Wen et al. *Nat Neurosci.* (2020), which is specifically restricted to SCN tissue and sampled across 12 circadian time points. Analyzing this dataset, we were able to confirm our key findings from the ABC Atlas regarding the specificity of expression of astrocyte vs neuronal GABA synthesis pathways (see new Figure 4 G to J), but also to investigate cell-type specific time-of-day expression. We found that expression of *Aldh1a1*, but not *Maob*, is significantly fluctuating with a peak at CT13 after 2 days in DD, consistent in timing with increasing extracellular GABA levels. This is also consistent with a third RNA-Seq data from Pembroke et al. *eLife* (2015), (wgpembroke.com/shiny/SCNseq). We did not observe an induction of the putrescine GABA synthesis pathway in neurons at any point during the circadian cycle, nor an induction of *GAD65/67* expression in astrocytes, suggesting that astrocytes and neurons do not switch GABA synthesis pathways across the circadian cycle.

In response to R1 comment #3, we also analyzed time-of-day-dependent regulation of genes involved in GABA uptake. SCN astrocytes have been shown to take up GABA from the extracellular space via GAT3 (Patton et al. *PNAS* 2023). In our analysis of, we find *Slc6a11*, which encodes GAT3, expression to be rhythmic in SCN astrocytes with a peak at CT6 (see Figure EV3H-J below). This independently confirms the rhythmic astrocytic expression of *Slc6a11* with peak during the circadian daytime reported by Patton et al. Thus, both the GABA synthesis and GABA uptake pathways in SCN astrocytes appear to be rhythmically regulated, peaking at opposite times of the circadian cycle. This may enable SCN astrocytes to switch between producing GABA during the circadian night (which inhibits neuronal activation), and taking GABA up from the extracellular space during the circadian day, thereby generating circadian rhythms of extracellular GABA. Inhibiting either of these pathways disrupts the rhythms of extracellular GABA in the SCN, as shown by our inhibition of GABA synthesis enzymes MAO-B and ALDH1A1, and the inhibition of GAT3-mediated uptake shown by Patton et al *PNAS* (2023).

Figure 4: The polyamine-to-GABA biosynthetic pathway is specifically expressed in hypothalamic astrocytes but not in SCN neurons. (G) UMAP plot of data of SCN-restricted scRNA-Seq dataset³⁴, with cell type annotation. (H) Normalized gene expression levels of genes involved in GABA biosynthesis. N= 12,018 SCN neurons and N=8,429 SCN astrocytes. Statistical test: two-way ANOVA with post-hoc Šidák's test. (I) Time series of normalized gene expression of neuronal GABA biosynthesis genes *Gad1* and *Gad2* in SCN neurons and astrocytes. EJTK Cycle rhythmicity test with Benjamini-Hochberg correction, all $p > 0.05$. (J) Time series of normalized gene expression levels of astrocytic GABA biosynthesis genes *Aldh1a1* and *Maob* in SCN neurons and astrocytes. EJTK Cycle rhythmicity test with Benjamini-Hochberg correction, $p=0.022$ for *Aldh1a1* in astrocytes peaking at CT13 (indicated as f on the plot), all other time series $p>0.05$. All data show mean \pm SEM, ns= non-significant, **= $p<0.01$, ****= $p<0.0001$.

Fig. EV3H-J: (H) Top panel shows percentage of SCN neurons and hypothalamic astrocytes expressing GABA transporters *Slc6a11*, *Slc6a1* or *Best1* in the Yao et al. scRNA-Seq dataset. Bottom panel shows normalized gene expression levels of each GABA transporter gene. N= 1,836 SCN neurons and N=20,549 hypothalamic astrocytes. (I) Normalized gene expression levels of GABA transporters in SCN neurons and SCN astrocytes from the Wen et al. (2020) scRNA-Seq dataset. N= 12,018 SCN neurons and N=8,429 SCN astrocytes. (J) Time series of normalized gene expression levels of *Slc6a11*, encoding GAT3, in SCN astrocytes and neurons. eJTK Cycle rhythmicity test with Benjamini-Hochberg correction, $p = 0.004$ with circadian peak at CT6 in astrocytes and not significantly rhythmic in neurons, as indicated. All graphs show mean \pm SEM, and panels H and I show two-way ANOVA with post-hoc Šidák's test, ** = $p<0.01$, **** = $p<0.0001$.

11. Did the authors do any checks on slice health or health of the astrocytes after multi-day treatment with pharmaceutical agents that alter basic metabolic pathways?
12. Can the authors be sure that these effects are specific to GABA metabolism and are not an effect of just making the slice or astrocytes unhealthy or reactive? The washout experiment is appreciated but doesn't really say anything about pathway specificity.

To address these two related points, we have now conducted additional experiments to confirm that the treatments employed do not affect the overall health of the SCN slices or induce astrocytic reactivity, and that they are specific to GABA production, as outlined below.

- i. **Slice health.** First, we analyzed the rhythmicity of individual cells within the SCN during A37 treatment and found no significant differences in the fraction of rhythmic Syn-jRCaMP1a or PER2::LUC-expressing cells (Figure 7D and K, shown below) during the treatment. These findings suggest that A37 specifically affects GABA oscillations without compromising the rhythmicity of neuronal calcium signals or PER2::LUC expression, indicating that the SCN slices remain viable and functionally competent during treatment.

All the recorded rhythms, including the extracellular GABA rhythm, fully recovered upon washout of the A37 treatment (Figure 6B-E, shown below), indicating there are no lasting effects, as it would be expected for a treatment that impairs slice viability.

Moreover, we have conducted new dose-response experiments for A37 (Fig. EV5, below), demonstrating that the reduction in GABA rhythm amplitude is strongly dose-dependent, and again fully reversed upon drug washout. This dose dependence further supports the specificity of the treatment's effect on GABA production rather than a general toxic effect. In contrast, and as a comparison, when a toxic concentration of A37 were used (100 μ M), the signal from all the rhythmic reporters (GABA and PER2::LUC) immediately and irreversibly flattened (Fig. EV5D).

Fig 7D and K: (D) Fraction of rhythmic Syn-jRCaMP1a cells across SCN slices before and after treatment with DMSO or A37 (n=200-400 cells of 4-5 SCN slices per condition). (K) Fraction of rhythmic PER2::LUC cells across SCN slices (n=150-350 cells of 4-6 SCN slices per condition).

Fig 6B-E: (B) Averaged, aligned time series of reporter of extracellular GABA (Syn-GABASnFR) in organotypic SCN slices before and after treatment with 25 μ M A37 (blue) or DMSO vehicle (black). N=4-6 SCN slices per condition. (C) Amplitude of first circadian cycle (over 30h) after treatment with A37 or DMSO relative to baseline. (D) RAE before and after treatment. (E) t values obtained from eJTK Cycle rhythmicity test on time series of Syn-GABASnFR before, after treatment and after washout of with A37 or DMSO vehicle. All graphs, including time series, show mean \pm SEM. Pairwise comparison in C shows two-tailed unpaired t-test. All other graphs show two-way mixed effects analysis with matching, with post-hoc Šidák's test. ns=non-significant, **= p <0.01.

Figure EV5: A37-mediated ALDH1A1 inhibition suppresses extracellular rhythms of GABA in a dose-dependent manner. (A) Representative time series of SCN slices expressing Syn-GABASnFR before and after treatment with increasing concentrations of A37 from left to right: DMSO (0 μ M A37), 10 μ M, 25 μ M and 50 μ M A37. (B) Amplitude of the first cycle (30h) of Syn-GABASnFR rhythms after treatment with increasing concentrations of A37 relative to baseline. One-way ANOVA, with post-hoc Tukey's t-test shown. N=3-6 SCNs per condition. (C) RAE of Syn-GABASnFR rhythms with A37 treatment and after washout of increasing concentrations of A37 relative to baseline. Mixed effects analysis with matching, time point effect p <0.01, A37 dose effect p <0.01, with post-hoc Sidak's test. Comparisons across time points are shown in corresponding color. (D) Representative time series of PER2::LUC and Syn-GABASnFR before and after treatment with 100 μ M A37, showing immediate tissue death. All graphs are mean \pm SEM. *= p <0.05, **= p <0.01.

ii. **Astrocyte reactivity/health.** We conducted new immunohistofluorescence experiments to evaluate GFAP and GABA levels in SCN slices treated with selegiline, the treatment that elicited the strongest effect on GABA rhythms, or vehicle (DMSO). Consistent with a treatment that impairs GABA production and opposite to what is expected for GABA uptake, we observed a significant reduction in GABA staining intensity (Fig. EV4A-B, shown below). We did not observe an increase in GFAP immunoreactivity, an established marker of reactive gliosis (Buffo et al., PNAS, 2008) (Fig. EV4A-B). We also pre-transduced SCN slices with AAVs expressing Gfap-mCherry::Cre to monitor any increase of astrocyte number with selegiline. We found no increase in the fluorescence intensity of these markers, or in the number of

astrocytes, which would be expected with a proliferative reactive astrocytosis (Buffo et al., PNAS, 2008) (Fig. EV4B-C).

- iii. **GABA specificity.** In addition to the significant reduction in GABA levels observed in the immunohistofluorescence experiments outlined above (Fig. EV4A-B), we observed that treatment with both selegiline and A37, while disrupting oscillations of extracellular GABA, did not impair the rhythmicity of Syn-jRCaMP1a or PER2::LUC signals, which exhibit robust rhythms throughout the treatment period. We have added this co-recorded data to new Figures 5 and 6 (shown below). Additionally, We also examined the effects of selegiline on extracellular glutamate rhythms, which are mediated by astrocytes in the SCN (Brancaccio et al., Neuron 2017). Unlike the abolishment of GABA rhythms with selegiline (Fig. 5C), general rhythmicity of extracellular glutamate rhythms was not affected (Fig. EV4E) and amplitude, robustness, or periodicity of the rhythms showed only mild, non significant effects (Fig. EV4D-H), further supporting the notion that the effects of selegiline are specific to GABA synthesis, and not due to a general toxic effect on astrocytes.

Taken together, these findings provide strong evidence that the effects of selegiline and A37 are specifically related to impaired GABA production, do not compromise SCN slice viability or induce astrocyte reactivity.

Figure EV4: Selegiline treatment decreases GABA concentration in SCN slices without significantly affecting GFAP immunoreactivity or circadian rhythms of extracellular glutamate. (A) Representative confocal images of fixed SCN slices expressing Gfap-mCherry::Cre, and stained with antibodies against GFAP and GABA, 4 days after treatment with Selegiline or DMSO. Scale bar = 200μm. (B) Quantification of mean fluorescence intensity of GFAP antibody, Gfap-mCherry::Cre and GABA antibody in SCN slices treated with Selegiline or DMSO. Two-way ANOVA with matching and post-hoc Šidák's test. (C) Number of Gfap-mCherry::Cre expressing cells per 1000μm² tissue in slices treated with DMSO or Selegiline. Two-tailed unpaired t-test. (D) Averaged, aligned time series of extracellular glutamate reporter (GfaABC1D-iGluSnFR) before and after treatment with 200μM Selegiline (teal) or DMSO (black). N=4-5 SCN slices per condition. (E) Left panel shows t values obtained from eJTK Cycle rhythmicity test on time series of GfaABC1D-iGluSnFR before and within 1-3 days after treatment with Selegiline or DMSO. Right panel shows the p-value obtained from eJTK Cycle rhythmicity test empirically calculated against random noise data. (F) GfaABC1D-iGluSnFR amplitude of first cycle of rhythms (over 30h) after treatment with Selegiline or DMSO relative to baseline. Two-tailed unpaired t-test. (G) RAE of GfaABC1D-iGluSnFR rhythms before and after Selegiline treatment. (H) Period of GfaABC1D-iGluSnFR rhythms before and after Selegiline treatment. Graphs E, G and H show two-way mixed effects analysis with matching, with post-hoc Šidák's test. All graphs, including time series, show mean ± SEM, except right panel in E which shows median ± interquartile range due to logarithmic scale, ns=non-significant, **=p<0.01.

Figure 5: Pharmacological inhibition of MAO-B abolishes circadian rhythms of extracellular GABA in SCN slices and shortens the circadian period of neuronal calcium and clock gene expression. (A) Schematic of Selegiline action on polyamine GABA biosynthesis in astrocytes. (B) Averaged, aligned time series of Syn-GABASnFR in SCN slices before and after treatment with 200 μ M Selegiline (blue) or DMSO vehicle (black). N=4-5 SCNs per condition. (C) Left panel shows t values obtained from eJTK Cycle rhythmicity test on time series of Syn-GABASnFR before and within 1-3 days after treatment with Selegiline or DMSO. Right panel shows the p-value obtained from eJTK Cycle rhythmicity test empirically calculated against random noise data. (D) Averaged, aligned time series of neuronal calcium (Syn-jRCaMP1a) before and after treatment with 200 μ M Selegiline (red) or DMSO (black). N=7-8 SCN slices per condition. (E) t values obtained from eJTK Cycle rhythmicity test on time series of Syn-jRCaMP1a before and within 1-3 days after treatment with Selegiline or DMSO. (F) Syn-jRCaMP1a amplitude of first cycle of rhythms (over 30h) after treatment with Selegiline or DMSO relative to baseline. (G) RAE of Syn-jRCaMP1a rhythms before and after treatment. (H) Period of Syn-jRCaMP1a rhythms before and after treatment. (I) Averaged, aligned time series of PER2::LUC before and after treatment with 200 μ M Selegiline (purple), or DMSO (black). N=4-5 SCNs per condition. (J) t values obtained from eJTK Cycle rhythmicity test on time series of PER2::LUC before and within 1-3 days after treatment with Selegiline or DMSO. (K) PER2::LUC amplitude of first cycle of rhythms (over 30h) after treatment with Selegiline or DMSO, relative to baseline. (L) RAE of PER2::LUC rhythms before and after Selegiline treatment. (M) Period of PER2::LUC rhythms before and after Selegiline treatment. All graphs, including time series, show mean \pm SEM, except right panel in C, showing median \pm interquartile range due to logarithmic scale. Pairwise comparison in F and K show two-tailed unpaired t-test. All other graphs show two-way mixed effects analysis with matching, with post-hoc Šidák's test. ns=non-significant, * p <0.05.

Figure 6: Pharmacological inhibition of ALDH1A1 temporarily suppresses rhythms of extracellular GABA in SCN organotypic slices. (A) Schematic of A37 action on polyamine GABA biosynthesis in astrocytes. (B) Averaged, aligned time series of reporter of extracellular GABA (Syn-GABASnFR) in organotypic SCN slices before and after treatment with 25µM A37 (blue) or DMSO vehicle (black). N=4-6 SCN slices per condition. (C) Amplitude of first circadian cycle (over 30h) after treatment with A37 or DMSO relative to baseline. (D) RAE before and after treatment, and after washout. (E) t values obtained from eJTK Cycle rhythmicity test on time series of Syn-GABASnFR before, after treatment and after washout of A37 or DMSO vehicle. (F) Period of Syn-GABASnFR rhythms before and after treatment. (G) Averaged, aligned time series of co-detected neuronal calcium (Syn-jRCaMP1a) before and after treatment with A37 (red) or DMSO (black). N=4-6 SCN slices per condition. (H) Amplitude of first cycle of rhythms (over 30h) after treatment with A37 or DMSO relative to baseline. (I) RAE of Syn-jRCaMP1a relative to baseline after treatment and washout of A37 or DMSO. (J) Period of Syn-jRCaMP1a rhythms before and after treatment. (K) Averaged, aligned time series of co-detected PER2::LUC before and after treatment with A37 (purple) or DMSO (black). N=4-6 SCN slices per condition. (L) Amplitude of first cycle of rhythms (over 30h) after treatment with A37 or DMSO relative to baseline. (M) RAE of PER2::LUC relative to baseline after treatment and washout of A37 or DMSO. (N) Period of PER2::LUC rhythms before and after treatment. All graphs, including time series, show mean±SEM. Pairwise comparison in C, H and L show two-tailed unpaired t-test. All other graphs show two-way mixed effects analysis with matching, with post-hoc Šidák's test. ns=non-significant, *p<0.05, **=p<0.01.

13. I think the biggest conceptual sticking point I have is what if it is GABA buffering by astrocytes that is regulated by time-of-day, as buffering neurotransmitters and extracellular ions are a fundamental aspect of astrocyte function, instead of production of GABA via a non-canonical pathway? Most of the data in this paper could be explained by uptake just as much as release from astrocytes, and it may fit in with the astrocytic literature better. Use of more specific tools to prevent production, or experiments critically testing uptake, would be needed. As of right now, I'm not sure the results are interpretable one way or another.

We thank the Reviewer for their comment. While we do not dispute existing literature on GABA uptake as a more established mechanism for GABA regulation (Ishibashi et al., *Int. J. Mol. Sci.* 2019), we note that this is especially true at much shorter timescales. What happens on the multi-hour daily timescale addressed in our study is far less characterized and understood, with only one recent report functionally addressing GABA uptake by astrocytes on a circadian scale (Patton et al. 2023). While we contribute new evidence that GABA synthesis by astrocytes is critically important to generate extracellular GABA rhythms, this does not rule out contributions from GABA uptake. In fact, in the revised manuscript we provide evidence that GABA rhythms could emerge by concerted daytime uptake of synaptic GABA and nighttime astrocyte GABA synthesis, as outlined below.

A first general consideration is that we have used two independent compounds which are highly selective for two molecularly unrelated enzymes, previously shown by others to be involved in GABA synthesis: A37 (also known as CM037) for ALDH1A1 (Morgan and Hurley, *Journal of Medicinal Chemistry* 2015; Bhalla et al., *BioRxiv* 2023), and selegiline (also known as deprenyl) inhibiting MAO-B (Magyar et al., 1967; Knoll et al., 1978; Park et al. *Science* 2019; Srivastava et al., *Cells* 2020), and not GABA transporters (Yoon et al., *Journal Physiol.* 2014). Therefore we have no expectations that either of these independent compounds would target GABA uptake, given their established selectivity. While molecularly unrelated, however, treatment with these two compounds leads to a similar circadian phenotype, as measured by several readouts (see answer to previous comments #11 and 12), and consistent with their biochemical convergent role within the putrescine to GABA biosynthetic pathway.

We present new data showing a significant reduction of GABA in SCN slices treated with selegiline by additional immunohistofluorescence experiments (Fig. EV4A-B, below), as expected for a treatment that interferes with the production of GABA, rather than blockade of uptake. In contrast, previous studies involving pharmacological inhibition of GABA uptake by astrocytes (Patton et al., *PNAS* 2023), or astrocyte-specific *Bmal1* KO models, leading to reduced GAT1 and GAT3 (Barca-Mayo et al., *Nat Comms* 2017), show increased GABA A-receptor mediated signalling and an accumulation of extracellular GABA.

Finally, we have now shown that rhythms of extracellular glutamate, produced by astrocytes in the SCN (Brancaccio et al., *Neuron* 2017), are only mildly (and not significantly) affected by selegiline treatment, thus ruling out a generic effect of altered transmitter buffering by astrocytes (Fig. EV4 D to H, below)

The Reviewer's comment has however prompted to also evaluate GABA uptake within our working model of circadian regulation of extracellular GABA rhythms in the SCN. By using the SCN-specific scRNA-Seq dataset from Wen et al. *Nat Neurosci.* (2020), (also confirmed in a second dataset- wgpembroke.com/shiny/SCNseq, Pembroke et al., *eLife*, 2015), we have been able to show that *Aldh1a1* expression has a circadian oscillation which peaks at nighttime (CT13) (Fig 4J, below). This is consistent with the nighttime peak of extracellular

GABA measured in our live imaging experiments. On the other hand, we found that the astrocytic GABA transporter GAT3 (*Slc6a11*) is also rhythmically expressed in SCN astrocytes, but with an opposite daytime peak (CT6), consistent with the idea that low levels of extracellular GABA measured in our experiments may be mediated by astrocytic uptake of synaptically released GABA when GABAergic SCN neurons are maximally active (see Fig. EV3 below). This independently confirms the rhythmic astrocytic expression of *Slc6a11* with peak during the circadian daytime previously reported by Patton et al. *PNAS* 2023.

Thus, both the GABA synthesis and GABA uptake pathways in SCN astrocytes appear to be rhythmically regulated, peaking at opposite times of the circadian cycle. This temporal regulation indicates a dynamic switch in SCN astrocytes, from GABA production at night to GABA uptake during the day, thereby generating robust circadian oscillations of extracellular GABA levels. Disruption of either pathway, via inhibition of GABA synthesis enzymes MAO-B or ALDH1A1, as shown in our study, or GAT3 inhibition as reported by Patton et al., leads to dysregulated circadian cycling of GABA.

We have added a more detailed discussion of these results to the discussion section, as follows:

“Notably, pharmacological inhibition of GAT3 in SCN slices leads to an accumulation of extracellular GABA, showing that GAT3 mediates GABA uptake in the SCN (Patton *et al*, 2023). If astrocytes can regulate extracellular GABA rhythms both via synthesis and uptake, how are GABA levels regulated by astrocytes across the circadian day? Using the scRNA-Seq dataset from (Wen *et al*, 2020), we found that *Aldh1a1* expression peaks at CT13 (Fig. 4J), consistent with an independent dataset from (Pembroke *et al*, 2015) (wgpembroke.com/hiny/SCNseq). In contrast, *Slc6a11*, which encodes GAT3, peaked at CT6 in SCN astrocytes, consistent with (Patton *et al*, 2023) (Fig. EV3J). This suggests that astrocytes may generate extracellular GABA rhythms by switching from increased daytime GAT3-mediated GABA uptake, removing synaptically released GABA, to nighttime astrocytic GABA synthesis, replenishing extracellular GABA levels and inhibiting SCN neurons. This daily astrocyte switch will generate the circadian oscillations of extracellular GABA observed here and by (Patton *et al*, 2023). Disruption of either GABA production or uptake leads to a dysregulation of extracellular GABA rhythms due to low (Fig EV4B), or excess extracellular GABA (Patton *et al*, 2023), both ultimately disrupting circadian cycling of extracellular GABA tone. “

Figure EV4: Selegiline treatment decreases GABA concentration in SCN slices without significantly affecting GFAP immunoreactivity or circadian rhythms of extracellular glutamate. (A) Representative confocal images of fixed SCN slices expressing Gfap-mCherry::Cre, and stained with antibodies against GFAP and GABA, 4 days after treatment with Selegiline or DMSO. Scale bar = 200 μm. (B) Quantification of mean fluorescence intensity of GFAP antibody, Gfap-mCherry::Cre and GABA antibody in SCN slices treated with Selegiline or DMSO. Two-way ANOVA with matching and post-hoc Šidák's test. (C) Number of Gfap-mCherry::Cre expressing cells per 1000 μm² tissue in slices treated with DMSO or Selegiline. Two-tailed unpaired t-test. (D) Averaged, aligned time series of extracellular glutamate reporter (GfaABC1D-iGluSnFR) before and after treatment with 200 μM Selegiline (teal) or DMSO (black). N=4-5 SCN slices per condition. (E) Left panel shows t values obtained from eJTK Cycle rhythmicity test on time series of GfaABC1D-iGluSnFR before and within 1-3 days after treatment with Selegiline or DMSO. Right panel shows the p-value obtained from eJTK Cycle rhythmicity test empirically calculated against random noise data. (F) GfaABC1D-iGluSnFR amplitude of first cycle of rhythms (over 30h) after treatment with Selegiline or DMSO relative to baseline. Two-tailed unpaired t-test. (G) RAE of GfaABC1D-iGluSnFR rhythms before and after Selegiline treatment. (H) Period of GfaABC1D-iGluSnFR rhythms before and after Selegiline treatment. Graphs E, G and H show two-way mixed effects analysis with matching, with post-hoc Šidák's test. All graphs, including time series, show mean ± SEM, except right panel in E which shows median ± interquartile range due to logarithmic scale, ns=non-significant, **=p<0.01.

Fig. EV3H-J: (H) Top panel shows percentage of SCN neurons and hypothalamic astrocytes expressing GABA transporters *Slc6a11*, *Slc6a1* or *Best1* in the Yao et al. scRNA-Seq dataset. Bottom panel shows normalized gene expression levels of each GABA transporter gene. N= 1,836 SCN neurons and N=20,549 hypothalamic astrocytes. (I) Normalized gene expression levels of GABA transporters in SCN neurons and SCN astrocytes from the Wen et al. (2020) scRNA-Seq dataset. N= 12,018 SCN neurons and N=8,429 SCN astrocytes. (J) Time series of normalized gene expression levels of *Slc6a11*, encoding GAT3, in SCN astrocytes and neurons. eJTK Cycle rhythmicity test with Benjamini-Hochberg correction, $p = 0.004$ with circadian peak at CT6 in astrocytes and not significantly rhythmic in neurons, as indicated. All graphs show mean \pm SEM, and panels H and I show two-way ANOVA with post-hoc Šidák's test, ** = $p < 0.01$, **** = $p < 0.0001$.

Dear Dr. Brancaccio,

Congratulations on a great revision! Overall, the referees have been positive. However, there is one remaining concern that we ask you to (non-experimentally) address in a new revision. When you submit your revised version, please also take care of the following editorial items and add this also to your point-by-point response:

1. Please remove the author contribution section from the main manuscript.
2. In the appendix file, please remove the headings for "Supplementary information"
3. Please remove the reagent and tools table from the main manuscript and upload as individual file using the template provided in our author guidelines.
4. We require the publication of source data, particularly for electrophoretic gels and blots and graphs, with the aim of making primary data more accessible and transparent to the reader. It would be great if you could provide me with a PDF file per figure that contains the original, uncropped and unprocessed scans of all or key gels used in the figure or for graphs, an Excel spreadsheet with the original data used to generate the graphs. The PDF files should be labeled with the appropriate figure/panel number, and should have molecular weight marker; further annotation could be useful but is not essential. The PDF files will be published online with the article as supplementary "Source Data" files.
5. We include a synopsis of the paper (see <http://emboj.embopress.org/>). Please provide me with a general summary statement and 3-5 bullet points that capture the key findings of the paper.
6. We also need a summary figure for the synopsis. The size should be 550 wide by 200-440 high (pixels). You can also use something from the figures if that is easier.
7. Please remove the line from the Acknowledgement about Biorender and add it to the Methods in a subsection with the heading "Graphics"
8. Please remove the conflict of interest from the first page of the manuscript text.
9. Please remove the movie legends from the appendix (also the table of contents) and zip them to the corresponding movie file. Correct nomenclature is "Movie EV1" etc.
10. Please note that in legend of figure 6, description of figure panel "N" is mislabelled as "J". Please look into it.
11. Please note that the exact p values are not provided in the legends of figures 1D, 1F, 2E, 2F, 2G, 2H, 3C, 3E, 3L, 4F, 4H, 5C, 5H, 5K, 5L, 5M, 6C, 6D, 6J, 7B, 7I, EV2D, EV2E, EV2G, EV3H, EV3I, EV4B, EV5B, EV5C.
12. Please note that information related to n is missing in the legends of figures 3F, 3G, 3L, 4I, 4J, 5C, 5E, 5F, 5G, 5H, 5J, 5K, 5L, 5M, 6C, 6D, 6E, 6F, 6H, 6I, 6J, 6L, 6M, 6N, 7D, 7L, EV3C, EV3F, EV3J, EV4B, EV4C, EV4E, EV4F, EV4G, EV4H, EV5C.
13. Although 'n' is provided, please describe the nature of entity for 'n' in the legend of figures 2G, 3E, EV2D.
14. Please note that the scale bar needs to be defined for figure EV1D.

Thank you for the opportunity to consider your work for publication. I look forward to your revision.

Yours sincerely,

Kelly M Anderson, PhD
Editor, The EMBO Journal
k.anderson@embojournal.org

Use the link below to submit your revision:

Referee #1:

The revised version of the article entitled "Astrocytic GABA produced from polyamines synchronizes neuronal circadian timekeeping in the suprachiasmatic nucleus" by Brancaccio and colleagues provides evidence of a new role of astrocytes in the suprachiasmatic nuclei, the master circadian pacemaker in mammals. The authors carefully considered and responded to all concerns, provide more data and explanations that have strengthen their initial conclusions. I recommend the acceptance of the manuscript in it's current version.

Referee #2:

The authors answered my questions and I have no further comments.

Referee #3:

In this revised manuscript, the authors suggest that a non-canonical GABA production pathway from astrocytes mediates extracellular GABA rhythmicity across the clock center of the brain, the SCN. The authors have responded well to the bulk of my previous critiques. The only minor point is that the SCN does not normally stain for GFAP, so using that as a marker for astrocyte reactivity may not be feasible. However, this does not invalidate their other slice health assays.

Response to Review

Editor's comments:

1. Please remove the author contribution section from the main manuscript.

We have now removed the author contribution section.

2. In the appendix file, please remove the headings for "Supplementary information"

The headings have now been removed.

3. Please remove the reagent and tools table from the main manuscript and upload as individual file using the template provided in our author guidelines.

The table has been removed from the main manuscript and attached as a separate word file.

4. We require the publication of source data, particularly for electrophoretic gels and blots and graphs, with the aim of making primary data more accessible and transparent to the reader. It would be great if you could provide me with a PDF file per figure that contains the original, uncropped and unprocessed scans of all or key gels used in the figure or for graphs, an Excel spreadsheet with the original data used to generate the graphs. The PDF files should be labeled with the appropriate figure/panel number, and should have molecular weight marker; further annotation could be useful but is not essential. The PDF files will be published online with the article as supplementary "Source Data" files.

All source data used to generate the graphs in each figure has now been added as an excel file. Each sheet contains the data and, where appropriate, statistical tests for each figure panel.

5. We include a synopsis of the paper (see <http://emboj.embopress.org/>). Please provide me with a general summary statement and 3-5 bullet points that capture the key findings of the paper.

Summary Statement

Astrocyte-produced GABA acts as a nighttime signal that synchronizes the neuronal circuit and contributes to the generation of circadian rhythms in the suprachiasmatic nucleus.

Synopsis

Astrocytes can drive circadian behavior in mammals, however, the nature of the temporal information generated by astrocytes is largely unknown. This study identifies GABA produced by polyamine degradation in astrocytes as a critical signal that synchronizes neuronal activity in the suprachiasmatic nucleus (SCN), which orchestrates circadian rhythms in mammals.

- *Circadian rhythms of astrocytic activity have a homogeneous phase across the SCN, as opposed to phase waves of neuronal activity*
- *Extracellular GABA mirrors the spatiotemporal organization of astrocytic rhythms peaking during the nighttime, as opposed to the daytime peak of neuronal activity*
- *Inhibition of synaptic GABA release desynchronizes neuronal circadian rhythms in the SCN, but does not affect extracellular GABA rhythms*

- *Inhibition of astrocytic GABA synthesis disrupts circadian rhythms of extracellular GABA and desynchronizes neuronal circadian activity, suggesting a role as an internal circadian synchronizer for the SCN circuit ("astrozeit")*

6. We also need a summary figure for the synopsis. The size should be 550 wide by 200-440 high (pixels). You can also use something from the figures if that is easier.

We have submitted a summary figure for the synopsis (also below).

7. Please remove the line from the Acknowledgement about Biorender and add it to the Methods in a subsection with the heading "Graphics"

We have now removed this from Acknowledgements and added into a Methods subsection title Graphics instead.

8. Please remove the conflict of interest from the first page of the manuscript text.

This has now been removed.

9. Please remove the movie legends from the appendix (also the table of contents) and zip them to the corresponding movie file. Correct nomenclature is "Movie EV1" etc.

These have now been removed, nomenclature changed and movie legends zipped with the movie files

10. Please note that in legend of figure 6, description of figure panel "N" is mislabelled as "J". Please look into it.

Thanks for pointing this out. We have now corrected the figure legend to reflect the correct panel number.

11. Please note that the exact p values are not provided in the legends of figures 1D, 1F, 2E, 2F, 2G, 2H, 3C, 3E, 3L, 4F, 4H, 5C, 5H, 5K, 5L, 5M, 6C, 6D, 6J, 7B, 7I, EV2D, EV2E, EV2G, EV3H, EV3I, EV4B, EV5B, EV5C.

Exact p-values for significant differences are now specified in the figure legend. In addition, we have added a table in the Appendix (Appendix Table S1), which includes the details of all

the statistical tests performed, the overall mean differences, and the exact p-values for all the post hoc multiple comparison tests performed, regardless of their significance. Where applicable, we have also added reference to this Table in both figure legends and main text, so the interested reader can easily access a fully detailed report of the statistical tests performed and their outcomes, without sacrificing general readability. Finally, all the statistical tests performed and relative outcomes are also included in the source data provided.

Exact p-values are not reported for $p < 0.0001$ in line with the standard reporting for most scientific statistics programs such as GraphPad Prism, as most methods for computing p values are not numerically accurate below a certain point and reporting extremely small p-values can suggest a misleading level of precision without adding meaningful insight (for discussion, see Greenland et al. European Journal of Epidemiology 2016).

12. Please note that information related to n is missing in the legends of figures 3F, 3G, 3L, 4I, 4J, 5C, 5E, 5F, 5G, 5H, 5J, 5K, 5L, 5M, 6C, 6D, 6E, 6F, 6H, 6I, 6J, 6L, 6M, 6N, 7D, 7L, EV3C, EV3F, EV3J, EV4B, EV4C, EV4E, EV4F, EV4G, EV4H, EV5C.

This information has now been added to the figure legend.

For Figures 3E-L, the same experimental samples are shown throughout the figure, so the N number has been added to the end of the figure legend as follows: "Figure panels E, F, G, L and M show the same experimental samples, N=5 SCN slices per condition for PER2::LUC and N=8 SCN slices per condition for Syn-GABASnFR."

For Figure 5, 6 and EV4, the same experimental samples are shown across multiple panels for each reporter. This has now been specified in the first panel in which it occurs for each reporter, as follows: "N=7-8 SCN slices per condition for panels D-H."

13. Although 'n' is provided, please describe the nature of entity for 'n' in the legend of figures 2G, 3E, EV2D.

In 2G, 3E and EV2D the nature of 'N' is now specified as SCN slices. For Figure 3E, this has been done at the end of the figure legend (see response to Comment #12 above).

14. Please note that the scale bar needs to be defined for figure EV1D.

Thank you for pointing this out. We have now added the scale bar definition to the figure legend.

Reviewer's comments:

Referee #1:

The revised version of the article entitled "Astrocytic GABA produced from polyamines synchronizes neuronal circadian timekeeping in the suprachiasmatic nucleus" by Brancaccio and colleagues provides evidence of a new role of astrocytes in the suprachiasmatic nuclei, the master circadian pacemaker in mammals.

The authors carefully considered and responded to all concerns, provide more data and

explanations that strengthened their initial conclusions. I recommend the acceptance of the manuscript in its current version.

We thank the Referee for their positive evaluation of our work

Referee #2:

The authors answered my questions and I have no further comments.

We thank the Referee for their positive evaluation of our work

Referee #3:

In this revised manuscript, the authors suggest that a non-canonical GABA production pathway from astrocytes mediates extracellular GABA rhythmicity across the clock center of the brain, the SCN. The authors have responded well to the bulk of my previous critiques. The only minor point is that the SCN does not normally stain for GFAP, so using that as a marker for astrocyte reactivity may not be feasible. However, this does not invalidate their other slice health assays.

We thank the Referee for their positive evaluation of our work. While they are generally satisfied with our response, they raise a minor point, which we address below.

GFAP is a well-established marker of astrocytosis, not only in various brain regions, but also specifically in the SCN. The SCN expresses significant levels of GFAP in physiological conditions as shown by reports by several other groups, (e.g. Lavialle and Serviere, Neuroreport 1993, Becquet et al. Glia 2008, Leone J. of Neurosci. Res. 2006), as well as ourselves (Brancaccio et al. Neuron 2017). Moreover, GFAP also is a marker for astrocyte reactivity in the SCN, similarly to other brain regions, in both mice and humans (e.g. Eeza et al., Journal of Alzheimer's Disease; Stopa et al. 1999, J Neuropathol Exp Neurol. We have further clarified in the main text, by adding a short paragraph including these references.

Additional comment: In revising the manuscript, we noticed that in Figure EV3, the graphs in panel (H) bottom and panel (I) were accidentally interchanged. They are from separate scRNA-Seq datasets, but show very similar results. We have now rectified it.

Dear Marco,

Congratulations on an excellent manuscript, I am very pleased to inform you that it has been accepted for publication in The EMBO Journal! Thank you for your comprehensive responses to the referees' concerns and for addressing our additional editorial requests.

Your manuscript will be processed for publication by EMBO Press. There are a few minor issues with some of the figure legends that we will ask you to address, and we might also send you a few suggestions for textual improvement of the abstract and the synopsis text. Your manuscript will then be copy edited and you will receive page proofs prior to publication. Please note that you will be contacted by Springer Nature Author Services to complete licensing and payment information.

If you have any questions, please do not hesitate to contact the Editorial Office. Thank you very much for your contribution to The EMBO Journal, working with you has been a pleasure!

Best wishes,

Ioannis
